# On Powerful Ways to Generate: Autoregression, Diffusion, and Beyond

**Chenxiao Yang**[†]   **Cai Zhou**[§]   **David Wipf**[‡]   **Zhiyuan Li**[†]

† Toyota Technological Institute at Chicago   § Massachusetts Institute of Technology
‡ School of Computing & Data Science, The University of Hong Kong
{chenxiao,zhiyuanli}@ttic.edu

## Abstract

Diffusion language models have recently emerged as a competitive alternative to autoregressive language models. Beyond next-token generation, they are more efficient and flexible by enabling parallel and any-order token generation. However, despite empirical successes, their computational power and fundamental limitations remain poorly understood. In this paper, we formally study whether non-autoregressive generation in Masked Diffusion Models (MDM) enables solving problems beyond the reach of Auto-Regressive Models (ARM). Our results show that MDM with sufficiently large context length is computationally universal with decoding steps matching the optimal parallel time complexity in PRAM. However, when controlling for other factors, MDM's flexibility to generate in any-order does not expand what ARM can already solve. To address this, we propose a new form of generation called any-process generation, which extends MDM with capabilities to remask, insert and delete tokens, allowing self-correction, length-variable editing, and adaptive parallelism. Theoretically and empirically, we demonstrate these capabilities enable scalability to significantly harder reasoning problems that are otherwise intractable for ARM and vanilla MDM. Additionally, they prove essential for generation tasks where objects naturally evolve through non-sequential processes, crucial for extending current LLMs beyond natural language to domains such as coding and science.

## 1 Introduction

The underlying generation process of almost everything in nature follows a unidirectional arrow of time. Perhaps most representative of all, spoken language is produced through a sequential process where each word builds upon preceding context in causal temporal order. This generic inductive bias has been encoded into *Auto-Regressive Models (ARM)* (Shannon, 1951), through next-token generation. Despite its simplicity, ARM when scaled through training on vast corpora, has produced remarkably powerful models capable of general-purpose task completion and reasoning, like GPT (Radford et al., 2018; 2019; Brown et al., 2020; Achiam et al., 2023).

Yet reality might be more convoluted. Humans, when tackling challenging tasks, naturally undergo a non-sequential process of searching for solutions, evaluating and refining them, backtracking when needed, and iterating until answers are found. Such complexity is not fully captured by current ARM. While it is debatable if human intelligence fundamentally follows this left-to-right process (LeCun, 2023; Malach, 2023; Bachmann & Nagarajan, 2024; Berglund et al., 2023; Nagarajan et al., 2025), ARM appears increasingly ill-suited when we venture beyond natural language.

For example, code generation must be subject to global constraints like balanced parentheses and well-typedness. Maintaining validity at each intermediate step makes transitions from one state to another easier, thus naturally involving updates such as inserting functions, adding branches, or changing input types. In biology, many domains remain largely beyond the reach of current LLMs, as molecular structures such as proteins and genes are combinatorial objects that can be modeled as graphs, trees, or strings that satisfy physical constraints. Their generation proceeds most naturally through structure-aware edits, e.g., swapping protein domains, inserting binding motifs into sequence graphs, or recombining DNA/RNA segments (Wang et al., 2023).

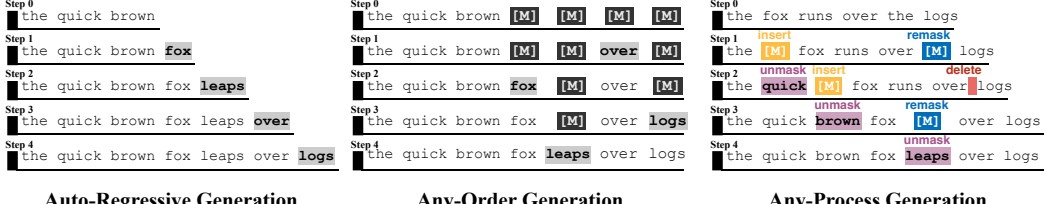

Figure 1: Comparison between autoregressive generation, any-order generation (standard MDM) and any-process generation (our MDM).

Given the long-standing pursuit of building foundation models powerful enough to handle increasingly complex reasoning tasks and general enough to work across diverse domains beyond natural language, it becomes important and timely to rethink the generation process itself, as a mechanism separate from architectural specifics, by formally asking:

*How do we formally compare various ways to generate, and what opportunities may lie beyond next-token generation?*

Recent work suggests that next-token generation is not the only viable path. *Masked Diffusion Models (MDM)* (Hoogeboom et al., 2021; Austin et al., 2021; Lou et al., 2024; Sahoo et al., 2024; Shi et al., 2024) offer a compelling alternative procedure that, instead of causally generating tokens one by one, permits any-order generation and produces multiple tokens in parallel, with recent large-scale instantiations (DeepMind, 2025; Labs et al., 2025; Nie et al., 2025; Ye et al., 2025) showing comparable performance with AR-based LLMs. Interestingly, besides faster decoding (up to $10\times$ speedups), MDM's generation process brings empirical improvements on some order-sensitive tasks such as reversed-order poem completion (Nie et al., 2025; Berglund et al., 2023) and Sudoku puzzles (Kim et al., 2025c; Shah et al., 2024). This motivates us to formally study it and compare with ARM.

Perhaps counterintuitively, we find **while MDM is indeed more powerful than ARM in terms of parallelism and efficiency for simple tasks, the benefits of seemingly greater flexibility are surprisingly limited.** Like ARM (Merrill & Sabharwal, 2024; Feng et al., 2024; Li et al., 2024), MDM also achieves Turing-completeness, but does so more efficiently with optimal parallel time complexity (Theorem 1), thus enabling *exponential* speedups for simple parallelizable problems. However, for harder reasoning tasks, MDM faces similar fundamental limitations as ARM: both struggle with problems requiring backtracking and rewriting capabilities, and cannot handle them given realistic space resources (Theorem 2). Moreover, when controlling for other factors including degree of parallelism and architecture, any-order generation itself does not expand what ARM can already handle (Theorem 3), since any computation performed by MDM can be reorganized into left-to-right order to align with the underlying arrow of time. Therefore we ask:

*What are provably more powerful ways to generate?*

As an initial step, we propose **Any-Process Generation**, inspired by natural generative mechanisms found across domains. It extends standard MDM beyond its existing **unmask** capability with three additional operations (see Figure 1): **remask** (converting decoded tokens back to masks), **insert** (adding new mask tokens at any position), and **delete** (removing mask tokens), all learned end-to-end from data without architectural changes. Freed from conventional physics-inspired diffusion frameworks, any-process generation removes unnecessary restrictions on mask ratios, decoding steps, sequence lengths and stopping criteria, enabling structural editing and test-time scaling. With these modifications, we show that MDM brings significant promise with both encouraging theoretical and empirical results as follows.

**Scalability to Hard Problems:** The capability to rewrite and backtrack breaks the non-erasable limitations of ARM and standard MDM, enabling our model to achieve both optimal parallel time and space complexity (Theorem 4), thus solving many NP-hard problems with polynomial space through test-time scaling, i.e. an exponential improvement from P achieved by ARM and standard MDM. Empirically, on Sudoku puzzles (Figure 2(a)), our model achieves 99.28% accuracy using only 100 training instances, outperforming ARM (87.18%) and any-order MDM (89.49%) with $5\times$ parameters trained on 1.8M instances, which is orders of magnitude more.

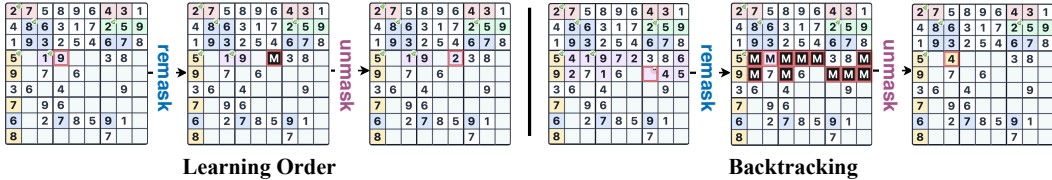

(a) **Sudoku** (NP-Complete Problem). Scaling up inference-time computes to solve significantly harder problems by allowing rewrites and backtracking using the **remask** operation (§ 5.1).

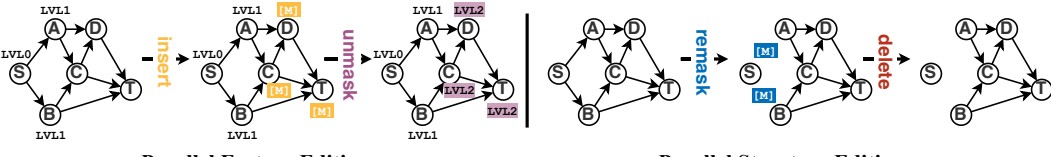

(b) **Coding** (Matched Parentheses / Dyck-$k$). Generating any-sized two-sided Dyck-$k$ is impossible for ARM, while our model can easily do so with the **insert** operation (§ 5.2).

(c) **Parity** (Counting 1s). Length generalizing parity and counting problems are enabled by learning a simple elimination algorithm with the **remask** and **delete** operations (§ 5.3).

(d) **DNA Recombination** (Splicing System). An example in science / biology where DNA segments are spliced and pasted using combinations of operations, which is hard for ARM (§ 5.2).

(e) **Graph Editing.** Editing combinatorial structures where feature / structural evolution and parallel computation are naturally integrated (§ 5.3). Using ARM to simulate is hard (§ 5.4).

Figure 2: Examples of any-process generation for different tasks.

**Generality to Non-Sequential Objects:** The flexibility to rewrite, insert, and delete tokens enables structure-aware generation processes for objects that inherently resist sequential construction, such as DNA recombination in biology (Figure 2(d)) and 2D graph generation (Figure 2(e)). This advantage can be formalized through a simple task of matched parentheses generation (two-sided Dyck-k language), i.e. one of the most basic constraints in coding. Theorem 5 proves that ARM cannot cover the entire support beyond a length threshold, because enforcing the generation into a left-to-right order demands global foresight of future tokens that is beyond the computational power of constant-depth Transformers. In contrast, AP-MDM can easily do so at arbitrary length using insert operations (Figure 2(b)). Empirically, we verify the structural generation capability of AP-MDM on a challenging graph editing task. The results show that our approach maintains perfect accuracy for increasingly larger graphs, while ARM performance degrades significantly as graph size increases.

**Learning and (OOD) Generalization:** Our approach enables learning previously-impossible simpler algorithms that significantly improve learning and generalization. For parity checking (Figure 2(c)), our model achieves 100% generalization to arbitrary lengths after training on only length-2 sequences, while even the latest GPT models struggle on this embarrassingly simple task.

Finally, envisioning a future with access to data of the underlying generation processes of objects we wish to generate, such as code revisions, math proof drafts, or molecular formation processes, any-process MDM is theoretically and empirically more suitable than ARM (Theorem 6) since AP-MDM is hard to be simulated by ARM due to the complexity introduced by editing operations.

## 2 PRELIMINARY

**Auto-Regressive Model (ARM)** Let $\Sigma$ be a finite-sized vocabulary and $\pi : \Sigma^* \to \Sigma$ be a next-token predictor, which maps a sequence $\mathbf{x} = (x_1, x_2, \cdots, x_n) \in \Sigma^n$ to a token $x_{n+1} \in \Sigma$. An autoregressive model (ARM) is defined based on a sequence-to-sequence mapping $f : \Sigma^* \to \Sigma^*$, concatenating input sequence $\mathbf{x}$ and the next token $\pi(\mathbf{x})$, i.e. $f(\mathbf{x}) = (\mathbf{x}, \pi(\mathbf{x}))$. ARM formulates generation as an iterative process by repeatedly applying $f$ to the current sequence. In practice, $f$ is typically parameterized by a Transformer (Vaswani et al., 2017) with causal attention and learnable parameters $\theta$. The notion of ARM here also aligns with *Chain-of-Thought (CoT)* (Wei et al., 2022) in many other works and we will use them interchangeably throughout this paper.

**Masked Diffusion Model (MDM)** Let $\bar{\Sigma} = \Sigma \cup \{\mathsf{M}\}$ be the extended vocabulary where $\mathsf{M}$ is an absorbing mask token (Austin et al., 2021). Consider sequences $\mathbf{x}_t = (x_{t,1}, x_{t,2}, \ldots, x_{t,S}) \in \bar{\Sigma}^S$ indexed by time $t \in [T]$, where $S$ is the maximum context length, $T$ is the number of decoding steps, $\mathbf{x}_0 = \{\mathsf{M}\}^S$ is the fully masked sequence and $\mathbf{x}_T \in \Sigma^S$ is the target clean sequence.[1] A masked diffusion model (MDM) (Lou et al., 2024; Sahoo et al., 2024; Shi et al., 2024) also relies on a sequence-to-sequence mapping $f : \bar{\Sigma}^S \to \bar{\Sigma}^S$ with $\mathbf{x}_{t+1} = f(\mathbf{x}_t)$, formulating generation as an iterative process by repeatedly applying $f$ to progressively unmask tokens from the all-mask state.

Among many MDM variants, we consider the following standard design choices from recent large language diffusion models (Nie et al., 2025): **1)** linear noise schedule with $S = P \cdot T$ for integer $P$, where each step reveals exactly $P$ tokens; **2)** confidence-based adaptive decoding (Chang et al., 2022) rather than random token selection; **3)** encoder-only Transformer architecture without timestep embedding; **4)** conditional generation where input prompt $\mathbf{x}$ of length $n$ is a prefix of $\mathbf{x}_0$, with $n$ calculated within context length $S$, aligned with reasoning problem setup for ARM. Detailed MDM introduction and encoder-only Transformer definition are in § B and § F, respectively.

## 3 A THEORY OF MASKED DIFFUSION

The generation process in MDM is unique in two different ways: it generates multiple tokens in parallel and permits any-order generation. We now investigate whether and how exactly these properties, in their own right, translate into concrete advantages.

### 3.1 POWER OF PARALLELISM

Prior work (Merrill & Sabharwal, 2024; Feng et al., 2024; Li et al., 2024) has shown that ARM with sufficiently many intermediate steps is Turing-complete and thus can solve any computable problem. Analogous to the role of intermediate steps in ARM, two governing resources determine MDM's power: **1)** number of decoding (denoising) steps $T(n)$, and **2)** maximum context length $S(n)$ (equivalently, the maximum number of tokens available to decode).

**Definition 1** (MDM). *Let $\mathsf{MDM}(S(n), T(n))$ be the class of decision problems solvable by MDM (§ 2) with maximum context length $S(n)$ and at most $T(n)$ decoding steps, using some constant depth and $\log(n)$ embedding size encoder-only Transformer. Also, let $\mathsf{MDM}(S(n)) = \bigcup_{T(n)} \mathsf{MDM}(S(n), T(n))$.*

To formally characterize MDM's expressivity in relation to $T(n)$ and $S(n)$, we establish a connection with the canonical parallel computation model called *Parallel Random Access Machine (PRAM)* (Fortune & Wyllie, 1978; JáJá, 1992), which is the RAM model extended to multiple processors executing over shared memory. See detailed introduction and a formal definition of the variant we use in § E.

**Definition 2** (PRAM). *Let $P(n)$ be the number of processors budget, and $w(n) = \Theta(\log n)$ the word size. Define $\mathsf{PRAM}(P(n), T(n))$ as the class of decision problems solvable by a* uniform *CREW PRAM (see § E for CREW specification) using at most $P(n)$ processors in at most $T(n)$ parallel time.*

**Theorem 1** (MDM Simulation of PRAM, Informal). *For any PRAM program that runs on input $\mathbf{x} \in \Sigma^n$ in at most $T(n)$ parallel time with $P(n)$ maximum processors, there exists an MDM on input $\mathbf{x}$, padded to $S(n) = \mathcal{O}(P(n) \cdot T(n))$, that matches the PRAM output in $\mathcal{O}(T(n))$ decoding steps, i.e. $\mathsf{PRAM}(P(n), T(n)) \subseteq \mathsf{MDM}(\mathcal{O}(P(n) \cdot T(n)), \mathcal{O}(T(n)))$. See formal statement in Theorem 8.*

**This demonstrates that MDM can simulate any PRAM algorithm with optimal parallel time complexity, thereby it is not only Turing-complete as ARM already achieves, but can also solve**

---

[1] Unlike convention in diffusion model where larger $t$ denotes earlier inference steps, we use $t$ following an intuitive feed-forward ordering during inference, i.e. the focus of this paper.

**problems significantly faster with parallelization, something ARM cannot offer.** The speedup can be *exponential* compared to ARM's serial time complexity: for efficiently parallelizable problems in NC (Arora & Barak, 2009),[2] graph connectivity can be solved in $\mathcal{O}(\log n)$ decoding steps versus ARM's linear complexity, and context-free languages including Dyck-k require only $\mathcal{O}(\log^2 n)$ steps. These tasks have been demonstrated hard or inefficient for ARM in previous literature (Strobl et al., 2024; Zhu et al., 2025).

**Remark** While MDMs can achieve optimal decoding steps (and thus wall-clock time) through parallel decoding, encoder-based architectures require re-encoding the entire sequence at each step and cannot utilize KV caching, resulting in higher per-step FLOPs compared to decoder-based ARM. In theoretical construction for Theorem 1, we do not find such trade-off unavoidable, suggesting potential to improve per-step FLOPs while maintaining the parallel time advantage for current MDMs.

## 3.2 (UN)SCALABILITY TO HARD TASKS

While noteworthy, the computational power described above comes with a non-negligible cost: solving a problem requires context length $S(n)$ to scale as $\mathcal{O}(T(n) \cdot P(n))$ (the total parallel work), a quantity at least as large as the serial time complexity (with $P(n) = 1$), per Brent's Theorem (JáJá, 1992). Particularly, in resource-constrained regimes, we have:

**Theorem 2.** $MDM(S(n)) \subseteq PRAM(1, \tilde{\mathcal{O}}(S^3(n)))$, *where logarithmic factors are hidden in* $\tilde{\mathcal{O}}$.

In other words, MDM with context length $S(n)$ cannot solve problems requiring more than $\tilde{\mathcal{O}}(S^3(n))$ serial time. This limitation is also shared by ARM (Yang et al., 2025).

**This implies MDM is inherently not scalable to solving hard reasoning or generation tasks**: for problems beyond P (e.g., NP-hard problems), this would require superpolynomial context length (under standard complexity assumptions), practically intractable in terms of both memory and per-step FLOPs. The root cause lies in MDM's irreversible token generation: once decoded, those positions cannot be reused or rewritten. As reflected in the construction of Theorem 1, each memory write must be permanently stored as tokens, forcing space to scale with computation time.

In contrast, human reasoning on hard problems naturally involves continuous revision, exploration of alternative paths, and correction of mistakes before reaching final conclusions. Generation tasks are no different: for instance, generating planar graphs (drawable on planes without edge crossings) with minimum splitting numbers is NP-complete and naturally involves iteratively adding nodes, checking planarity constraints, and backtracking when violations occur. Such a process has not been captured by either ARM or MDM.

## 3.3 (LIMITED) POWER OF ANY-ORDER GENERATION

Any-order generation seems to offer extra flexibility over auto-regression, but does it truly translate into computational advantages? To attribute gains to any-order generation itself, we control for orthogonal factors differentiating ARM and MDM: **1)** the number of tokens generated per step, and **2)** the backbone architecture (decoder v.s. encoder). The former has already been shown to confer stronger parallelism to MDM (§ 3.1); the latter provides internal parallelization benefits (Ewer et al., 2024) and improved expressiveness through padding with dummy tokens (i.e. M) (Merrill & Sabharwal, 2025).

Therefore, we fix MDM to emit exactly one token per step and ARM to use the encoder-backbone with mask tokens padding the sequence to the same length (called *Masked-ARM*), or equivalently:

**Definition 3** (Masked-ARM). *A Masked-ARM is defined as an MDM (§ 2) that is forced to decode in left-to-right order and one token per step.*

**Perhaps counterintuitively, we show that the computational benefits from any-order generation are rather limited: any-order generation itself does not expand what ARM can already solve**

**Theorem 3** (Left-to-Right v.s. Any-Order, Informal). *For any AO-MDM with context length $S(n)$ decoding one token per step, there exists a Masked-ARM with length $\mathcal{O}(S(n))$ and extra constant*

---

[2]NC is the complexity class for efficiently parallelizable problems, those that are solvable in polylog($n$) time using poly($n$) processors; NC $\subseteq$ P and it is open whether NC $=$ P (Greenlaw et al., 1995). PRAM is the canonical model for this notion as a Turing machine is for P.

*layers, that can produce the same generation process for any given input* **x***, by explicitly specifying both where to write (position) and what to write (token). See formal statement in Theorem* 9.

Simulating any-order generation with autoregressive models is not hard because the attention mechanism is good at fetching information from any position, and re-organizing it internally in the correct order to perform the same computation. While Masked-ARM need not replicate MDM's exact final sequence, an additional post-processing step can align their outputs without affecting the theoretical conclusion. There are also some empirical evidence showing the effectiveness of ARM simulating any-order (Xue et al., 2025).

But not all intricacies inherent in natural generation processes can be easily sequentialized. Coding for example (as well as many natural scientific processes alike), involves anywhere editing where a new valid state depends upon previous valid states that may not be contained in the final sequence. And even when described in left-to-right temporal order, reproducing the state requires more than simple re-organization, which attention is already provably good at. Hence such a complex process is not captured by ARM or MDM as currently instantiated.

**Remark** We note that MDM's observed advantages in practice may lie in discovering an optimal order (Kim et al., 2025c) from data, where left-to-right ordering need not exactly correspond to the optimal temporal generation order, though computationally equivalent (Theorem 3).

## 4 ANY-PROCESS GENERATION

We now introduce *Any-Process Generation*, a more powerful generation paradigm that extends the any-order masked diffusion from § 2 (referred to as AO-MDM hereafter) by removing various restrictions to capture natural processes not present in existing generation strategies.

**A General Formulation** Let $f_\theta : \bar{\Sigma}^* \to \bar{\Sigma}^* \times \Sigma^* \times \mathcal{C}^*$ be a function, by default parameterized by the same Transformer architecture, which on input $\mathbf{x}_k \in \bar{\Sigma}^*$ returns the triple $(\mathbf{x}_k, \mathbf{y}_k, \mathbf{c}_k)$ with $\mathbf{y}_k \in \Sigma^{|\mathbf{x}_k|}$ and $\mathbf{c}_k \in \mathcal{C}^{|\mathbf{x}_k|}$, where $\mathbf{x}_k$ may contain one or more $\mathsf{M}$ tokens (a masked sequence) while $\mathbf{y}_k$ is mask-free; $\mathbf{y}_k$ and $\mathbf{c}_k$ have the same length as the input. Core to the design of this generation process is a parameter-free (and optionally non-deterministic) transition function $g : \bar{\Sigma}^* \times \Sigma^* \times \mathcal{C}^* \to \bar{\Sigma}^*$, which takes $(\mathbf{x}_k, \mathbf{y}_k, \mathbf{c}_k)$ as input to produce the next sequence $\mathbf{x}_{k+1} \in \bar{\Sigma}^*$ that can differ in length from $\mathbf{x}_k$. The inclusion of input $\mathbf{x}_k$ itself in the output of $f_\theta(\mathbf{x}_k)$ ensures that $g$ has access to which positions are masks initially. Overall, generation is formulated as the iterative application of $f_\theta$ and $g$ until some stopping criterion is met:

$$\mathbf{x}_{t+1} = g\big(f_\theta(\mathbf{x}_t)\big), \quad \text{and hence} \quad \mathbf{x}_t = (g \circ f_\theta)^t(\mathbf{x}_0) = g \circ f_\theta \circ g \circ f_\theta \circ \cdots \circ g \circ f_\theta(\mathbf{x}_0). \quad (1)$$

Notably, unlike vanilla MDM, this framework imposes no restrictions on mask ratios at any given time step, therefore each decoding step can unmask an arbitrary number of tokens, and $\mathbf{x}_0$ can be the input prompt $\mathbf{x}$ directly with no initial mask, as in ARM. This framework also does not limit the maximum number of decoding steps $T$, the stopping criterion is flexible and need not be a fully unmasked sequence, as in ARM. We dub this class of models as *Any-Process MDM* (AP-MDM) and will detail a specific instantiation. It is not difficult to see AO-MDM is a special case of AP-MDM.

### 4.1 AN INSTANTIATION WITH UNMASK, REMASK, INSERT, AND DELETE OPERATIONS

Define $\mathcal{C} = \{0, 1\}^3$. For each position $i$ and time step $t$, the per-position control is a 3-bit vector $c_{t,i} = (c_{t,i}[1], c_{t,i}[2], c_{t,i}[3]) \in \mathcal{C}$ reserved for different purposes that will be detailed below. Correspondingly, we write $\mathbf{c}_t = (c_{t,1}, \ldots, c_{t,|\mathbf{x}_t|}) \in \mathcal{C}^{|\mathbf{x}_t|}$.

**Capability: Rewrite via Remask** We use the first bit of the per-position control $c_{t,i}[1] \in \{0, 1\}$ to control remasking (and whether to unmask) and define $\forall y \in \Sigma$:

$$\mathbf{remask}_{x_{t,i}, c_{t,i}}(y) = \begin{cases} \mathsf{M} & \text{if } c_{t,i}[1] = 1 \\ y & \text{if } x_{t,i} = \mathsf{M} \text{ and } c_{t,i}[1] = 0 \\ x_{t,i} & \text{otherwise} \end{cases} \quad (2)$$

In other words, when $c_{t,i}[1] = 1$, position $i$ is a mask after decoding regardless of its previous state; otherwise, standard unmasking follows as usual. This operation enables self-correction and *test-time scaling*, allowing models to scale computation exponentially in $S(n)$ before state repetition occurs.

Additionally, since the remasking signal can be learned from data, models can adaptively determine both decoding order and parallelization degree at each step.

**Capability: Length-Variable Edit via Insert / Delete** We use the second and third bits of the per-position control $c_{t,i}$ to govern insertion and deletion, respectively. Define $\forall y \in \bar{\Sigma} \cup \{\epsilon\}$:

$$\textbf{insert}_{c_{t,i}}(y) = \begin{cases} (y, \mathsf{M}) & \text{if } c_{t,i}[2] = 1 \\ y & \text{otherwise} \end{cases}, \quad \textbf{delete}_{x_{t,i}, c_{t,i}}(y) = \begin{cases} \epsilon & \text{if } x_{t,i} = \mathsf{M} \text{ and } c_{t,i}[3] = 1 \\ y & \text{otherwise} \end{cases} \quad (3)$$

where $\epsilon$ denotes the empty string. In other words, insert adds a mask token after position $i$ when $c_{t,i}[2] = 1$, and delete removes position $i$ when it was originally a mask token ($x_{t,i} = \mathsf{M}$) and $c_{t,i}[3] = 1$. These operations enable dynamic sequence length adjustment based on problem complexity, with the insert operation allowing sequence length to grow exponentially as *each mask token can spawn additional masks*. Furthermore, the delete operation provides computational efficiency by freeing space during stages that require less extensive computation, reducing overall FLOPs waste. This mechanism can work orthogonally with semi-autoregression (Arriola et al., 2025) that expands space at the end of the sequence.

**Composition** To summarize, each decoding step applies the three operations coordinate-wise and concatenates the resulting segments:

$$f_\theta(\mathbf{x}_t) = (\mathbf{x}_t, \mathbf{y}_t, \mathbf{c}_t), \quad g(\mathbf{x}_t, \mathbf{y}_t, \mathbf{c}_t) = (s_{t,1}, s_{t,2}, \ldots, s_{t,|\mathbf{x}_t|}) = \mathbf{x}_{t+1}$$
$$\text{where } s_{t,i} = \textbf{insert}_{c_{t,i}} \circ \textbf{delete}_{x_{t,i}, c_{t,i}} \circ \textbf{remask}_{x_{t,i}, c_{t,i}}(y_{t,i}), \quad \forall i \in [|\mathbf{x}_t|] \quad (4)$$

An algorithmic description of any-process generation with these operations is given in Algorithm 1, § C.4. And the stopping criterion can be flexibly defined, e.g. one can use generation of an EOS token (as in ARM) or convergence to a repeated sequence (loop occurs). Architecturally, implementing these capabilities requires no changes to the Transformer structure, only three additional logit dimensions are needed for producing control signals, i.e. three extra linear heads (details in § C.1).

**Pre-Training / Fine-Tuning / Data Availability** All three operations can be trained end-to-end from text corpora using self-supervised objectives, preserving MDM's scalability to large-scale training (details in § C.2). The minimal architectural changes also enable direct fine-tuning from existing large diffusion models, possibly using supervised data where the underlying generation process is explicitly constructed (details in § C.3). This training flexibility separates our approach from alternatives like looped Transformers (Dehghani et al., 2018; Giannou et al., 2023), which are expressive but notoriously difficult to train due to lack of intermediate supervision.

**Design Considerations** The proposed three operations all revolve around the mask token $\mathsf{M}$, leveraging existing MDM's strong unmasking capability while only adding modular extensions that are easier to pretrain or fine-tune than learning harder operations such as inversion of uniform noise (Sahoo et al., 2025). Moreover, while the definitions of $g$ and $\mathbf{c}_t$ in (4) suffice for achieving theoretical benefits detailed later, they are not necessary conditions; other designs are possible.

**Positional Encoding** When insertion or deletion operations modify sequence length, we update the positional encodings according to the new token positions. This position re-encoding mechanism is essential for AP-MDM's expressivity advantages (§ 5) as we will show, allowing the model to represent more complex processes where position shifts carry meaningful structural information.

We note that the ideas of remasking and editing have been individually explored in some prior/concurrent works (Wang et al., 2025; Peng et al., 2025; von Rütte et al., 2025; Havasi et al., 2025; Wu et al., 2025). However, there has not been a systematic study of guidance principles for yielding provable computational benefits and their practical implications. See discussions in § A.

## 5 THE POWER OF ANY-PROCESS GENERATION

We now show how any-process generation circumvents various difficulties that current ARM and MDM encounter when handling tasks across different domains and complexities.

### 5.1 UNIVERSALLY EFFICIENT COMPUTATION

> **Benefit 1**: *Scalability to significantly harder problems through rewriting and backtracking.*

| Method | # Samples | # Param | Accuracy |
|---|---|---|---|
| ARM (w/o ordering) | 1.8M | 42M | 9.73% |
| ARM (with ordering) | 1.8M | 42M | 87.18% |
| AO-MDM (vanilla) | 1.8M | 6M | 6.88% |
| AO-MDM (Top-K probability) | 1.8M | 6M | 18.51% |
| AO-MDM (Top-K prob. margin) | 1.8M | 6M | 89.49% |
| AP-MDM | 100 | 1.2M | 99.28% |

**(a)** Comparison of accuracy on Sudoku.

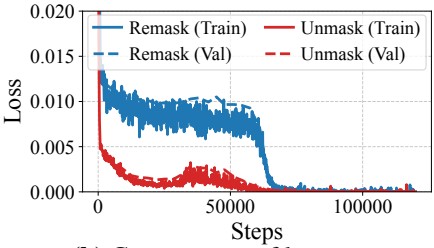

**(b)** Convergence of losses.

Figure 3: Experimental results on Sudoku puzzles. Results of ARM and AO-MDM are taken from Kim et al. (2025c). Losses are defined in § C.2.

As discussed in § 3.2, inherently hard problems (e.g. many NP-hard tasks) typically do not admit sequential processes but require iterative "search–verify–revise" loops, which hold across various domains: from theorem proving, solving code challenges to synthesis of complex structures in nature. The pathological way current generation paradigms let discardable tokens accumulate indefinitely creates scaling barriers, where space explodes and each step incurs ever-increasing computational cost (Theorem 2). We now demonstrate how AP-MDM resolves this:

**Theorem 4** (AP-MDM Simulation of PRAM, Informal). *For any PRAM program that runs in at most $T(n)$ parallel time, $P(n)$ processors and $S(n)$ memory usage, there exists an AP-MDM that matches PRAM output on any given input with $\mathcal{O}(S(n))$ context length and $\mathcal{O}(T(n))$ decoding steps. See formal statement in Theorem 10.*

By comparison, standard MDM requires space scaling with the total work $\mathcal{O}(P(n) \cdot T(n))$ (Theorem 1), whereas AP-MDM requires only the actual space needed, achieving both optimal parallel time and space complexity (Theorem 4). This implies AP-MDM not only retains the efficiency of parallelization, but also dramatically expands the range of solvable problems. In particular, given polynomial context length, AP-MDM can solve problems in PSPACE rather than just P, which is an exponential improvement that makes many NP-hard problems solvable with test-time scaling.

**Experiment: Sudoku Puzzles**  We conduct experiments on Sudoku puzzles, i.e. an NP-complete problem when generalized to $n^2 \times n^2$ grids, requiring both the capability of any-order generation, and the capability to rewrite. As illustrated in Figure 2(a), AP-MDM can use the **remask** (and standard **unmask**) operations to choose the easiest position to fill first, and also erase failed branches and try alternative assignments, effectively scaling compute to solve harder instances.

We follow the experimental setup from (Kim et al., 2025c) but with a key difference: while the original work used 1.8M training puzzles, we use only 100 (moderately hard) instances for training AP-MDM. Despite this significantly reduced dataset, our approach achieves near-perfect accuracy (99.28%) on most Sudoku puzzles, outperforming both AO-MDM and ARM that use substantially more samples and larger model sizes, as shown in Figure 3. Any-process generation is more sample-efficient because if the model is allowed to conduct more computes and more steps to solve the same problem, each step would become easier to learn. Orthogonally, we find from the training dynamics in Figure 3 that the model quickly learns to identify and fill the easiest positions (unmasking loss drops rapidly), while learning the order (which position to fill next) proves more challenging.

## 5.2 STRUCTURAL GENERATION: EXAMPLES IN CODING AND SCIENCE

> **Benefit 2**: *Generating (or reasoning over) complex structured objects that evolve non-sequentially, common across domains beyond natural language (e.g. coding, science).*

When the evolving object involves some complex structures (e.g. trees, graphs, strings with constraints) that do not inherently build up linearly, forcing the generation into a sequential procedure can introduce unnecessary computational difficulties. Such scenarios are especially common across domains beyond natural language (e.g. coding, biology), which current LLMs struggle with.

**Example 1: Coding**  Programs generally require satisfying global constraints like syntax and semantics at every intermediate state during development, since building each state upon the previous valid one is easier than jumping directly to the final solution. To illustrate this, consider the basic task of generating matched parentheses (the Dyck-$k$ language with $k$ types of parentheses), as illustrated

in Figure 2(b). Natural generation involves inserting parentheses anywhere without breaking balance constraints, while left-to-right generation requires global foresight and constant validity checking during generation, provably impossible for Transformers at scale. Particularly, we consider a variant called the two-sided Dyck-k (Definition in § L):

**Theorem 5** (Generating Two-Sided Dyck-$k$, Informal). *For any $k \geq 2$, there exists a stochastic AP-MDM whose generation distribution has support exactly equal to the two-sided Dyck-k language, i.e., a string has positive generation probability if and only if it is in the language. Conversely, for any fixed ARM, there exists a length threshold beyond which the ARM cannot guarantee positive probability for all strings in two-sided Dyck-k. See formal statement in Theorem 12.*

This result holds because generating Dyck language for arbitrary length is as hard as recognizing it, i.e., deciding if the current sequence has matched parentheses, which Transformer restricted in $\mathsf{TC}^0$ expressivity cannot do. Intuitively, both ARM and vanilla MDM require global foresight capabilities that are computationally hard for constant-depth Transformers: ARM must foresee the matching closing brackets when generating opening ones (requiring global planning of the nested structure), while vanilla MDM must predetermine the number of tokens between each matched pair at initialization and commit to fixed positions that cannot be modified later. In contrast, AP-MDM fundamentally circumvents these difficulties through insertion operations, allowing the model to iteratively build the structure through local decisions that are simple to represent.

**Example 2: Science / Biology** Consider linear splicing (Head, 1987; Păun, 1996), which is DNA recombination abstracted (and perhaps over-simplified) as cutting two strings and cross-pasting their halves, as illustrated in Figure 2(d). Iterating such rules from a finite seed set generates a splicing language, and any regular language with a marker added to the left side can be generated by such a system (Head, 1998; Kari & Kopecki, 2012), while regular language has been proven impossible for constant-depth Transformers (Liu et al., 2022; Li et al., 2024).

To empirically verify AP-MDM's advantage on tasks concerning non-sequential objects, we consider a graph generation task which involves structural editing.

**Experiment: Graph Generation** Given a directed graph and a prompt specifying source and target nodes $s$ and $t$, the model is required to generate a modified graph that disconnects $s$ and $t$ by removing the minimum number of edges. This is equivalent to finding the min-cut. Efficient algorithms for generation typically involve iterative editing: **1)** Use BFS to find a path from $s$ to $t$; **2)** Augment this path and modify the graph structure; **3)** Repeat until $s$ and $t$ are disconnected, then remove the min-cut edges. Any-process generation is naturally suited for such graph editing tasks, leveraging insert/**delete**/remask operations for adaptive structural and feature modifications and MDM's parallel capabilities, as illustrated in Figure 2(e). As shown in Figure 4, our model achieves almost perfect accuracy for increasingly larger graphs. Meanwhile, when we train ARM to simulate the same process, it fails to perform well as graph size increases.

## 5.3 LEARNING AND OOD GENERALIZATION

> **Benefit 3**: *Enabling the use of simpler algorithms to solve problems, thereby improving sample efficiency and (out-of-distribution) generalization.*

**Experiment: Parity** Given a binary sequence $\mathbf{x} \in \{0, 1\}^n$, parity involves determining if there are an even or odd number of 1s. This task is conceptually trivial but embarrassingly difficult for LLMs. Intuitively, the difficulty arises because ARM is forced to attend the entire input and learn a position-invariant target function, which is hard on training sequences with finite-length. With any-process generation, the model circumvents this difficulty by learning a simple elimination algorithm: examine the first two tokens, **delete** all 0s if the pair contains any 0 or **delete** the pair if both are 1s, then repeat until all are processed (Figure 2(c)). The answer is true if any 1s remain. This mimics how humans solve the problem, a simple length-generalizable approach only possible with deletion.

As shown in Figure 4, our model achieves 100% accuracy on **any length** after training on only $n = 2$ length sequences with a tiny model ($\sim$200 parameters), while ARM with orders of magnitude more parameters and samples fails to generalize beyond training lengths.

| Graph Size (# Nodes) | ARM Acc. | AP-MDM Acc. |
|:---:|:---:|:---:|
| 4 | 90.32% | 100% |
| 5 | 43.04% | 100% |
| 6 | 0.30% | 100% |
| 7 | N/A | 100% |
| 8 | N/A | 99.99% |
| 9 | N/A | 99.97% |
| 10 | N/A | 99.92% |

**(a)** Graph generation via editing.

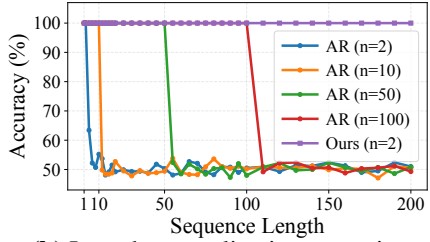

**(b)** Length generalization on parity.

Figure 4: Graph generation and parity task results.

## 5.4 HARDNESS OF BEING SIMULATED

> ***Benefit 4****: If in the future we have access to data of underlying generation processes (e.g. revision history of code, articles, math proof drafts, molecular formation processes), any-process MDM is more suitable than ARM for practical training.*

Besides scalability to harder tasks (§ 5.1) and universality across domains (§ 5.2), a crucial question remains: suppose given access to datasets containing revision histories of the objects we wish to generate, would AP-MDM be the most appropriate model for such data and large-scale training? To answer this, we consider ARM as a competitor as it is Turing-complete, and equally expressive as AO-MDM (Theorem 3) when controlled for orthogonal factors.

We next show ARM is inherently unsuitable for training on editing datasets in two ways. **Firstly**, unlike any-order generation (Theorem 3), AP-MDM's editing operations are hard to be simulated by ARM by explicitly specifying editing operations applied at each decoding step; particularly

**Theorem 6** (Hardness of Simulating AP-MDM, Informal)**.** *There exists a constant-depth AP-MDM, such that no constant-depth ARM can simulate the generation process of that AP-MDM using a sequence of triplets, i.e., what operation to use (*unmask*, *remask*, *insert*, *delete*)*, where to apply the operation (position) and what to write for the unmask operation (token), on any input* **x***, under some complexity hardness assumptions in* § M*. See formal statement in Theorem 13.*

The key difficulty of simulating AP-MDM lies in the additional complexity brought by the insertion and deletion operations that trigger position shifts: ARM must internally reconstruct the entire editing trajectory to determine what token currently occupies each position (formalized via the PRESERVE problem in § M), which is computationally hard for constant-depth Transformers.

Empirically, we show that representing our generation process using triplets described above for ARM simulation indeed becomes increasingly difficult to train as sequence length grows, as demonstrated in the graph generation task in § 5.2.

**Secondly**, if we disregard the resource constraints from § 5.1 and § 5.2, simulation becomes possible through additional intermediate steps, but this could require highly contrived trajectories that defeat the purpose of practical training, e.g. periodically summarize the current state, or using more than constant tokens to represent each application of an operation.

## 6 CONCLUSION

This paper provides formal analysis of generation processes and shows, provably and empirically, that moving beyond standard autoregression and current masked diffusion yields more powerful models. These results suggest concrete design principles for frontier LLMs, pointing to training and decoding schemes that scale to increasingly hard tasks and generalize across domains such as code and the sciences. See further contextualization with respect to related work in § A.

## USAGE OF LARGE LANGUAGE MODELS

In this work, we use LLMs for literature retrieval and discovery, writing assistance and polishing, and code writing and debugging support. We carefully monitor potential issues such as plagiarism or factual inaccuracies to ensure academic integrity.

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

## Contents

## A  RELATED WORK

Masked diffusion models (Hoogeboom et al., 2021; Austin et al., 2021; Lou et al., 2024; Sahoo et al., 2024; Shi et al., 2024) extend continuous diffusion models (Sohl-Dickstein et al., 2015; Ho et al., 2020; Song et al., 2020) to discrete data. Early work applied these models to specialized domains such as graph generation (Vignac et al., 2022; Sun & Yang, 2023), protein design (Gruver et al., 2023), and drug discovery (Lee et al., 2025), where non-sequential generation provides natural advantages. The field has evolved with recent commercial-scale language models like Gemini Diffusion (DeepMind, 2025) and Mercury (Labs et al., 2025), which demonstrate competitive performance on language generation, reasoning, and coding tasks. This suggests that MDMs can serve as viable alternatives to the autoregressive models that currently dominate LLMs. Against this background, this paper investigates the fundamental computational differences between generation paradigms and explores whether more powerful generation methods exist.

Several works have explored extensions to standard MDM through mechanisms that enable rewriting and editing (von Rütte et al., 2025; Wang et al., 2025; Peng et al., 2025; Havasi et al., 2025; Wu et al., 2025; Kim et al., 2025a), which relate to our any-process generation framework. Wang et al. (2025) introduces random remasking during inference, though this capability is not learned from data. Lou et al. (2024); von Rütte et al. (2025); Sahoo et al. (2025) propose adding uniform noise in the forward process rather than using masks, with models learning to revert them in the backward process, but this approach generally underperforms since modifying tokens directly appears more difficult than unmasking. Peng et al. (2025) introduces path planning to control generation, though the planner is not trained end-to-end with the base model. Concurrent with ours: Havasi et al. (2025) introduces edit operations to flow matching frameworks but faces similar limitations as uniform noise approaches; Kim et al. (2025a) and Kim et al. (2025b) introduce to insert tokens at any position and remask tokens, while Wu et al. (2025) proposes expansion and delete, but these capabilities per se are insufficient for handling hard reasoning tasks as discussed in § 3.

Before MDM, there are also some earlier explorations of non-autoregressive generation processes. Insertion-based models such as the Insertion Transformer (Stern et al., 2019), InDIGO (Gu et al., 2019a), non-monotonic sequential text generation (Welleck et al., 2019), the Levenshtein Transformer (Gu et al., 2019b), and InsNet (Lu et al., 2022) generate text by repeatedly inserting (and sometimes deleting) tokens at arbitrary positions, while XLNet (Yang et al., 2019) generalizes autoregressive pretraining to random permutations of the factorization order. LaserTagger (Malmi et al., 2019) and Mask-Predict (Ghazvininejad et al., 2019) formulate generation as iteratively predicting edit tags or filling masks in parallel. These methods demonstrate that alternative generation orders and local edits can bring empirical benefits on specific tasks, but most of them are defined based on particular architectures and decoding heuristics. In contrast, this work provides a unified formalization that subsumes insertion, deletion, and (re)masking operations as special cases, and the first systematic analysis of their computational power with comparison to ARM and MDM.

## B  BACKGROUND: MASKED DIFFUSION MODEL

We introduce the preliminaries of diffusion language models or masked diffusion models (MDM), following the notation established in § 2. Let $\bar{\Sigma} = \Sigma \cup \{M\}$ be the extended vocabulary where $M$ is the mask special token. Consider sequences $\mathbf{x}_t = (x_{t,1}, x_{t,2}, \ldots, x_{t,S}) \in \bar{\Sigma}^S$ indexed by time $t \in [T]$, where $S$ is the maximum context length, $T$ is the number of decoding steps, $\mathbf{x}_0 = \{M\}^S$ is the fully masked sequence and $\mathbf{x}_T \in \Sigma^S$ is the target clean sequence.

**Forward Noising Process**  The forward noising process constructs training data by generating noisy versions $\mathbf{x}_t$ from clean sequences $\mathbf{x}_T$. Unlike the discrete inference steps in § 2, training uses continuous time $t \in [0, T]$ with larger $t$ denoting later denoising steps. MDM employs the *absorbing mask kernel* (Austin et al., 2021) where the signal ratio $\alpha_t = t/T$ represents the marginal probability that a token remains unmasked. Since $\alpha_t$ increases monotonically with $t$ (i.e., $\alpha_s < \alpha_t$ for $s < t$), later timesteps preserve more original tokens, consistent with our convention where $t = 0$ is fully masked and $t = T$ is clean. At each position $i$, tokens either stay unchanged or become $M$, and once masked, they "absorb" into this state. The marginal distribution is:

$$q(x_{t,i} \mid x_{T,i}) = \text{Cat}(x_{t,i}; \alpha_t \mathbf{e}_{x_{T,i}} + (1 - \alpha_t)\mathbf{e}_M) \tag{5}$$

where $\mathbf{e}_v$ denotes the one-hot vector for token $v \in \bar{\Sigma}$. To obtain noised sequence $\mathbf{x}_t$ from $\mathbf{x}_T$, we compute the masking probability $1 - \alpha_t$ and mask each position independently with this probability.

**Training Objective**  The reverse process aims to recover $\mathbf{x}_T$ from $\mathbf{x}_0$. MDM parameterizes $p_\theta(\mathbf{x}_T \mid \mathbf{x}_t, t)$ to predict the clean data directly (but as mentioned in § 2, many recent large-scale MDMs omit explicit timestep embeddings). For the absorbing mask kernel, the true posterior $q(\mathbf{x}_s \mid \mathbf{x}_t, \mathbf{x}_T)$ for $s < t$ has an analytical form. For each position $i$:

$$q(x_{s,i} \mid x_{t,i}, x_{T,i}) = \begin{cases} \text{Cat}(x_{s,i}; \mathbf{e}_{x_{t,i}}), & \text{if } x_{t,i} \neq \mathsf{M} \\ \text{Cat}\left(x_{s,i}; \frac{(1-\alpha_s)\mathbf{e}_\mathsf{M}+(\alpha_t-\alpha_s)\mathbf{e}_{x_{T,i}}}{1-\alpha_s}\right), & \text{if } x_{t,i} = \mathsf{M} \end{cases} \tag{6}$$

This means that, if position $i$ is not masked at time $t$, it remains unchanged at time $s$; if position $i$ is masked at time $t$, it transitions probabilistically between the original token and mask. The training objective is derived from the variational lower bound. For the absorbing mask kernel, it simplifies to:

$$\mathcal{L}_{\text{CE}}(\theta) = \mathbb{E}_{t\sim\mathcal{U}(0,T),\mathbf{x}_T\sim p_{\text{data}},\mathbf{x}_t\sim q(\mathbf{x}_t|\mathbf{x}_T)} \left[ -\frac{1}{|\{i : x_{t,i} = \mathsf{M}\}|} \sum_{i:x_{t,i}=\mathsf{M}} \log p_\theta(x_{T,i} \mid \mathbf{x}_t, t) \right] \tag{7}$$

The loss is computed only on masked positions and averaged over the number of masked tokens, making this equivalent to conditional masked language modeling with proper normalization. Here, $t$ is sampled uniformly from the continuous interval $[0, T]$ during training, $\mathbf{x}_T$ is sampled from the data distribution, and $\mathbf{x}_t$ is obtained by applying the forward noising process.

## C  METHODOLOGICAL DETAILS

### C.1  MODEL ARCHITECTURE

As described in § 4, AP-MDM extends standard MDM with three additional capabilities: **remask** (rewrite via remasking), **insert** (insert masks), and **delete** (remove redundant masks). In principle, these capabilities can be implemented using a shared encoder-only Transformer backbone with three additional linear heads, adding minimal computational overhead.

**Architecture**  Following Equation (4), AP-MDM uses four prediction heads on top of a shared encoder-only Transformer backbone. Given the hidden representation $\mathbf{h}_{t,i} \in \mathbb{R}^d$ for position $i$, the heads output logits and probabilities:

$$p_\theta(y_{t,i} \mid \mathbf{x}_t) = \text{softmax}(\mathbf{W}_U\mathbf{h}_{t,i} + \mathbf{b}_U), \quad \mathbf{W}_U \in \mathbb{R}^{|\Sigma|\times d} \quad \text{(unmask)} \tag{8}$$

$$c_{t,i}[1] = \sigma(\mathbf{W}_R\mathbf{h}_{t,i} + b_R), \quad \mathbf{W}_R \in \mathbb{R}^{d\times 1} \quad \text{(remask)} \tag{9}$$

$$c_{t,i}[2] = \sigma(\mathbf{W}_I\mathbf{h}_{t,i} + b_I), \quad \mathbf{W}_I \in \mathbb{R}^{d\times 1} \quad \text{(insert)} \tag{10}$$

$$c_{t,i}[3] = \sigma(\mathbf{W}_D\mathbf{h}_{t,i} + b_D), \quad \mathbf{W}_D \in \mathbb{R}^{d\times 1} \quad \text{(delete)} \tag{11}$$

where the unmask head outputs a probability distribution $p_\theta(y_{t,i} \mid \mathbf{x}_t)$ over the vocabulary $\Sigma$ for predicting tokens at masked positions, while the three control signal heads output binary probabilities for the corresponding operations. During inference, tokens are obtained via $y_{t,i} = \arg\max p_\theta(\cdot \mid \mathbf{x}_t)$ and control signals are obtained by thresholding.

### C.2  SELF-SUPERVISED TRAINING

In principle, one can design specialized loss functions corresponding to each operation, alongside the standard unmasking loss from MDM, by constructing self-supervised signals from the inherent structure of text data through augmentation strategies.

**Unmasking Loss**  The unmasking loss would follow standard MDM training as described in Appendix B. For each training sample $\mathbf{x}_T$, one can sample $t \sim \mathcal{U}(0, T)$ and apply the forward masking process with signal ratio $\alpha_t = t/T$ to create $\mathbf{x}_t$:

$$x_{t,i} = \begin{cases} x_{T,i} & \text{with probability } \alpha_t \\ \mathsf{M} & \text{with probability } 1 - \alpha_t \end{cases} \tag{12}$$

The model would learn to predict original tokens at masked positions with time weighting:

$$\mathcal{L}_{\text{unmask}} = \mathbb{E}_{t,\mathbf{x}_T,\mathbf{x}_t} \left[ \frac{1}{\sum_i m_i} \sum_{i=1}^{|\mathbf{x}_T|} m_i \cdot \left( -\frac{\log p_\theta(x_{T,i} \mid \mathbf{x}_t)}{t} \right) \right] \tag{13}$$

where $m_i = 1$ if position $i$ is valid (according to attention mask), 0 otherwise.

**Remasking Loss**  The remasking loss could train the model to identify incorrect tokens that should be remasked. For each sample $\mathbf{x}_T$, one can sample $t \sim \mathcal{U}(0,T)$ and create a corrupted sequence $\tilde{\mathbf{x}}_t$ using batch-internal shuffling (which effectively samples from the empirical token distribution rather than a biased uniform distribution):

$$\tilde{x}_{t,i} = \begin{cases} x_{T,i} & \text{with probability } \alpha_t \\ \text{shuffled token} & \text{with probability } 1 - \alpha_t \end{cases} \tag{14}$$

The remasking labels are $c_{t,i}[1] = \mathbf{1}[x_{T,i} \neq \tilde{x}_{t,i}]$, and the loss uses binary cross-entropy:

$$\mathcal{L}_{\text{remask}} = \mathbb{E}_{t,\mathbf{x}_T,\tilde{\mathbf{x}}_t} \left[ \frac{1}{\sum_i m_i} \sum_{i=1}^{|\mathbf{x}_T|} m_i \cdot \text{BCE}(\text{logit}_{R,i}, c_{t,i}[1]) \right] \tag{15}$$

where $m_i$ indicates valid positions and BCE denotes binary cross-entropy with logits.

**Insert Loss**  The insert loss could teach the model to identify positions where additional content is needed. For each sample $\mathbf{x}_T$, one can sample deletion probability $\delta \sim \mathcal{U}(0,1)$ and generate deletion indicators for each position $i$. One would create the deflated sequence $\tilde{\mathbf{x}}$ by removing tokens at randomly selected positions. The insert labels would be $c_{t,j}[2] = 1$ for positions $j$ that remain in $\tilde{\mathbf{x}}$ where the next position was deleted, 0 otherwise:

$$\mathcal{L}_{\text{insert}} = \mathbb{E}_{\delta,\mathbf{x}_T,\tilde{\mathbf{x}}} \left[ \frac{1}{\sum_j m_j} \sum_{j=1}^{|\tilde{\mathbf{x}}|} m_j \cdot \text{BCE}(\text{logit}_{I,j}, c_{t,j}[2]) \right] \tag{16}$$

where $m_j$ indicates valid positions in the deflated sequence.

**Delete Loss**  The delete loss could train the model to distinguish between necessary and redundant mask tokens. One can use a two-step masking process: first apply standard MDM masking with $\alpha_t = t/T$ to create $\mathbf{x}_{\text{base}}$, then sample insertion probability $\gamma \sim \mathcal{U}(0,1)$ to insert additional M tokens at randomly selected positions. The delete labels would distinguish mask origins:

$$c_{t,i}[3] = \begin{cases} 1 & \text{if } \hat{x}_{t,i} = \mathsf{M} \text{ and inserted in step 2} \\ 0 & \text{otherwise} \end{cases} \tag{17}$$

The loss uses binary cross-entropy:

$$\mathcal{L}_{\text{delete}} = \mathbb{E}_{\gamma,\mathbf{x}_T,\hat{\mathbf{x}}_t} \left[ \frac{1}{\sum_k m_k} \sum_{k=1}^{|\hat{\mathbf{x}}_t|} m_k \cdot \text{BCE}(\text{logit}_{D,k}, c_{t,k}[3]) \right] \tag{18}$$

where $m_k$ indicates valid positions in the contracted sequence.

**Combined Training Objective**  In principle, an AP-MDM training objective could balance all four capabilities:

$$\mathcal{L}_{\text{AP-MDM}} = \mathcal{L}_{\text{unmask}} + \lambda_r \mathcal{L}_{\text{remask}} + \lambda_i \mathcal{L}_{\text{insert}} + \lambda_d \mathcal{L}_{\text{delete}} \tag{19}$$

where $\lambda_r, \lambda_i, \lambda_d > 0$ are hyperparameters controlling the relative importance of each operation with default value 1.

### C.3  SUPERVISED TRAINING

In addition to the self-supervised training approach described in § C.2, AP-MDM can also be trained with explicit supervision when state-transition data is available. This applies to scenarios where we have access to the underlying generation process. It can also be combined with the self-supervised approach in § C.2 for hybrid training, e.g. self-supervised in pretraining and supervised in finetuning.

---

**Algorithm 1** Any-Process Generation with **unmask**, **remask**, **insert**, and **delete** Operations

---

**Require:** Trained model $f_\theta$, input prompt $\mathbf{x}$
**Ensure:** Generated sequence $\mathbf{x}_{\text{final}}$
1: $\mathbf{x}_0 \leftarrow \mathbf{x}$          ▷ Initialize with input prompt
2: $t \leftarrow 0$
3: **while** stopping criterion not met **do**
4:      $(\mathbf{x}_t, \mathbf{y}_t, \mathbf{c}_t) \leftarrow f_\theta(\mathbf{x}_t)$          ▷ Model forward pass
5:      $\mathbf{z}_t \leftarrow []$          ▷ Initialize temporary sequence
6:      **for** $i = 1$ to $|\mathbf{x}_t|$ **do**
7:          **if** $c_{t,i}[3] = 1$ and $x_{t,i} = \mathsf{M}$ **then**          ▷ **delete** operation
8:              **continue**          ▷ Skip this position
9:          **end if**
10:          ▷ Determine token value: Remask > Unmask > Keep
11:          **if** $c_{t,i}[1] = 1$ **then**          ▷ **remask** operation
12:              Append $\mathsf{M}$ to $\mathbf{z}_t$
13:          **else if** $x_{t,i} = \mathsf{M}$ **then**          ▷ **unmask** operation
14:              Append $y_{t,i}$ to $\mathbf{z}_t$
15:          **else**          ▷ Keep unchanged
16:              Append $x_{t,i}$ to $\mathbf{z}_t$
17:          **end if**
18:
19:          **if** $c_{t,i}[2] = 1$ **then**          ▷ **insert** operation
20:              Append $\mathsf{M}$ to $\mathbf{z}_t$
21:          **end if**
22:      **end for**
23:      $\mathbf{x}_{t+1} \leftarrow \mathbf{z}_t$
24:      $t \leftarrow t + 1$
25: **end while**
26: **return** $\mathbf{x}_t$

---

**Data Format**     Each training sample consists of a state-transition tuple $(\mathbf{x}_k, \mathbf{y}^*, \mathbf{c}^*)$ where:

- $\mathbf{x}_k \in \bar{\Sigma}^*$: current state (potentially containing $\mathsf{M}$ tokens)

- $\mathbf{y}^* \in \Sigma^{|\mathbf{x}_k|}$: target tokens for each position

- $\mathbf{c}^* = (c_1^*, \ldots, c_{|\mathbf{x}_k|}^*)$ where $c_i^* \in \mathcal{C} = \{0,1\}^3$: ground-truth control signals

**Training Objective**     Given the state-transition data, the model learns to predict both the target tokens and control signals. The training objective consists of four components:

For positions where $x_{k,i} = \mathsf{M}$ and $y_i^* \neq \mathsf{M}$, predict the target token:

$$\mathcal{L}_{\textbf{unmask}}^{\text{sup}} = -\frac{1}{|\{i : x_{k,i} = \mathsf{M}, y_i^* \neq \mathsf{M}\}|} \sum_{i: x_{k,i}=\mathsf{M}, y_i^* \neq \mathsf{M}} \log p_\theta(y_i^* \mid \mathbf{x}_k) \tag{20}$$

For all valid positions, predict the control signals:

$$\mathcal{L}_{\textbf{remask}}^{\text{sup}} = \frac{1}{\sum_i m_i} \sum_{i=1}^{|\mathbf{x}_k|} m_i \cdot \text{BCE}(\text{logit}_{R,i}, c_i^*[1]) \tag{21}$$

$$\mathcal{L}_{\textbf{insert}}^{\text{sup}} = \frac{1}{\sum_i m_i} \sum_{i=1}^{|\mathbf{x}_k|} m_i \cdot \text{BCE}(\text{logit}_{I,i}, c_i^*[2]) \tag{22}$$

$$\mathcal{L}_{\textbf{delete}}^{\text{sup}} = \frac{1}{\sum_i m_i} \sum_{i=1}^{|\mathbf{x}_k|} m_i \cdot \text{BCE}(\text{logit}_{D,i}, c_i^*[3]) \tag{23}$$

where $m_i$ indicates valid positions and BCE denotes binary cross-entropy with logits.

**Combined Objective:**

$$\mathcal{L}_{\text{AP-MDM}}^{\text{sup}} = \mathcal{L}_{\text{unmask}}^{\text{sup}} + \lambda_r \mathcal{L}_{\text{remask}}^{\text{sup}} + \lambda_i \mathcal{L}_{\text{insert}}^{\text{sup}} + \lambda_d \mathcal{L}_{\text{delete}}^{\text{sup}} \tag{24}$$

### C.4 INFERENCE-TIME ALGORITHM

As described in § 4, AP-MDM generation follows an iterative process formulated as $\mathbf{x}_{t+1} = g(f_\theta(\mathbf{x}_t))$. We provide the complete inference-time sampling algorithm that implements the transition function $g$ following Equation (4).

The algorithm implements the transition function $g(\mathbf{x}_t, \mathbf{y}_t, \mathbf{c}_t)$ as defined in Equation (4). Thresholds $\tau_r, \tau_i, \tau_d$ control when operations are applied (default value $0.5$ for all). The algorithm supports variable-length generation through dynamic insertion and deletion, and terminates when a stopping criterion is met (e.g., sequence convergence, generation of special tokens, or reaching a maximum iteration limit).

## D IMPLEMENTATION DETAILS

We provide implementation details for the experiments described in the main paper. All models are built on encoder-only Transformer architecture as described in § F, with task-specific configurations detailed below.

**Model Architecture**    All models are based on encoder-only Transformer architecture with rotary positional embeddings (RoPE). For AP-MDM, we add three binary classification heads (remask, insert, delete) on top of the standard unmask head. We use no timestep embedding and set time conditioning to zero during supervised training. For Sudoku, we use 6 layers, 4 attention heads, hidden dimension $d = 256$, feed-forward dimension $4d = 1024$, maximum sequence length 400, vocabulary size 31. For Parity, we use 1 layer, 1 attention head, hidden dimension $d = 4$, feed-forward dimension $4d = 16$, maximum sequence length 3, vocabulary size 6, with approximately 200 total parameters. For Graph, we use 8 layers, 4 attention heads, hidden dimension $d = 256$, feed-forward dimension $4d = 1024$, maximum sequence length set to accommodate graphs of varying sizes, vocabulary size 55; the same configuration is used for the ARM baseline.

**Training Data Generation**    For supervised training as described in § C.3, training data consists of state-transition tuples $(\mathbf{x}_k, \mathbf{y}^*, \mathbf{c}^*)$ generated by simulating the natural solving or generation algorithms for each task. Starting from ground-truth target solutions, we execute task-specific algorithms and record the intermediate computation states at each step. For each state, we capture the current sequence $\mathbf{x}_k$ (with unknown or unfilled positions represented as $\mathsf{M}$), the target values $\mathbf{y}^*$ to be revealed, and the control decisions $\mathbf{c}^*$ indicating which operations (unmask, remask, insert, delete) should be applied. Concrete examples of data generation for Sudoku, Parity, and Graph tasks are provided in § N, § P, and § O, respectively.

**Training Details**    We train all models using AdamW optimizer with learning rate $10^{-4}$, $(\beta_1, \beta_2) = (0.9, 0.999)$, weight decay 0.01, and batch size 256. Learning rate follows a constant schedule with 250 warmup steps. We use mixed precision training (bfloat16) with gradient clipping at 1.0. Loss weights are set to $\lambda_r = \lambda_i = \lambda_d = 1.0$ by default. Models are trained for up to 1M steps or until convergence. For Graph generation, we use 100K graph instances for training both AP-MDM and the ARM baseline. For Parity, we use only 4 training samples for AP-MDM, while the ARM baseline is trained with up to 10K instances.

## E PARALLEL RANDOM ACCESS MACHINE

The Random Access Machine (RAM) (Arora & Barak, 2009) serves as the foundational theoretical model for sequential computation, featuring a single processor that can access any memory location in unit time regardless of address—hence "random access", along with a finite set of registers and basic arithmetic/logical operations. This contrasts with models like Turing machines where memory

access is sequential. The RAM's key strength lies in its realistic abstraction of modern computers: it captures the essential computational primitives (arithmetic, memory access, conditional branching) while abstracting away hardware details, making it ideal for algorithm analysis.

The Parallel Random Access Machine (PRAM) (Fortune & Wyllie, 1978; JáJá, 1992) extends this familiar RAM model to parallel computation by allowing $P(n)$ processors to operate synchronously on shared memory with $\mathcal{O}(\log n)$-bit word size. [3] Each processor in PRAM maintains its own program counter and unique identifier, enabling conditional branching and coordinated computation. The model operates in discrete synchronous time steps where all active processors execute simultaneously, inheriting RAM's unit-cost random access property while adding the complexity of concurrent memory operations.

**PRAM Variants**  PRAM has several variants, which differ in their memory access discipline, forming a hierarchy with precise complexity relationships. Let EREW, CREW, CRCW-Common, CRCW-Arbitrary and CRCW-Priority denote the classes of problems solvable in polynomial parallel time with polynomially many processors under each model, listed in order of increasing expressivity:

- **EREW** (Exclusive Read, Exclusive Write): No concurrent access to any memory cell. Most restrictive but captures essential parallelism.

- **CREW** (Concurrent Read, Exclusive Write): Multiple processors may read the same cell simultaneously. Enables broadcast in $\mathcal{O}(1)$ time vs. $\Theta(\log n)$ in EREW.

- **CRCW-Common**: Concurrent writes allowed only if all writers agree on the value. Boolean OR computable in $\mathcal{O}(1)$ time.

- **CRCW-Arbitrary**: Any concurrent writer may succeed; the choice is made arbitrarily (often modeled as random selection).

- **CRCW-Priority**: Concurrent writes resolved by processor priority with various schemes (e.g., minimum/maximum index, sum of conflicting values).

Crucially, any algorithm in a stronger model can be simulated in a weaker model with at most $\mathcal{O}(\log n)$ parallel time overhead (JáJá, 1992). This polylogarithmic separation appears in basic primitives, broadcast requires $\Theta(\log n)$ rounds in EREW but $\mathcal{O}(1)$ in CREW, yet the models remain polynomially equivalent for most complexity-theoretic purposes. **We adopt the CREW model throughout this paper**, where different processors are not allowed to write to the same memory cell simultaneously.

PRAM, as an idealized abstraction of shared-memory multiprocessor systems, enables precise analysis of parallel algorithms and gives rise to parallel complexity classes such as NC (Arora & Barak, 2009) (problems solvable in polylogarithmic parallel time using polynomially many processors). For example, PRAM can simulate algorithms on trees, linear arrays, meshes, and hypercubes without loss of parallel time, while reverse simulation costs at most $\mathcal{O}(\log^2 P(n))$ overhead; Boolean circuits of depth $D$ can be simulated on CREW in $\mathcal{O}(D)$ time, making PRAM a natural model for measuring parallel time complexity in theory.

Below, we provide a more formal definition that will be used in proofs.

E.1  DEFINITION AND EXECUTION PROCESS OF WORD-RAM

We formalize the standard word-RAM that matches a single-processor PRAM (i.e., $P(n) = 1$). Throughout, let the input length be $n$ and fix the word size $w(n) = \Theta(\log n)$.

**Word Size, Universe, and Addresses.**  Let the word universe be $\mathbb{U} = \{0, 1, \ldots, 2^w - 1\}$ with arithmetic modulo $2^w$ (two's-complement semantics). The address space is $\mathcal{A} = \{0, 1, \ldots, S(n)-1\}$ for some $S(n) \leq n^{\mathcal{O}(1)}$. Memory is a mapping $M : \mathcal{A} \to \mathbb{U}$, *zero-initialized.*

---

[3]The $\mathcal{O}(\log n)$-bit word size choice ensures that pointer arithmetic, indexing, and basic integer operations on polynomially bounded values are unit-time, matching the standard RAM assumptions and avoiding artificial speedups due to unrealistically wide words.

Let $a(n) = \lceil \log_2 S(n) \rceil$ denote the address width. We adopt the transdichotomous condition:

$$w(n) \geq \max\{\lceil \log_2 S(n) \rceil,\ \lceil \log_2 P(n) \rceil\} \qquad \text{and} \qquad w(n) = \Theta(\log n). \qquad (25)$$

This ensures every address and processor ID fits in one word, enabling well-typed register-indirect addressing and processor identification.

**Instruction Set and Semantics.** The machine operates with register names $\mathsf{Reg} = \{0, 1, \ldots, k\}$ (for constant $k$), register file $R \in \mathbb{U}^{k+1}$, immediate constants $\mathsf{Imm} \subset \mathbb{Z}$ (a fixed finite set independent of $n$ and $w$), and label identifiers $\mathsf{Lab}$ for jump targets.[4] We assume a constant-size register file with $|\mathsf{Reg}| \geq 2$ in the proof (any constant $\geq 2$ suffices up to constant factors). Programs define a partial label table $\mathrm{addr} : \mathsf{Lab} \rightharpoonup \{0, \ldots, \ell\}$ mapping each declared label to its instruction index (injective).

The instruction alphabet $\mathsf{Instr}$ consists of the following parameterized forms ($r, s \in \mathsf{Reg}$, $c \in \mathsf{Imm}$, $L \in \mathsf{Lab}$):

$$\mathsf{Instr} = \{\ \texttt{LOAD}\ r,\ [s],\ \texttt{STORE}\ [s],\ r,\ \texttt{LOADI}\ r,\ c\ \}$$
$$\cup\ \{\ \texttt{ADD}\ r,\ s,\ \texttt{SUB}\ r,\ s,\ \texttt{AND}\ r,\ s,\ \texttt{XOR}\ r,\ s,\ \texttt{SHL}\ r,\ s,\ \texttt{SHR}\ r,\ s\ \}$$
$$\cup\ \{\ \texttt{BRZ}\ r,\ L,\ \texttt{JMP}\ L,\ \texttt{HALT}\ \}.$$

Unbracketed registers $r, s$ denote their word values $R_r, R_s \in \mathbb{U}$. The bracketed form $[s]$ denotes register-indirect addressing: $\texttt{LOAD}\ r, [s]$ reads $M[R_s]$ into $R_r$, and $\texttt{STORE}\ [s], r$ writes $R_r$ to $M[R_s]$. Bracketed operands are only allowed in $\texttt{LOAD}/\texttt{STORE}$; nested or arithmetic addressing (e.g., $[[s]], [r+c]$) is not part of this ISA. If $R_s \notin \mathcal{A}$, execution traps. Immediates are loaded as $c \bmod 2^w$.

The semantics of the instructions are as follows. We write $\sigma \to \sigma'$ for one execution step. Unless a jump changes it, set $\mathrm{pc} \leftarrow \mathrm{pc} + 1$ where $\mathrm{pc} \in \{0, \ldots, \ell\} \cup \{\texttt{HALT}\}$ is the program counter. Let $\oplus$ and $\wedge$ denote bitwise XOR and AND; let $\ll$ and $\gg$ denote logical shifts; all arithmetic is modulo $2^w$.

- $\texttt{LOAD}\ r, [s]$: $\ a \leftarrow R_s$; $R_r \leftarrow M[a]$.
- $\texttt{STORE}\ [s], r$: $\ a \leftarrow R_s$; $M[a] \leftarrow R_r$.
- $\texttt{LOADI}\ r, c$: $\ R_r \leftarrow c \bmod 2^w$.
- $\texttt{ADD}\ r, s\ /\ \texttt{SUB}\ r, s$: $\ R_r \leftarrow (R_r \pm R_s) \bmod 2^w$.
- $\texttt{AND}\ r, s\ /\ \texttt{XOR}\ r, s$: $\ R_r \leftarrow R_r \wedge R_s\ \ /\ \ R_r \leftarrow R_r \oplus R_s$.
- $\texttt{SHL}\ r, s$ or $\texttt{SHR}\ r, s$: $\ h \leftarrow R_s \bmod w$; $\texttt{SHL}: R_r \leftarrow (R_r \ll h) \bmod 2^w$; $\texttt{SHR}: R_r \leftarrow \lfloor R_r / 2^h \rfloor$ (logical right shift, zero fill).
- $\texttt{BRZ}\ r, L$: If $R_r = 0$ then $\mathrm{pc} \leftarrow \mathrm{addr}(L)$ else (no change to pc beyond $+1$).
- $\texttt{JMP}\ L$: $\mathrm{pc} \leftarrow \mathrm{addr}(L)$.
- $\mathrm{pc} \leftarrow \texttt{HALT}$ and execution stops.

Intuitively, $\texttt{LOAD}$ and $\texttt{STORE}$ handle memory access through register-indirect addressing, $\texttt{LOADI}$ loads immediate constants, $\texttt{ADD}/\texttt{SUB}$ perform modular arithmetic, $\texttt{AND}/\texttt{XOR}$ enable bitwise operations, $\texttt{SHL}/\texttt{SHR}$ provide bit shifts. $\texttt{BRZ}$ (branch if zero) enables conditional branching, $\texttt{JMP}$ provides unconditional jumps, and $\texttt{HALT}$ terminates execution.

**Programs and Configurations.** A *program* is a pair $\mathcal{P} = (I_0, \ldots, I_\ell, \mathrm{addr})$ with $I_i \in \mathsf{Instr}$ and a partial label table $\mathrm{addr} : \mathsf{Lab} \rightharpoonup \{0, \ldots, \ell\}$ mapping each declared label to its instruction index (injective). The program is *well formed* if whenever some $I_i$ equals $\texttt{JMP}\ L$ or $\texttt{BRZ}\ r, L$, then $L \in \mathrm{dom}(\mathrm{addr})$. Code is immutable during execution and independent of $n$ (and thus the RAM model considered here is uniform). A *configuration* is $\sigma = (\mathrm{pc}, R, M)$ where $\mathrm{pc} \in \{0, \ldots, \ell\} \cup \{\texttt{HALT}\}$ is the program counter, $R \in \mathbb{U}^{k+1}$ is the register file, and $M : \mathcal{A} \to \mathbb{U}$ is memory. Input occupies $M[0..n-1]$; output is read from a designated location upon termination.

**Initialization.** Given an input instance of length $n$, initialization proceeds as follows:

---

[4]A label is a human-readable name for a program location (instruction index) serving as a jump/branch target.

1. Build the label table $\mathrm{addr}$ from the loaded code and check well-formedness (every label operand in the code must appear exactly once as a declared label).

2. Zero-initialize memory $M$ and write the input into a designated block (e.g., $M[0..n-1]$) using the agreed-upon encoding.

3. Set all registers to zero: $R_i \leftarrow 0$ for $i \in \{0, \dots, k\}$.

4. Set the program counter to the first instruction: $\mathrm{pc} \leftarrow 0$.

The choice $w \geq \lceil \log_2 S(n) \rceil$ ensures that register-indirect addressing is well-typed: a bracketed operand $[s]$ uses $R_s$ as an address in $\mathcal{A}$.

**Execution Cycle.** While $\mathrm{pc} \neq \texttt{HALT}$ and no trap occurs, the machine advances in discrete steps. Each successful step costs one time unit. Let $I_{\mathrm{pc}}$ denote the instruction at index $\mathrm{pc}$. Each step follows the fetch-decode-execute-commit cycle:

1. **Fetch**: Read the current instruction $I \leftarrow I_{\mathrm{pc}}$. If $\mathrm{pc} \notin \{0, \dots, \ell\}$, the run is invalid and we define this as a trap.

2. **Decode and read operands**: Parse the opcode and operands of $I$ without changing the machine state. Unbracketed registers $r, s$ denote their current word values $R_r, R_s \in \mathbb{U}$ (used as data). A bracketed operand $[s]$ denotes the candidate address $a \leftarrow R_s$. An immediate $c \in \mathsf{Imm}$ is interpreted as $c \bmod 2^w$. A label $L$ resolves to $\mathrm{addr}(L)$ (guaranteed by well-formedness). No writes occur in this phase.

3. **Execute**: Apply the instruction semantics of $I$ to compute a finite *write-set* $W$ (register and/or memory locations with their new values) and the *next program counter* $\mathrm{pc}_{\mathrm{next}}$. For memory-referencing instructions, a bracketed operand $[s]$ is valid only if $a = R_s \in \mathcal{A}$; otherwise a trap occurs. By default $\mathrm{pc}_{\mathrm{next}} = \mathrm{pc} + 1$, except for jumps/branches/halting which set it to $\mathrm{addr}(L)$ (or $\texttt{HALT}$).

4. **Commit (writeback)**: Atomically apply the writes in $W$ to $(R, M)$ and then set $\mathrm{pc} \leftarrow \mathrm{pc}_{\mathrm{next}}$. Atomicity means all effects of the step become visible only at the end of the step.

5. **Cost and continuation**: If no trap occurred, charge one unit of time for this step and proceed to the next; otherwise the run aborts (abnormal termination), and only successfully committed steps are counted.

**Termination and Complexity.** Execution halts when $\mathrm{pc} = \texttt{HALT}$. The algorithm's output is read from the designated output location(s) in memory (or registers) as specified by the program. Under the assumptions above and for well-formed programs with legal memory accesses, the step relation is deterministic and yields a unique next state at each iteration. The *time complexity* of an algorithm is the number of executed instructions before halting. A *trap* aborts the run immediately (abnormal termination); only successfully committed steps are counted in time.

The RAM model defined here is polynomially equivalent to bit-complexity RAM (a $\Theta(\log n)$ factor separates their running times) and to richer word-RAMs that add $\texttt{MUL}/\texttt{DIV}/\texttt{POPCNT}/\texttt{CLZ}$ (whose presence typically improves only by constant or $\log \log n$ factors).

### E.2 Extension to CREW PRAM

We extend the Word-RAM defined above to a parallel machine with a processor-budget function $P : \mathbb{N} \to \mathbb{N}$ (typically $P(n) \leq n^{\mathcal{O}(1)}$). All word-size/address-width assumptions, the instruction alphabet $\mathsf{Instr}$, the immediate-set restriction, and the *single-processor* instruction semantics are exactly as in the Word-RAM subsection.

**Processors and Shared State.** Processors are indexed by $i \in \{0, \dots, P(n) - 1\}$. Each processor has its own program counter and register file; memory is shared:

$$\Sigma = \big((\mathrm{pc}_0, \dots, \mathrm{pc}_{P(n)-1}), (R^0, \dots, R^{P(n)-1}), M\big),$$

where $\mathrm{pc}_i \in \{0, \dots, \ell\} \cup \{\texttt{HALT}\}$ and $R^i = (R_0^i, \dots, R_k^i) \in \mathbb{U}^{k+1}$. All processors run the same program $\mathcal{P} = (I_0, \dots, I_\ell, \mathrm{addr})$.

---

**Algorithm 2** Single-Processor Execution (Word-RAM semantics with PID init)

---

*Note.* In PRAM, STORE generates a pending write committed at the end of the round under the CREW rule. In the single-processor case, the store can be applied immediately.

**Require:** Program $\mathcal{P} = (I_0, \ldots, I_\ell, \mathrm{addr})$, shared memory $M : \mathcal{A} \to \mathbb{U}$, word size $w$, processor id pid, processor budget $P(n)$ (optional)

1: **Init:** $pc \leftarrow 0$; $R[j] \leftarrow 0$ for all $j$; $R[0] \leftarrow$ pid; **optional:** $R[1] \leftarrow P(n) \bmod 2^w$
2: **while** $pc \neq \mathrm{HALT}$ **do**
3:     $I \leftarrow I_{pc}$                                                            $\triangleright$ fetch
4:     $pc_{\text{next}} \leftarrow pc + 1$                                      $\triangleright$ default fall-through
5:     **if** $I$ is LOAD $r, [s]$ **then**                            $\triangleright$ decode
6:         $a \leftarrow R[s]$
7:         **if** $a \notin \mathcal{A}$ **then**
8:             **trap**
9:         **end if**
10:         $R[r] \leftarrow M[a]$                                    $\triangleright$ execute
11:     **else if** $I$ is STORE $[s], r$ **then**
12:         $a \leftarrow R[s]$
13:         **if** $a \notin \mathcal{A}$ **then**
14:             **trap**
15:         **end if**
16:         $M[a] \leftarrow R[r]$    $\triangleright$ in PRAM semantics, this is a write event to be committed this round
17:     **else if** $I$ is LOADI $r, c$ **then**
18:         $R[r] \leftarrow c \bmod 2^w$
19:     **else if** $I$ is ADD $r, s$ **then**
20:         $R[r] \leftarrow (R[r] + R[s]) \bmod 2^w$
21:     **else if** $I$ is SUB $r, s$ **then**
22:         $R[r] \leftarrow (R[r] - R[s]) \bmod 2^w$
23:     **else if** $I$ is AND $r, s$ **then**
24:         $R[r] \leftarrow R[r] \wedge R[s]$                              $\triangleright$ bitwise AND
25:     **else if** $I$ is XOR $r, s$ **then**
26:         $R[r] \leftarrow R[r] \oplus R[s]$                              $\triangleright$ bitwise XOR
27:     **else if** $I$ is SHL $r, s$ **then**
28:         $h \leftarrow R[s] \bmod w$
29:         $R[r] \leftarrow (R[r] \ll h) \bmod 2^w$
30:     **else if** $I$ is SHR $r, s$ **then**
31:         $h \leftarrow R[s] \bmod w$
32:         $R[r] \leftarrow \lfloor R[r]/2^h \rfloor$                   $\triangleright$ logical right shift, zero-fill
33:     **else if** $I$ is BRZ $r, L$ **then**
34:         **if** $R[r] = 0$ **then**
35:             $pc_{\text{next}} \leftarrow \mathrm{addr}(L)$
36:         **end if**
37:     **else if** $I$ is JMP $L$ **then**
38:         $pc_{\text{next}} \leftarrow \mathrm{addr}(L)$
39:     **else if** $I$ is HALT **then**
40:         $pc_{\text{next}} \leftarrow \mathrm{HALT}$
41:     **else**
42:         **trap**                          $\triangleright$ unknown opcode or malformed operands
43:     **end if**
44:     $pc \leftarrow pc_{\text{next}}$         $\triangleright$ commit PC; regs/memory updated in each branch above
45: **end while**

---

**Initialization (with Processor IDs).** At time $t = 0$:

1. Build $\mathrm{addr}$ and check well-formedness (as in Word-RAM); zero-initialize $M$ and write the input block.

2. For each $i \in \{0, \ldots, P(n) - 1\}$, set $pc_i \leftarrow 0$ and clear registers; then write *processor-local identifiers:* $\underline{R_0^i \leftarrow i}$ and, if $P(n) \leq 2^{w(n)}$, optionally $\underline{R_1^i \leftarrow P(n) \bmod 2^{w(n)}}$. All other $R_j^i \leftarrow 0$.

These two words are provided so that processors can branch, partition work, and self-disable if unused.

**Concurrent-Access Policy (CREW).**   Multiple processors may *read* the same address in the same round; *writes must be exclusive*: if two or more writes target the same address in a round, the run traps (abnormal termination).

**Round Semantics (Referencing the Word-RAM Step).**   Each active processor executes exactly one instruction using the single-processor Word-RAM step semantics; the only new aspects are (i) simultaneous execution by many processors and (ii) end-of-round memory commit subject to the CREW policy. Execution proceeds in synchronous rounds $t = 0, 1, 2, \ldots$ with state $\Sigma_t = ((\mathrm{pc}_i^t)_i, (R^{i,t})_i, M^t)$. In round $t$, each active processor $i$ with $\mathrm{pc}_i^t \in \{0, \ldots, \ell\}$ executes instruction $I_{\mathrm{pc}_i^t}$ on its local snapshot ($\mathrm{pc} = \mathrm{pc}_i^t$, $R = R^{i,t}$) and shared memory $M^t$. After all processors compute their local effects, the round commits: register writebacks $R^{i,t} \to R^{i,t+1}$ (independently), then memory writes to $M^{t+1}$ under CREW constraints, finally program counter updates $\mathrm{pc}_i^{t+1}$.

**Termination and Cost Measures.**   The parallel run terminates when $\mathrm{pc}_i^t = \texttt{HALT}$ for all $i$ (or traps on an invalid access/conflict). One round costs one unit of *parallel time*. The *work* is the total number of executed instructions $W(n) = \sum_t |\{\, i : \mathrm{pc}_i^t \in \{0, \ldots, \ell\} \,\}|$, and the *span* is $T_\infty(n)$ (the critical-path length). For $P(n)$ processors the Brent bound holds (JáJá, 1992):

$$T_{P(n)}(n) \;\le\; \left\lceil \frac{W(n)}{P(n)} \right\rceil + T_\infty(n).$$

A single-processor run ($P(n) = 1$) coincides with the Word-RAM model.

**Remarks (on $P(n)$, Processor IDs, and Unused Processors).**   (1) *Uniformity:* the code $\mathcal{P}$ and the immediate set Imm are independent of $n$; only the hardware parameters $w(n)$, $S(n)$, and $P(n)$ scale with input size. (2) *Processor IDs:* the values $i$ and $P(n)$ are provided via initialization registers ($R_0^i$ and optionally $R_1^i$) for branching and work partitioning; programs may copy/overwrite them. (3) *Unused processors:* if an algorithm needs only $m(n) \le P(n)$ processors, each processor executes a short self-filter based on $i$ (e.g., $\texttt{if } i \ge m(n) \texttt{ then HALT}$), or computes its assigned block; processors with empty assignment halt in $O(1)$ rounds, which does not affect the asymptotic parallel time.

The algorithm for a single-processor in PRAM is shown in Algorithm 2.

## F   ENCODER TRANSFORMER ARCHITECTURE

This section presents encoder-only Transformers, which form the backbone of MDM. We will first establish the sequence-wise extension operation, then define the core components, including bidirectional self-attention, multi-head mechanisms, and feed-forward layers, before assembling the complete architecture.

We consider an encoder-only Transformer with $H$ heads, $L$ layers, hidden size $d$, and feed-forward width $w$. We will use the following notations:

**Definition 4** (Position-Indexed Seq-to-Embedding Function). *For a set $B$, let $\mathcal{H}(B)$ denote the set of all functions $\psi$ such that for every sequence $x = (x_1, \ldots, x_n) \in \Sigma^*$ and every index $i \in [n]$, the value $\psi(x, i) \in B$ is defined. We write this succinctly as*

$$\psi : (\Sigma^*, \mathbb{N}) \to B. \tag{26}$$

*and call this* position-indexed seq-to-embedding function. *We also define $\mathcal{H} = \cup_{d \in \mathbb{N}^+} \mathcal{H}(\mathbb{R}^d)$ as the union of all such classes across real spaces of all output dimensions.*

**Definition 5** (Canonical Extension to Seq-to-Seq Function). *Given a position-indexed seq-to-embedding function $\psi \in \mathcal{H}(B)$, its* canonical extension *is defined as:*

$$\overline{\psi} : \Sigma^* \to B^* \quad where \quad [\overline{\psi}(x)]_i = \psi(x, i) \qquad (i \in [|x|]). \tag{27}$$

*For elementwise functions $g : \mathbb{R}^d \to \mathbb{R}^{d'}$, we define $\overline{g} : (\mathbb{R}^d)^* \to (\mathbb{R}^{d'})^*$ by $[\overline{g}(h_{1:n})]_i = g(h_i)$, which is a special case where $\psi(h_{1:n}, i) = g(h_i)$ (ignoring cross-position context). When the arity is clear, we reuse the bar notation for both position-indexed and elementwise extensions.*

We now define the individual components of encoder Transformers:

**Bidirectional Self-Attention.** The key difference from decoder Transformers is bidirectional attention, where each position can attend to all positions in the sequence. Let $d_h$ be the head dimension. For $W_Q, W_K, W_V \in \mathbb{R}^{d_h \times d}$ and $W_O \in \mathbb{R}^{d \times d_h}$, we define single-head attention on sequence of embeddings $h_{1:n} \in (\mathbb{R}^d)^n$ for any $n \in \mathbb{N}^+$:

$$q_i = W_Q h_i, \quad k_j = W_K h_j, \quad v_j = W_V h_j \tag{28}$$

$$[\mathrm{SA}_\theta(h_{1:n})]_i = W_O \sum_{j=1}^{n} \alpha_{ij} v_j \tag{29}$$

with $\alpha_{i,\cdot} = \mathrm{softmax}\big((q_i^\top k_j)_{j=1}^n\big)$, $\theta = (W_Q, W_K, W_V, W_O)$. Position $i$ attends to all $j \in [n]$ without causal restrictions. We use standard $1/\sqrt{d_h}$ scaling.

**Multi-Head Attention.** With $\theta_{\mathrm{MHA}} = (\theta^{(1)}, \ldots, \theta^{(H)})$, we combine heads via summation:

$$[\mathrm{MHA}_{\theta_{\mathrm{MHA}}}(h_{1:n})]_i = \sum_{t=1}^{H} [\mathrm{SA}_{\theta^{(t)}}(h_{1:n})]_i \tag{30}$$

for any $i \in [n]$. Note that this differs from practical implementations which concatenate heads with dimension $d/H$ each, but maintains equivalent theoretical expressivity.

**Feed-Forward and Projection.** Let $w = d_{\mathrm{FF}}$. For $W_1 \in \mathbb{R}^{w \times d}$ and $W_2 \in \mathbb{R}^{d \times w}$:

$$\mathrm{FF}_\theta(h) = W_2 \, \sigma(W_1 h) \tag{31}$$

For output projection, $\mathrm{PROJ}_\vartheta : \mathbb{R}^d \to \mathbb{R}^{|\Sigma|}$ with $\mathrm{PROJ}_\vartheta(h) = \vartheta h$ and $\vartheta \in \mathbb{R}^{|\Sigma| \times d}$. We apply these via sequence-wise extension: $\overline{\mathrm{FF}}$ and $\overline{\mathrm{PROJ}}$.

For AP-MDM as described in § 4, besides the above heads for **unmask**, it would require three additional binary classification heads on top of the final layer: $\mathrm{PROJ}_R : \mathbb{R}^d \to \mathbb{R}$ for **remask**, $\mathrm{PROJ}_I : \mathbb{R}^d \to \mathbb{R}$ for **insert**, and $\mathrm{PROJ}_D : \mathbb{R}^d \to \mathbb{R}$ for **delete** operations, each followed by sigmoid activation. Therefore, $\mathrm{PROJ}_\vartheta$ is a mapping from $\mathbb{R}^d$ to $\mathbb{R}^{|\Sigma|+3}$.

**Embeddings.** Define token embedding $\mathrm{TE} : \Sigma \to \mathbb{R}^d$ and positional embedding $\mathrm{PE} : \mathbb{N}^+ \to \mathbb{R}^d$ (which can be flexibly chosen and will be specified when used). Combined as $\overline{\mathrm{TE}} + \overline{\mathrm{PE}}$. We write $\mathrm{TE}, \mathrm{PE}$ for their sequence-wise extensions when clear from context.

**Residual Connections.** The identity function $\mathrm{Id}_d : \mathbb{R}^d \to \mathbb{R}^d$ is defined by $\mathrm{Id}_d(x) = x$. A *residual connection* is defined as $f + \mathrm{Id}_d$, where $\mathrm{Id}_d$ is the identity function.

Next, we assemble these components into the encoder transformer architecture:

**Definition 6** (Encoder Transformer layer). *An encoder layer is defined as:*

$$\mathrm{EncTF}_{\mathrm{MHA},\mathrm{FF}} = (\overline{\mathrm{FF}_{\mathrm{FF}}} + \overline{\mathrm{Id}_d}) \circ (\mathrm{MHA}_{\theta_{\mathrm{MHA}}} + \overline{\mathrm{Id}_d}) : (\mathbb{R}^d)^* \to (\mathbb{R}^d)^* \tag{32}$$

**Definition 7** (Encoder Transformer). *With parameters* $\theta = (\theta_{\mathrm{TE}}, \theta_{\mathrm{PE}}, (\theta_{\mathrm{MHA}}^{(\ell)})_{\ell=1}^L, (\theta_{\mathrm{FF}}^{(\ell)})_{\ell=1}^L, \theta_{\mathrm{PROJ}})$, *the encoder transformer is:*

$$\mathrm{Enc}_\theta = \overline{\mathrm{PROJ}_{\theta_{\mathrm{PROJ}}}} \circ \left( \bigcirc_{\ell=1}^L \mathrm{EncTF}_{\theta_{\mathrm{MHA}}^{(\ell)}, \theta_{\mathrm{FF}}^{(\ell)}} \right) \circ \left( \mathrm{TE}_{\theta_{\mathrm{TE}}} + \mathrm{PE}_{\theta_{\mathrm{PE}}} \right) \tag{33}$$

*The model applies embeddings, then $L$ encoder layers, then position-wise projection to vocabulary logits. Output length equals input length $n$.*

## G  KEY TOOL: ENCODER FULL-ACCESS SEQUENCE PROCESSING (E-FASP)

In this section, we develop Full-Access Sequence Processing for encoders (E-FASP), a programming language whose programs describe the construction process of seq-to-embedding functions that are equivalent to those computed by encoder-only Transformers. This extends the FASP framework originally developed for decoder-only Transformers in Yang et al. (2025). Similar connections have also been established in Weiss et al. (2021); Yang & Chiang (2024).

E-FASP is the key technical tool that will be used to prove Theorem 1, Theorem 3 and Theorem 4.

### G.1   Definition of E-FASP

**Notations.**   Recall that in § F, we defined the position-indexed seq-to-embedding function space $\mathcal{H}(B)$ as the set of all functions $\psi$ that map a sequence and a position index to an element in $B$:

$$\psi : (\Sigma^*, [\cdot]) \to B \tag{34}$$

That is, for every sequence $\mathbf{x} = (x_1, \ldots, x_n) \in \Sigma^*$ and every index $i \in [n]$, we have $\psi(\mathbf{x}, i) \in B$. We also define $\mathcal{H} = \cup_{d \in \mathbb{N}^+} \mathcal{H}(\mathbb{R}^d)$ as the union of all such classes across real spaces of all output dimensions. For any position-indexed function $\psi \in \mathcal{H}(B)$, its canonical extension $\overline{\psi} : \Sigma^* \to B^*$ is defined by $[\overline{\psi}(\mathbf{x})]_i = \psi(\mathbf{x}, i)$ for $i \in [|\mathbf{x}|]$. This allows us to convert position-indexed functions to sequence-to-sequence functions when needed.

Also recall that $\mathrm{PE} : \mathbb{N}^+ \to \mathbb{R}^d$ is a positional embedding, and we additionally define $\mathcal{T}_{\mathrm{ACT}}$ as a class of activation functions. We formally define E-FASP as follows:

**Definition 8** (Encoder-FASP). *An E-FASP program is a sequence of position-indexed seq-to-embedding functions $\psi_1, \ldots, \psi_T$ constructed inductively. At each step $t \in [T]$, the program maintains a set of defineable position-indexed seq-to-embedding functions $\mathcal{P}_t$, and defines a new function by applying operators to functions in $\mathcal{P}_t$. We define the defineable functions at step $t \in [T]$:*

$$\mathcal{P}_t \triangleq \{\mathrm{TE}, \mathrm{PE}\} \cup \{\psi_i \mid 1 \leq i \leq t - 1\} \tag{35}$$

*where $\mathrm{TE}(\mathbf{x}, i) = \mathrm{TE}(x_i)$ and $\mathrm{PE}(\mathbf{x}, i) = \mathrm{PE}(i)$ are the token and positional embedding functions respectively, viewed as position-indexed seq-to-embedding functions. $\psi_t$ at step $t$ has to be defined by applying one of the following four primitive operators on already-defined functions from $\mathcal{P}_t$:*

1. ***Concatenation****: For $\psi, \psi' \in \mathcal{P}_t$ with $\psi \in \mathcal{H}(\mathbb{R}^{d_1})$ and $\psi' \in \mathcal{H}(\mathbb{R}^{d_2})$, define*

$$[\psi, \psi'](\mathbf{x}, i) = (\psi(\mathbf{x}, i) \| \psi'(\mathbf{x}, i)) \in \mathbb{R}^{d_1 + d_2} \tag{36}$$

   *where $\|$ denotes vector concatenation.*

2. ***Linear Projection****: For $\psi \in \mathcal{P}_t$ with $\psi \in \mathcal{H}(\mathbb{R}^d)$ and matrix $W \in \mathbb{R}^{d' \times d}$, define*

$$(W \circ \psi)(\mathbf{x}, i) = W \cdot \psi(\mathbf{x}, i) \in \mathbb{R}^{d'} \tag{37}$$

3. ***Nonlinear Activation****:[5] For $\psi \in \mathcal{P}_t$ with $\psi \in \mathcal{H}(\mathbb{R}^d)$ and $\sigma \in \mathcal{T}_{\mathrm{ACT}}$, define*

$$(\sigma \circ \psi)(\mathbf{x}, i) = \sigma(\psi(\mathbf{x}, i)) \tag{38}$$

4. ***Encoder Average-Hard Attention****: For $q, k \in \mathcal{H}(\mathbb{R}^d)$ and $v \in \mathcal{H}(\mathbb{R}^{d'})$ where $q, k, v \in \mathcal{P}_t$, define*

$$\mathtt{aha}(q, k, v)(\mathbf{x}, i) = \frac{1}{|A_i|} \sum_{j \in A_i} v(\mathbf{x}, j) \tag{39}$$

   *where $A_i = \arg\max_{j \in [|\mathbf{x}|]} \langle q(\mathbf{x}, i), k(\mathbf{x}, j) \rangle$ and ties are averaged uniformly. This attention can be seen as a special case of standard softmax attention with temperature approaching 0 (Merrill et al., 2022).*

*Finally, when we want to use E-FASP to define a function mapping from a sequence of tokens $\Sigma^*$ and a position index $i$ to a single token in $\Sigma$, we can define $\psi \in \mathcal{H}(\mathbb{R}^{|\Sigma|})$ and return $\arg\max \psi(\mathbf{x}, i)$ (the token corresponding to the largest logit at the position $i$).[6]*

We denote the set of all position-indexed seq-to-embedding functions definable by E-FASP with position embedding PE and activation functions $\mathcal{T}_{\mathrm{ACT}}$ as E-FASP$[\mathrm{PE}; \mathcal{T}_{\mathrm{ACT}}]$, where PE can be either BiPE or SEQ. The expressivity of E-FASP depends on the specific positional embedding and activation functions used.

---

[5]We allow multi-variable activation functions like Gated ReLU (ReGLU), $x, y \mapsto x[y]_+$.

[6]We could assume an arbitrary order to break ties, but we omit this for simplicity. In our examples we always ensure the argmax is unique.

## G.2 EQUIVALENCE WITH ENCODER TRANSFORMER

We now establish the equivalence between E-FASP and encoder-only Transformers, and define the specific instantiation considered in the proof of this paper.

**Definition 9** (Encoder Transformer Function Class). *Let $\mathcal{H}_{EncTF[\text{PE};\mathcal{T}_{\text{ACT}}]}$ be the class of seq-to-embedding functions that can be expressed by encoder-only Transformers of finite depth, where the positional embedding uses* PE *(either* BiPE *or* SEQ*), feed-forward layers use activation functions from $\mathcal{T}_{\text{ACT}}$, attention layers use average-hard attention as defined in Equation* (39)*, and all intermediate computations use finite precision arithmetic.*

It is straightforward to see that both variants are equivalent to their corresponding Transformer function classes:

**Lemma 7** (Equivalence of E-FASP and Encoder Transformers). *For any positional embedding* PE $\in \{$ BiPE, SEQ $\}$ *and activation function class $\mathcal{T}_{\text{ACT}}$, the following equivalence holds:*

$$E\text{-}FASP[\text{PE}; \mathcal{T}_{\text{ACT}}] = \mathcal{H}_{EncTF[\text{PE};\mathcal{T}_{\text{ACT}}]} \tag{40}$$

*Proof Sketch.* **Forward direction**: Each E-FASP primitive operator (concatenation, linear projection, nonlinear activation, encoder attention) directly corresponds to operations in encoder Transformers. Concatenation involves merging multiple smaller Transformers into a larger Transformer that produces the same output. **Reverse direction**: Any encoder Transformer can be expressed as an E-FASP program by decomposing each layer into primitive operations.

The detailed proof, including the treatment of closed operators and inductive construction, is invariant to decoder or encoder Transformers, and thus is identical to Yang et al. (2025); we omit the details. □

Intuitively, this equivalence holds because E-FASP programs capture the computational structure of encoder Transformers. Each step in an E-FASP program corresponds to defining a new seq-to-embedding function by applying primitive operators to previously defined functions, which mirrors how smaller Transformers are constructed into a deeper and wider Transformer that produces the same output. The Transformer corresponding to the program is of depth $\mathcal{O}(1)$ (given constant $T$) and embedding size $\mathcal{O}(\max\{d_{\text{PE}}, d_{\text{TE}}\})$, by construction in the proof of Lemma 7.

## G.3 TWO VARIANTS OF E-FASP

Throughout this paper, we consider two variants of E-FASP based on different positional embeddings, both using the same activation functions.

**Variant 1: Binary Positional Encoding** We define E-FASP with binary positional embedding BiPE : $\mathbb{N}^+ \to \{0,1\}^{\lceil \log_2 S(n) \rceil}$:

$$\text{BiPE}(i) = \text{binary representation of } i \text{ using } \lceil \log_2 S(n) \rceil \text{ bits} \tag{41}$$

This representation uses $\lceil \log_2 S(n) \rceil$ bits to represent all possible positions within the maximum context length $S(n)$, and aligns with the address representation in PRAM (§ E) for efficient bitwise arithmetic operations.

**Variant 2: Integer Positional Encoding** We also define E-FASP with integer positional embedding SEQ : $\mathbb{N}^+ \to \mathbb{N}^+$:

$$\text{SEQ}(i) = i \tag{42}$$

This is the identity mapping over $\mathbb{N}^+$ that directly uses the position index as a scalar feature, as considered in the original decoder-only FASP framework (Yang et al., 2025).

**Activation Functions** Both variants use the same class of activation functions $\mathcal{T}_{\text{ACT}} = \{\text{ReGLU}\}$, where Gated ReLU (ReGLU) (Dauphin et al., 2017) is defined as $\text{ReGLU}(x, y) = x \cdot [y]_+ = x \cdot \max(y, 0)$ for $x, y \in \mathbb{R}$. With Gated ReLU as the primitive activation, we can express ReLU and multiplication operations through the following identities:

$$\text{ReLU}(x) = \text{ReGLU}(x, 1), \quad x \times y = \text{ReGLU}(x, y) - \text{ReGLU}(x, -y) \tag{43}$$

Therefore, having ReGLU allows us to express both ReLU and multiplication (reverse is also true), making both variants equivalent:

$$\text{E-FASP}[\text{BiPE}; \text{ReGLU}] = \text{E-FASP}[\text{BiPE}; [\cdot]_+, \times] \tag{44}$$

$$\text{E-FASP}[\text{SEQ}; \text{ReGLU}] = \text{E-FASP}[\text{SEQ}; [\cdot]_+, \times] \tag{45}$$

where $[\cdot]_+$ and $\times$ are the ReLU and multiplication respectively.

### G.4 Original Supported Operators

With the four primitive operators in E-FASP and the activation functions defined above, the following operators can be included in both variants E-FASP$[\text{BiPE}; [\cdot]_+, \times]$ and E-FASP$[\text{SEQ}; [\cdot]_+, \times]$, adapted from the decoder version of FASP:

*Arithmetic Operators*

- $\text{add}(\psi_1, \psi_2) = \psi_1 + \psi_2$: Element-wise addition
- $\text{minus}(\psi_1, \psi_2) = \psi_1 - \psi_2$: Element-wise subtraction
- $\text{multi}(\psi_1, \psi_2) = \psi_1 \times \psi_2$: Element-wise multiplication
- $\text{max}(\psi_1, \psi_2)$: Element-wise maximum
- $\text{min}(\psi_1, \psi_2)$: Element-wise minimum

*Boolean Operators* For $\psi_1, \psi_2 \in \mathcal{H}(\{0, 1\})$:

- $\text{and}(\psi_1, \psi_2) = \min(\psi_1, \psi_2)$: Logical AND
- $\text{or}(\psi_1, \psi_2) = \neg(\neg\psi_1 \wedge \neg\psi_2)$: Logical OR
- $\text{not}(\psi) = 1 - \psi$: Logical NOT
- $\text{xor}(\psi_1, \psi_2)$: Logical XOR

*Comparison Operators* For $\psi_1, \psi_2 \in \mathcal{H}(\mathbb{Z})$:

- $\text{leq}(\psi_1, \psi_2) = [\psi_2 - \psi_1 + 1]_+ - [\psi_2 - \psi_1]_+$: Less than or equal
- $\text{geq}(\psi_1, \psi_2) = \text{leq}(\psi_2, \psi_1)$: Greater than or equal
- $\text{eq}(\psi_1, \psi_2) = \text{leq}(\psi_1, \psi_2) \wedge \text{leq}(\psi_2, \psi_1)$: Equality
- $\text{lt}(\psi_1, \psi_2) = \text{leq}(\psi_1, \psi_2 - 1)$: Less than
- $\text{gt}(\psi_1, \psi_2) = \text{lt}(\psi_2, \psi_1)$: Greater than

*Sequence Aggregation Operators*

- $\text{seq\_max}(\psi)$: Returns the maximum value across all positions in the sequence
- $\text{seq\_min}(\psi)$: Returns the minimum value across all positions in the sequence
- $\text{seq\_and}(\psi) = \text{seq\_min}(\psi)$: Logical AND across all positions
- $\text{seq\_or}(\psi) = \text{seq\_max}(\psi)$: Logical OR across all positions
- $\text{seq\_sum}(\psi)$: Sum of values across all positions (requires $\log n$ positional embedding)
- $\text{seq\_avg}(\psi) = \frac{1}{n} \sum_{j=1}^{n} \psi(x_{1:j})$: Average across all positions

*Positional Operators*

- $\text{is\_first}(i) = \mathbf{1}[i = 1]$: Indicator for first position
- $\text{inv\_seq\_len}(i) = 1/n$: Inverse of sequence length
- $\text{is\_pos\_k}(i) = \mathbf{1}[i = k]$: Indicator for position $k$

*Control Flow Operators*

- `if_then_else`($\psi_{\text{cond}}, \psi_{\text{true}}, \psi_{\text{false}}$) or `ite`($\psi_{\text{cond}}, \psi_{\text{true}}, \psi_{\text{false}}$): If-then-else conditional selection

*Attention Variants*

- `aha`($q, k, v$): Standard average-hard attention (encoder bidirectional)
- `rha`($q, k, v$): Rightmost-hard attention (breaks ties by selecting rightmost position)
- `rightmost_exact_match`($q, k, v$): Rightmost exact match (returns default if no exact match)

### G.5 ADDITIONAL OPERATORS AND JUSTIFICATIONS

Next we give the semantics of some additional operators used in the PRAM simulation programs and justify their closure in the `E-FASP` framework.

#### BITWISE ARITHMETIC OPERATORS

These operators are defined in the encoder-`E-FASP` framework using activations $\{[\cdot]_+, \times\}$ (equivalently ReGLU) and are independent of the specific positional embedding choice. All inputs and outputs are position-indexed seq-to-embedding functions in $\mathcal{H}(\{0,1\}^m)$ where $\psi(\mathbf{x}, i) \in \{0,1\}^m$ encodes an $m$-bit integer with LSB at coordinate 1. All arithmetic is modulo $2^m$.

**Bitwise Addition.** Given $\psi_1, \psi_2 \in \mathcal{H}(\{0,1\}^m)$, write at position $(\mathbf{x}, i)$:

$$\psi_1(\mathbf{x}, i) =: \mathbf{a} = (a_1, \ldots, a_m), \quad \psi_2(\mathbf{x}, i) =: \mathbf{b} = (b_1, \ldots, b_m) \in \{0,1\}^m \tag{46}$$

Bitwise addition is defined as adding two $m$-bit integers modulo $2^m$. This can be constructed using the primitive operators (and other operators that are already defined) in Definition 8, which follows an approach similar to standard carry-lookahead, and is a constant-depth, polylogarithmic-width construction:

Define for $k \in [m]$ the local propagate/generate bits:

$$p_k = a_k \oplus b_k = a_k + b_k - 2a_k b_k, \quad g_k = a_k \wedge b_k = a_k b_k \tag{47}$$

Let $S_0 = 0$ and $S_j = \sum_{t \leq j} p_t$ for $j \in [m]$ (computed by a single linear layer). For $1 \leq j < i \leq m$, define the interval-all-ones gate:

$$Q_{j,i} = \text{eq}_0\left((S_{i-1} - S_j) - ((i-1) - j)\right) \tag{48}$$

where $\text{eq}_k(u) := 2\left([u - (k - \frac{1}{2})]_+ - 2[u - k]_+ + [u - (k + \frac{1}{2})]_+\right)$ equals 1 at $u = k$ and 0 at all other integers.

The carry into bit $i$ is:

$$C_i = \begin{cases} 0, & i = 1 \\ 1 - \text{eq}_0\left(\sum_{j=1}^{i-1} g_j Q_{j,i}\right), & i \geq 2 \end{cases} \tag{49}$$

The sum bits are $s_i = p_i \oplus C_i = p_i + C_i - 2p_i C_i$. We define:

$$\texttt{bit\_add}_m(\psi_1, \psi_2)(\mathbf{x}, i) := \mathbf{s} = (s_1, \ldots, s_m) \in \{0,1\}^m \tag{50}$$

**Bitwise Subtraction.** For $\psi_1, \psi_2 \in \mathcal{H}(\{0,1\}^m)$, define:

$$\texttt{bit\_minus}_m(\psi_1, \psi_2) := \texttt{bit\_add}_m(\psi_1, \neg\psi_2) \dotplus 1 \tag{51}$$

where $\neg$ is bitwise NOT (elementwise $1 - \cdot$) and "$\dotplus 1$" adds the constant vector $\mathbf{e}_1 = (1, 0, \ldots, 0)$ via the same `bit_add`$_m$.

**Logical Shifts.** Let $\psi \in \mathcal{H}(\{0,1\}^m)$ and $\tau \in \mathcal{H}(\{0,1\}^m)$. At position $(\mathbf{x}, i)$, write $\mathbf{a} = \psi(\mathbf{x}, i) = (a_1, \ldots, a_m)$ and define the shift amount:

$$t = \text{int}(\tau) = \sum_{r=1}^{m} 2^{r-1}\tau_r \in \{0, \ldots, m\} \tag{52}$$

For $k \in [m]$, we define:

$$[\texttt{shift\_left}_m(\psi, \tau)]_k = \sum_{s=0}^{\min\{m, k-1\}} \text{eq}_s(t) \cdot a_{k-s} \tag{53}$$

$$[\texttt{shift\_right}_m(\psi, \tau)]_k = \sum_{s=0}^{\min\{m, m-k\}} \text{eq}_s(t) \cdot a_{k+s} \tag{54}$$

where out-of-range indices are treated as 0, and $\text{eq}_s(\cdot)$ is the integer-equality gate realized by three ReLUs.

**Complexity Analysis.** All operators act locally at each position on $\mathcal{H}(\{0,1\}^m)$ without cross-position communication, and are composed from E-FASP primitives. Throughout $m = \Theta(\log n)$.

The witness enumeration method for bitwise addition requires: **(i)** one linear layer for $(p, g)$ and prefix sums $(S_j)$; **(ii)** one nonlinear layer for witnesses $Q_{j,i}$ (each uses 3 ReLUs) and products $g_j Q_{j,i}$; **(iii)** linear aggregation and threshold for carries $C_i$; **(iv)** local polynomial for $s_i = p_i \oplus C_i$. This achieves constant depth (3-4 layers) and width $\mathcal{O}(m^2) = \mathcal{O}((\log n)^2)$ (polylogarithmic in $n$). Bitwise subtraction uses two's complement and reuses the same addition circuit with identical complexity bounds. Logical shifts compute the shift amount $t$ and all candidate shifts in parallel, then use equality gates for selection, also achieving constant depth and $\mathcal{O}(m^2)$ width.

All constructions use only E-FASP primitives (linear projections, ReLU/ReGLU activations, multiplication). By the equivalence established in § G, these are realizable by constant-depth encoder Transformers.

**Instruction Access Operations.** This operator enables instruction fetching from memory by address lookup, which is essential for PRAM simulation.

$$\text{get\_instruction}(\mathbf{x}, i) := \text{bin}(W_{\text{INSTR}} \circ \text{ite}(\text{is\_addr}(\mathbf{x}, \cdot), \text{bin}(\text{TE}(\mathbf{x}, \cdot)), \mathbf{0}_w))(\mathbf{x}, i) \in \{0, 1\}^w \tag{55}$$

where $W_{\text{INSTR}} : \mathbb{R}^w \to \mathbb{R}^w$ is a learned linear transformation (MLP layer) that maps address bits to instruction bits. This operator first extracts the address bits from address positions (even positions ¿ 1) by converting their token embeddings to binary representations, then applies the instruction lookup transformation $W_{\text{INSTR}}$ to produce the corresponding instruction encoding. The PRAM instruction set (LOAD, STORE, ADD, SUB, etc., as defined in § E) is hardcoded into the parameters of $W_{\text{INSTR}}$ during training, enabling the model to perform instruction fetching through learned address-to-instruction mappings.

# H    PROOF OF THEOREM 1

**Theorem 8** (MDM Simulation of PRAM, Formal). *For any PRAM program $\mathcal{P} = (I_0, \ldots, I_\ell, \text{addr})$ (with finite number of instructions $\ell$, and is uniform for all processors and input size $n$), that on input $\mathbf{x}_{val} \in \mathbb{U}^n$ with corresponding address $\mathbf{x}_{addr} \in \mathcal{A}^n$ that runs in $T(n)$ parallel time using at most $P(n)$ processors and outputs $PRAM_\mathcal{P}(\mathbf{x}_{addr}, \mathbf{x}_{val}) \in \mathbb{U}$ per procedure described in § E, there exists a bijection $\phi : \mathbb{U} \cup \mathcal{A} \to \Sigma$ and a special token $[SEP] \in \Sigma$, and a MDM with constant depth and $\log(n)$ embedding size encoder-only Transformer, on input $\mathbf{x} = ((\mathbf{z}_{2i}, \mathbf{z}_{2i+1})_{i=0}^{n-1}, [SEP]) \in \Sigma^{2n+1}$ where $\mathbf{z}_{2i} = \phi(\mathbf{x}_{addr,i})$ and $\mathbf{z}_{2i+1} = \phi(\mathbf{x}_{val,i})$, padded to $\mathcal{O}(P(n) \times T(n))$ context length, outputs $\phi(PRAM_\mathcal{P}(\mathbf{x}_{addr}, \mathbf{x}_{val}))$ with $\mathcal{O}(T(n))$ decoding steps.*

The proof demonstrates that AO-MDM can simulate any PRAM algorithm. The construction is based on E-FASP, the programming language we developed, whose definable programs are equivalent to

encoder-only Transformer function class (see § G). We prove Theorem 1 by: **(1)** defining the setup and input format for PRAM simulation; **(2)** constructing an `E-FASP` program that simulates PRAM execution in Algorithm 2.

**Choice of Architecture / `E-FASP` Variant**    For this simulation, we use the $\text{E-FASP}[\text{SEQ}; [\cdot]_+, \times]$ variant with integer positional encoding rather than the binary variant. This choice is crucial because MDM's context length can be exponentially large (e.g., for NP-hard problems), while PRAM's actual memory usage remains polynomial. Using $\log n$ bits to represent positions would be insufficient when the context length $n$ itself grows exponentially with the problem size, even though PRAM addresses can still be represented in $\log S(n)$ bits. To avoid confusion between MDM's context length and PRAM's memory space, we use $S_{\text{MDM}}(n)$ to denote the maximum context length and reserve $S(n)$ for PRAM's memory space.

**Input Format**    The input that encodes the PRAM's initial memory state is a sequence $\mathbf{x} = (x_1, \ldots, x_{2n+2}) \in \Sigma^{2n+2}$ of discrete tokens from the vocabulary $\Sigma$.

Let $w = \Theta(\log n)$ be the word width and recall from § E that the address width $a = \lceil \log_2 S(n) \rceil \leq w$ (addresses fit within words). The sequence has length $2n + 2 = 1 + 2n + 1$ where:

$$x_1 \in \Sigma \quad \text{(processor count token)} \tag{56}$$
$$x_{2i} \in \Sigma \quad \text{(address token)} \quad \text{for } i = 1, \ldots, n \tag{57}$$
$$x_{2i+1} \in \Sigma \quad \text{(data token)} \quad \text{for } i = 1, \ldots, n \tag{58}$$
$$x_{2n+2} = [\text{SEP}] \quad \text{(separator token)} \tag{59}$$

Through token embedding $\text{TE} : \Sigma \to \mathbb{R}^d$ and subsequent linear projections, these discrete tokens are mapped to their semantic bit representations:

$$\text{TE}(x_1) \mapsto P(n) \in \{0, 1\}^w \tag{60}$$
$$\text{TE}(x_{2i}) \mapsto \text{addr}_i \in \{0, 1\}^w \quad \text{(address bits)} \tag{61}$$
$$\text{TE}(x_{2i+1}) \mapsto \text{val}_i \in \{0, 1\}^w \quad \text{(data bits)} \tag{62}$$

for $i = 1, \ldots, n$. The $[\text{SEP}]$ token serves as a separator with special meaning in the computation trace, as detailed in the next subsection. Following the standard MDM notation from § 2, the actual input to the MDM is the padded sequence $\mathbf{x}_0 = (x_{0,1}, x_{0,2}, \ldots, x_{0,S_{\text{MDM}}(n)}) \in \bar{\Sigma}^{S_{\text{MDM}}(n)}$ where $\bar{\Sigma} = \Sigma \cup \{\mathsf{M}\}$:

$$x_{0,j} = x_j \quad \text{for } j = 1, \ldots, 2n + 2 \tag{63}$$
$$x_{0,j} = \mathsf{M} \quad \text{for } j = 2n + 3, \ldots, S_{\text{MDM}}(n) \tag{64}$$

We aim to show that there exists an encoder-only Transformer that, given the initial memory state of a PRAM as input, can output the exact same result as the PRAM algorithm after $T_{\text{MDM}}(n) = \mathcal{O}(T(n))$ decoding steps. The Transformer will have constant depth and context length $\mathcal{O}(S_{\text{MDM}}(n))$, where $S_{\text{MDM}}(n)$ represents the MDM's context budget.

We next provide an overview of the construction:

**Processor State and Computation Log Representation:**    First, we state the representation of processor state and computation log as tokens. We represent each processor's state and one round of computation using a fixed number of tokens: program counter (1 token), register file (5 tokens), and computation log (2 tokens), for a total of 8 tokens per processor.

*Program Counter (PC):* A single word encoding the current instruction address.

*Register File:* We maintain exactly 5 registers, each storing one word. This provides sufficient computational capacity while keeping the representation tractable.

*Computation Log:* This log is populated only when executing `STORE [s],r` instructions, recording the target address and stored value. For all other instructions, the log remains empty (represented by special tokens).

The computation trace for one parallel round can be represented as:

$$[\text{SEP}] \; \langle \text{PC}_1, \text{R}_{1,1}, \ldots, \text{R}_{1,5}, \text{Addr}_1, \text{Val}_1 \rangle \; \langle \text{PC}_2, \text{R}_{2,1}, \ldots, \text{R}_{2,5}, \text{Addr}_2, \text{Val}_2 \rangle \; \ldots \quad (65)$$

where $[\text{SEP}]$ serves as a separator token to distinguish different computation rounds and there are a total of $P(n)$ independent processors. Each processor $i \in \{1, \ldots, P(n)\}$ contributes an 8-tuple $\langle \text{PC}_i, \text{R}_{i,1}, \text{R}_{i,2}, \text{R}_{i,3}, \text{R}_{i,4}, \text{R}_{i,5}, \text{Addr}_i, \text{Val}_i \rangle$ representing its program counter, five register values, and memory write operation (address and value). The trace thus contains exactly $P(n)$ such 8-tuples per parallel round.

**Processor Assignment and Role Identification.** The algorithm begins by determining whether the current position contains a mask token (i.e. is_mask). If the position is a mask token (**Branch 1**), the algorithm continues by computing the distance to the nearest preceding $[\text{SEP}]$ token. If the position is not a mask token (**Branch 2**), it indicates this position has already been unmasked (computation has already finished), and the algorithm returns the input token (or an all-zero vector) which will not be unmasked per definition of MDM § 2.

To identify the processor ID, we find the rightmost $[\text{SEP}]$ token to the left of the current position (i.e. rightmost_sep_pos) and compute the distance between them (i.e. distance_to_sep). If this distance $> 8 \times P(n)$ (**Branch 1.1**), the position does not participate in the current computation round as it is not a token that should be "unmasked" in this round, in this case, the algorithm returns a special embedding (an all-zero vector), which results in uniform distribution during prediction and smallest confidence, ensuring that the MDM will not select this position for unmasking; if the distance $= 8 \times P(n) + 1$ (**Branch 1.2**), this position return the embedding of $[\text{SEP}]$, preparing for the next computation round; otherwise if the distance $< 8 \times P(n) + 1$ (**Branch 1.3**), the position participates in the computation of the current round.

For those positions participating in the current computation round. The corresponding processor ID (i.e. processor_id) is obtained by right-shifting distance_to_sep by 3 bits with zero-padding on the left (since each processor corresponds to exactly 8 tokens). The rightmost 3 bits represent the position within that processor (i.e. inner_processor_id).

**Initialization of Processor State.** We need to initialize the initial state of all processors at the beginning. This is determined by the current number of $[\text{SEP}]$ tokens in the sequence. Specifically, when the current position is a mask token and there is exactly one $[\text{SEP}]$ token (i.e. seq_sum(is_sep) $==$ 1) (**Branch 3**), we consider this the initialization state. All program counters are set to 0, all registers are set to 0, and all memory locations are set to 0.

**Fetch Instruction and Execution (Main Loop).** According to the processor ID and inner processor position, the algorithm fetches the instruction from the instruction memory (i.e. get_instruction), which is hard-coded into the parameters of the model, and execute it (using execute). Different instructions yields different execution semantics, and subsequently different 8 token state. Finally, according to the inner_processor_id, the algorithm chooses what to return. The algorithm terminates when the PC of all processors are HALT.

We now formally construct an E-FASP program that simulates PRAM execution (the single-processor algorithm detailed in § E). The semantics meanings and justifications of operators used in the program are summarized in § G. The only two global operators are seq_sum and rightmost_exact_match implementable by attention , otherwise are all local operators implementable by polylog-width constant-depth MLPs.

The context length of MDM used by this construction is $S_{\text{MDM}}(n) = \mathcal{O}(T_{\text{par}}(n) \times P(n))$, the decoding steps is $T_{\text{MDM}}(n) = \mathcal{O}(T_{\text{par}}(n))$, where $T_{\text{par}}(n)$ is the parallel time complexity and $P(n)$ is the processor count. For the constructed Transformer, embedding size is $\log(n)$ and depth is a constant.

```
# --------------------- Initialization ---------------------
is_sep = (TE == embed([SEP]))
is_mask = (TE == embed([MASK]))
is_init = (seq_sum(is_sep) == 1)

# Get the current and last [SEP] position
```

```
cur_sep = rightmost_exact_match(1, is_sep, PE)
dist_to_sep = PE- cur_sep
pn = rightmost_exact_match(1, is_first, TE)
spanned_pn = pn << 3

# Skip positions not participating in computation
if (not is_mask) or (dist_to_sep > spanned_pn): return 0
if dist_to_sep == spanned_pn + 1: return embed([SEP])
if is_init: return 0

# Initialization
pid = (dist_to_sep - 1) >> 3
inner_id = (dist_to_sep - 1 )[:3]

if is_init and inner_id == 1: return pid
if is_init and inner_id != 1: return 0

# --------------------- Read previous round state ---------------------
prev_sep = rightmost_exact_match(1, (is_sep and (PE < cur_sep)), PE)
prev_pid_base = prev_sep + 1 + (pid << 3)

pos_PC = prev_pid_base + 0
PC = rightmost_exact_match(pos_PC, PE, TE)
pos_R1 = prev_pid_base + 1
R1 = rightmost_exact_match(pos_R1, PE, TE)
pos_R2 = prev_pid_base + 2
R2 = rightmost_exact_match(pos_R2, PE, TE)
pos_R3 = prev_pid_base + 3
R3 = rightmost_exact_match(pos_R3, PE, TE)
pos_R4 = prev_pid_base + 4
R4 = rightmost_exact_match(pos_R4, PE, TE)
pos_R5 = prev_pid_base + 5
R5 = rightmost_exact_match(pos_R5, PE, TE)

if PC == HALT_CODE: return 0

# -------------------- Fetch and execute instruction --------------------
# Decode
I_type, op_r, op_s, op_c, label_addr = get_instruction(PC)

# Source/destination register
Rs = (R1 if op_s == 1 else
      R2 if op_s == 2 else
      R3 if op_s == 3 else
      R4 if op_s == 4 else
      R5 if op_s == 5 else 0)

Rr = (R1 if op_r == 1 else
      R2 if op_r == 2 else
      R3 if op_r == 3 else
      R4 if op_r == 4 else
      R5 if op_r == 5 else 0)

# Default effect
PC_next = PC + 1
WR_val = Rr
writes_reg = False
ADDR_out = 0 # Slot 6: only STORE overwrites address (a <= w, use word
    directly)
VAL_out = 0 # Slot 7: only STORE overwrites value

# Address read
if I_type == embed([LOAD]):
    ADDR_KEYS = (TE if is_addr else 0)
```

```
    ADDR_POSVAL = (PE if is_addr else 0)

    last_addr_pos_load = rightmost_exact_match(Rs, ADDR_KEYS, ADDR_POSVAL)
    load_val = rightmost_exact_match(last_addr_pos_load + 1, PE, TE)
    WR_val = load_val

# Per-instruction semantics
elif I_type == embed([STORE]): ADDR_out, VAL_out = Rs, Rr
elif I_type == embed([LOADI]): WR_val = op_c
elif I_type == embed([ADD]): WR_val = (Rr + Rs)
elif I_type == embed([SUB]): WR_val = (Rr - Rs)
elif I_type == embed([AND]): WR_val = (Rr & Rs)
elif I_type == embed([XOR]): WR_val = (Rr ^ Rs)
elif I_type == embed([SHL]): WR_val = (Rr << Rs)
elif I_type == embed([SHR]): WR_val = (Rr >> Rs)
elif I_type == embed([BRZ]) and Rr == 0: PC_next = label_addr
elif I_type == embed([JMP]): PC_next = label_addr
elif I_type == embed([HALT]): PC_next = HALT_CODE

# Register writeback: only for {LOAD, LOADI, ADD, SUB, AND, XOR, SHL, SHR
    }
writes_reg = (I_type == embed([LOAD])) or (I_type == embed([LOADI])) or \
             (I_type == embed([ADD])) or (I_type == embed([SUB])) or \
             (I_type == embed([AND])) or (I_type == embed([XOR])) or \
             (I_type == embed([SHL])) or (I_type == embed([SHR]))

R1_next = (WR_val if (writes_reg and op_r == 1) else R1)
R2_next = (WR_val if (writes_reg and op_r == 2) else R2)
R3_next = (WR_val if (writes_reg and op_r == 3) else R3)
R4_next = (WR_val if (writes_reg and op_r == 4) else R4)
R5_next = (WR_val if (writes_reg and op_r == 5) else R5)

# ---------- Return one of 8 slots according to inner_id ----------
if inner_id == 0: return PC_next
elif inner_id == 1: return R1_next
elif inner_id == 2: return R2_next
elif inner_id == 3: return R3_next
elif inner_id == 4: return R4_next
elif inner_id == 5: return R5_next
elif inner_id == 6: return ADDR_out
else: return VAL_out
```

# I  PROOF OF THEOREM 2

The result stems from MDM's total amount of computation being bounded by $S(n)$ in both total steps ($T(n) \leq S(n)$) and per-step capacity (polynomial embedding size), preventing it from solving problems requiring greater computational resources.

Fix an encoder-only MDM with context length $S(n)$ and $T(n)$ decoding steps. Throughout we assume constant depth/heads and log-precision arithmetic with hidden width $d = \Theta(\log(S(n) + T(n)))$ (binary positional code), as in our setup. Particularly, at each decoding step the model re-encodes a length-$S(n)$ sequence. A single forward pass is dominated by self-attention: for each position $i$ we form a query in $\mathbb{R}^d$ and take dot products with all $S(n)$ keys, then take the value-weighted sum. Counting FLOPs, one attention head costs

$$\Theta\big(S(n) \cdot S(n) \cdot d\big) = \Theta\big(S(n)^2 \log(S(n) + T(n))\big), \tag{66}$$

and the multi-head/multi-layer constants only change the leading constant. The position-wise MLP adds $\Theta(S(n) \cdot \text{poly}(d)) = \Theta\big(S(n) \text{polylog}(S(n) + T(n))\big)$ FLOPs and is lower order when $S(n) \gg d$. Thus one decoding step costs

$$\widetilde{\mathcal{O}}\big(S(n)^2\big) \quad \text{FLOPs}, \tag{67}$$

where $\widetilde{\mathcal{O}}(\cdot)$ suppresses polylog factors in $S(n) + T(n)$. Over $T(n)$ steps the total compute is $\widetilde{\mathcal{O}}(S(n)^2 T(n))$. In particular, when each step reveals at least one token (or a constant number), we have $T(n) \le S(n)$, yielding the unified cubic bound $\widetilde{\mathcal{O}}(S(n)^3)$. Hence any problem that needs $\omega(S(n)^3)$ serial time cannot be solved by MDM in the $(S(n), T(n))$ regime stated.

## J    PROOF OF THEOREM 3

Recall Definition 3, we defined **Masked-ARM** as an autoregressive model with encoder-only Transformer architecture that pads the input sequence with mask tokens to the maximum context length, which is also equivalent to a MDM with a fixed order (left-to-right) generation and generating one token at a time. Consider an AO-MDM with input format $\mathbf{x} = (x_1, \ldots, x_n)$ followed by a special separator token [SEP] at position $n + 1$.

**AO-MDM Intermediate State:** At any intermediate generation step, the AO-MDM state can be represented as $\mathbf{z} = (z_1, \ldots, z_{S(n)}) \in \bar{\Sigma}^{S(n)}$ where $\bar{\Sigma} = \Sigma \cup \{\mathsf{M}\}$. The sequence structure is:

$$z_j = x_j \quad \text{for } j \in [n] \quad \text{(fixed input portion)} \tag{68}$$
$$z_{n+1} = [\text{SEP}] \quad \text{(separator)} \tag{69}$$
$$z_j \in \Sigma \cup \{\mathsf{M}\} \quad \text{for } j \in \{n+2, \ldots, S(n)\} \quad \text{(generation region)} \tag{70}$$

Let $\mathcal{D} = (d_1, d_2, \ldots, d_k)$ denote the sequence of positions that have been decoded (unmasked) by the AO-MDM in chronological order, where $d_i \in \{n+2, \ldots, S(n)\}$ and $z_{d_i} \ne \mathsf{M}$ for all $i \in [k]$. The ordering reflects the temporal sequence in which the AO-MDM performed the unmasking operations.

**Definition 10** (Position/Content Tokens and Address Encoding). *Let $\Sigma$ be the base vocabulary. We reserve a subset $\Sigma_{\text{pos}} \subseteq \Sigma$ for position tokens and define a bijection $\text{encode} : \{1, \ldots, S(n)\} \to \Sigma_{\text{pos}}$ with inverse $\text{dec\_pos} : \Sigma_{\text{pos}} \to \{1, \ldots, S(n)\}$. For each decoded position $d_i$, define*

$$addr_{d_i} := \text{encode}(d_i) \in \Sigma_{\text{pos}} \subseteq \Sigma, \qquad tok_{d_i} := z_{d_i} \in \Sigma.$$

*Thus both address tokens and content tokens are drawn from the original vocabulary $\Sigma$. We also reserve a subset $\Sigma_{\text{op}} \subseteq \Sigma$ for operator tokens used later for AP-MDM edits (§ 4).*

**Masked-ARM Simulation:** For each decoded token at position $d_i \in \mathcal{D}$, the Masked-ARM represents it using a 2-tuple:

$$\langle addr_{d_i}, tok_{d_i} \rangle = \langle \text{encode}(d_i), z_{d_i} \rangle \tag{71}$$

where $\text{encode}(d_i)$ is a token representation of the positional index $d_i$, and $z_{d_i}$ is the actual decoded token. The target Masked-ARM sequence to be constructed is:

$$\mathbf{y}_{\text{ARM}} = (x_1, \ldots, x_n, [\text{SEP}], addr_{d_1}, tok_{d_1}, \ldots, addr_{d_k}, tok_{d_k}, \underbrace{\mathsf{M}, \ldots, \mathsf{M}}_{\text{remaining positions}}) \tag{72}$$

where the sequence $\mathcal{D} = (d_1, d_2, \ldots, d_k)$ preserves the chronological order of AO-MDM's decoding operations. The Masked-ARM sequence has total length $2S(n) - n - 1 = \mathcal{O}(S(n))$, since each AO-MDM token requires two tokens (address and content) in the Masked-ARM representation.

**Induction:** To prove We prove by induction that for any AO-MDM, there exists a corresponding Masked-ARM that can simulate the AO-MDM's generation process step by step for arbitrary input sequences.

**Theorem 9** (AO-MDM Simulation by Masked-ARM). *For any AO-MDM, there exists a corresponding Masked-ARM such that: for any input sequence $\mathbf{x} = (x_1, \ldots, x_n)$ and any intermediate state of the AO-MDM with decoded sequence $\mathcal{D} = (d_1, d_2, \ldots, d_k)$, the Masked-ARM, starting from the corresponding intermediate state $\mathbf{y}_{\text{ARM}}$, can generate the next address-token pair $\langle addr_{d_{k+1}}, tok_{d_{k+1}} \rangle$ such that:*

$$\text{encode}(d_{k+1}) = addr_{d_{k+1}} \quad \text{(address matches AO-MDM's next decode position)} \tag{73}$$
$$z_{d_{k+1}} = tok_{d_{k+1}} \quad \text{(token matches AO-MDM's next decode content)} \tag{74}$$

*where $d_{k+1}$ is the position that AO-MDM will decode next, and $z_{d_{k+1}}$ is the token that AO-MDM will generate at that position.*

*Proof.* To prove this result, we decompose the architecture of the AO-MDM into two parts: the input transformation part (which can be represented as an operator `mdm_embed`) that transforms the token and position into an embedding, and the output generation part (which can be represented as an operator `mdm_decode`) that transforms the embedding into logits, that is:

$$\text{AO-MDM}(\mathbf{x}) = \text{mdm\_decode}(\text{mdm\_embed}(\bar{\text{TE}}, \bar{\text{PE}}))(\mathbf{x}) \qquad (75)$$

where $\bar{\text{TE}}$ and $\bar{\text{PE}}$ are seq-to-seq functions defined in Definition 5.

This decomposition is invariant to the choice of token and position embedding functions and AO-MDM's parameter configuration. Simulating AO-MDM's generation process boils down to the following two steps:

**Step 1: Replicating `mdm_embed`.** We construct initial layers of the Masked-ARM that, given the Masked-ARM state $\mathbf{y}_{\text{ARM}}$, produce intermediate embeddings identical to $\text{mdm\_embed}(\mathbf{x})$ where $\mathbf{x}$ is the corresponding AO-MDM state. This transformation converts the address-token pair representation back into the embedding format that the AO-MDM expects, enabling the subsequent layers to perform identical computations. We write the `E-FASP` programs (which corresponds to the encoder Transformer construction) for the construction:

```
# --------------------- Get logits identical to AO-MDM
    ---------------------
mdm_logits = mdm_decode(embed_MDM)
tok_scores = score(mdm_logits)

# AO-MDM candidate set: positions > [SEP], still [MASK], and within valid
    range
cand_mask = (PE > sep_pos) and (TE_MDM == embed([MASK])) and (PE <= sn)
cand_score = (tok_scores if cand_mask else 0)

max_score = seq_max(cand_score)
is_best = cand_mask and (tok_scores == max_score)

# Select AO-MDM's next decode position and corresponding logits
next_pos = rightmost_exact_match(1, is_best, PE)
logits_next = rightmost_exact_match(next_pos, PE, mdm_logits)

# --------------------- Emit as Masked-ARM <addr, tok> order
    ---------------------
gen_slot = rightmost_exact_match(1,
                        (PE > sep_pos) and (PE <= sn) and (TE == embed
                            ([MASK])),
                        PE)
emit_addr = (((gen_slot - sep_pos)[:1]) == 0)

if PE== gen_slot:
   result = (next_pos if emit_addr else logits_next)
else:
   result = 0

return result
```

This completes the proof. □

We remark the proof relies on two assumptions: 1) the function $\bar{S}(\mathbf{x}) = S(|\mathbf{x}|)$ is deterministic and computable by encoder Transformer (this is implemented by the `sn` function in the `E-FASP` program for Step 1); 2) the confidence score is also deterministic and computable by encoder Transformer (this is implemented by the `score` operator in the `E-FASP` program for Step 2).

## K    PROOF OF THEOREM 4

**Theorem 10** (AP-MDM Simulation of PRAM, Formal)**.** *Let* $\mathcal{P} = (I_0, \ldots, I_\ell, \text{addr})$ *be a uniform PRAM program with a finite instruction set of size* $\ell$, *identical across processors and input size*

$n$. *On an initial memory state specified by address–value pairs $(\mathbf{x}_{\text{addr}}, \mathbf{x}_{\text{val}})$ with $\mathbf{x}_{\text{val}} \in \mathbb{U}^n$ and $\mathbf{x}_{\text{addr}} \in \mathcal{A}^n$, suppose $\mathcal{P}$ runs in parallel time $T(n)$ using at most $P(n)$ processors and at most $S(n)$ shared-memory words of $\Theta(\log n)$ bits, and outputs $\text{PRAM}_{\mathcal{P}}(\mathbf{x}_{\text{addr}}, \mathbf{x}_{\text{val}}) \in \mathbb{U}$ (see § E). Then there exists a bijection $\phi : \mathbb{U} \cup \mathcal{A} \to \Sigma$ and an AP-MDM which, on input*

$$\mathbf{x} = (z_0, z_1, \ldots, z_n) \in \Sigma^{n+1}, \quad z_0 = \phi(P(n)), \; z_i = \phi(\mathbf{x}_{\text{val},i}) \, \text{for } i = 1, \ldots, n,$$

*padded to context length $\mathcal{O}(S(n))$ (addresses provided implicitly by positional encodings), produces $\phi(\text{PRAM}_{\mathcal{P}}(\mathbf{x}_{\text{addr}}, \mathbf{x}_{\text{val}}))$ in $\mathcal{O}(T(n))$ decoding steps.*

We first show that AP-MDM can simulate a weaker model called Rewrite-MDM, which is sufficient for the result. Then we construct an `E-FASP` program that simulates PRAM execution in a space-efficient manner.

**Rewrite-MDM** follows the same framework as AP-MDM but with simplified control signals. For any token $y \in \Sigma$, define:

$$\textbf{\textcolor{blue}{remask}}_{x_{t,i}, c_{t,i}}(y) = \begin{cases} y & \text{if } c_{t,i}[1] = 1 \\ x_{t,i} & \text{if } c_{t,i}[1] = 0 \end{cases} \tag{76}$$

where $c_{t,i}[1] \in \{0, 1\}$ is a binary rewrite signal. In other words, when $c_{t,i}[1] = 1$, the model rewrites position $i$ with new content $y$; when $c_{t,i}[1] = 0$, it preserves the original content $x_{t,i}$ unchanged.

We next show how each transition $\mathbf{z}_t \to \mathbf{z}_{t+1}$ in Rewrite-MDM can be simulated by exactly three steps of AP-MDM as defined in § 4.

**Lemma 11** (AP-MDM Simulation of Rewrite-MDM). *For any Rewrite-MDM transition $\mathbf{z}_t \to \mathbf{z}_{t+1}$ on sequence of length $n$, there exists a sequence of three AP-MDM steps that produces the identical result.*

*Proof.* Given a Rewrite-MDM transition where we want to selectively rewrite positions in sequence $\mathbf{z}_t = (z_{t,1}, z_{t,2}, \ldots, z_{t,n})$ according to rewrite signal $\mathbf{r}_t = (r_{t,1}, r_{t,2}, \ldots, r_{t,n})$, we simulate this using the following three AP-MDM steps: $\mathbf{z}_t \to \mathbf{u}^{(1)} \to \mathbf{u}^{(2)} \to \mathbf{u}^{(3)} = \mathbf{z}_{t+1}$.

**Step 1 (Insert):** Starting from $\mathbf{z}_t$, apply **\textcolor{orange}{insert}** operation at every position $i \in [n]$ to create an expanded sequence of length $2n$:

$$\mathbf{u}^{(1)} = (g \circ f_\theta)(\mathbf{z}_t) \tag{77}$$

where $c_i^{(1)}[1] = 0$ (no remask), $c_i^{(1)}[2] = 1$ (insert), $c_i^{(1)}[3] = 0$ (no delete) for all $i \in [n]$. This yields $\mathbf{u}^{(1)} = (z_{t,1}, \mathsf{M}, z_{t,2}, \mathsf{M}, \ldots, z_{t,n}, \mathsf{M})$.

**Step 2 (Unmask and Remask):** Apply AP-MDM's $(g \circ f_\theta)$ operation on $\mathbf{u}^{(1)}$ with control signals:

$$\mathbf{u}^{(2)} = (g \circ f_\theta)(\mathbf{u}^{(1)}) \tag{78}$$

where the control signals are set as follows:

- For even positions $2i$ (newly inserted masks): $c_{2i}^{(2)}[1] = 0$ (unmask), $c_{2i}^{(2)}[2] = 0$ (no insert), $c_{2i}^{(2)}[3] = 0$ (no delete)

- For odd positions $2i - 1$ (original tokens): $c_{2i-1}^{(2)}[1] = r_{t,i}$ (remask), $c_{2i-1}^{(2)}[2] = 0$ (no insert), $c_{2i-1}^{(2)}[3] = 0$ (no delete)

which yield $\mathbf{u}^{(2)} = (\mathsf{M}, z_{t+1,1}, \mathsf{M}, z_{t+1,2}, \ldots, \mathsf{M}, z_{t+1,n})$.

**Step 3 (Delete):** Apply AP-MDM's $(g \circ f_\theta)$ operation again to **\textcolor{red}{delete}** all mask tokens at original positions:

$$\mathbf{u}^{(3)} = (g \circ f_\theta)(\mathbf{u}^{(2)}) \tag{79}$$

where for all positions $j$ in $\mathbf{u}^{(2)}$:

- For odd positions $2i-1$: $c_{2i-1}^{(3)}[1] = 0$ (no remask), $c_{2i-1}^{(3)}[2] = 0$ (no insert), $c_{2i-1}^{(3)}[3] = \mathbb{1}[u_{2i-1}^{(2)} = \mathsf{M}]$ (delete if mask)

- For even positions $2i$: $c_{2i}^{(3)}[1] = 0$ (no remask), $c_{2i}^{(3)}[2] = 0$ (no insert), $c_{2i}^{(3)}[3] = 0$ (no delete)

This removes all mask tokens at odd positions and recovers the original length $n$. By construction, $\mathbf{u}^{(3)} = \mathbf{z}_{t+1}$, completing the simulation.

**State Tracking Mechanism**    To enable the AP-MDM to autonomously determine which of the three simulation steps to execute, we augment sequences with special boundary tokens [BOS] and [EOS]. The model identifies the current phase by examining the boundary token configuration:

- **Step 1 (Insert):** Normal state with [BOS] at the beginning and [EOS] at the end

- **Step 2 (Unmask and Remask):** [EOS] is followed by a $\mathsf{M}$ token, indicating expanded state

- **Step 3 (Delete):** [BOS] is preceded by a $\mathsf{M}$ token, signaling cleanup phase

During Step 2, the model leverages the first bit of positional encodings (e.g. BiPE introduced in § G) to distinguish between original positions (odd indices) and newly inserted positions (even indices), enabling it to correctly apply remasking operations to original positions based on the rewrite signal $\mathbf{r}_t$ while unmasking new positions to write content from $\mathbf{w}_t$.

We omit the Transformer-based construction for the procedure described above for brevity, which can be done by a simple E-FASP program.    □

We use the Rewrite-MDM variant established above to simulate PRAM algorithms with optimal space complexity. Here we use the $\mathrm{E\text{-}FASP}[\mathrm{BiPE}; [\cdot]_+, \times]$ variant with binary positional encoding (§ G). The input that encodes the PRAM's initial memory state is a sequence $\mathbf{x} = (x_1, \ldots, x_{n+1}) \in \Sigma^{n+1}$ of discrete tokens from the vocabulary $\Sigma$:

$$x_1 \in \Sigma \quad \text{(processor count token)} \tag{80}$$

$$x_{i+1} \in \Sigma \quad \text{(data token)} \quad \text{for } i = 1, \ldots, n \tag{81}$$

Through token embedding $\mathrm{TE} : \Sigma \to \mathbb{R}^w$, these discrete tokens are mapped to their semantic bit representations:

$$\mathrm{TE}(x_1) \mapsto P(n) \in \{0,1\}^w \tag{82}$$

$$\mathrm{TE}(x_{i+1}) \mapsto \mathrm{val}_i \in \{0,1\}^w \quad \text{(data bits)} \tag{83}$$

for $i = 1, \ldots, n$. The actual input to the AP-MDM is the padded sequence $\mathbf{x}_0 = (x_{0,1}, x_{0,2}, \ldots, x_{0,S(n)}) \in \bar{\Sigma}^{S(n)}$ where $\bar{\Sigma} = \Sigma \cup \{\mathsf{M}\}$:

$$x_{0,j} = x_j \quad \text{for } j = 1, \ldots, n+1 \tag{84}$$

$$x_{0,j} = \mathsf{M} \quad \text{for } j = n+2, \ldots, S(n) \tag{85}$$

The crucial advantage of AP-MDM is that it can dynamically rewrite the content at any position using the **remask** operation, allowing the simulation to use space optimally as $\mathcal{O}(S(n))$ rather than the $\mathcal{O}(P(n) \times T(n))$ required by standard MDM.

We next provide an overview of the construction:

The key difference between how Rewrite-MDM and AO-MDM simulate PRAM is that Rewrite-MDM can directly rewrite the memory at any position and each computation does not necessarily have to be kept in the context forever. This enables us to get rid of the address token. Now representation of a processor can be simplified to:

$$\ldots \langle \mathrm{PC}_1, \mathrm{R}_{1,1}, \mathrm{R}_{1,2}, \mathrm{R}_{1,3} \rangle \, \langle \mathrm{PC}_2, \mathrm{R}_{2,1}, \mathrm{R}_{2,2}, \mathrm{R}_{2,3} \rangle \, \ldots \tag{86}$$

where we only use 3 registers (this is sufficient for the proof but can be extended to any $k \geq 2$).

Additionally, we do not append processor representations to the end of input $\mathbf{x}$ as in AO-MDM, but instead will initialize them at the end of the entire sequence. The remaining part of the sequence is used as a shared memory where token embeddings are data and positional encodings are addresses, aligning more closely with PRAM.

**Initialization.** When the last position is a mask token, we initialize the processor state and computation log at the end of the sequence. Roles of each token are calculated similarly as the construction in AO-MDM (except it is static throughout the generation process).

The execution of the program is similar to the construction in AO-MDM, except now the address is inherently associated with the positional encoding. The termination is also slightly different: the returned embedding has to contain an additional bit to indicate the rewrite operation. Also, the termination condition is no longer when all masked are unmasked but a flexibly defined one: in our case, when all processors are HALT.

Using the operators defined above, we now construct an E-FASP program that simulates PRAM execution. The program implements the single-processor algorithm detailed in § E.

```
# ---------------------- Roles & Layout ----------------------
is_mask = (TE == embed([MASK]))
pn = rightmost_exact_match(1, is_first, TE) # number of processors P(n)
last_p = rightmost_exact_match(1, is_last, PE) # last position index
proc_b = last_p - (pn << 2) + 1 # processor region start
in_proc = (PE >= proc_b) and (PE < proc_b + (pn << 2)) # in processor
    region
in_mem = (PE >= 2) and (PE < proc_b) # in memory region

# --------------------- Initialization ---------------------
last_is_mask = rightmost_exact_match(1, is_last, is_mask)
if last_is_mask and in_proc:
   inner = (PE - proc_b)[:2] # slot index 0..3
   # initialize processor region to 0, and R=1 (require rewrite)
   if inner == 0: return (0, 1) # PC
   elif inner == 1: return ((PE - proc_b) >> 2, 1) # R1
   elif inner == 2: return (0, 1) # R2
   else: return (0, 1) # R3
# no rewrite for other positions in initialization step
if last_is_mask and not in_proc:
   return (TE, 0)

# ========= Processor zone update (only if in_proc) =========
if in_proc:
   pid = (PE - proc_b) >> 2
   slot = (PE - proc_b)[:2] # 0:PC 1:R1 2:R2 3:R3

   # read previous round processor state (fixed slot)
   pc_pos = proc_b + (pid << 2) + 0
   r1_pos = proc_b + (pid << 2) + 1
   r2_pos = proc_b + (pid << 2) + 2
   r3_pos = proc_b + (pid << 2) + 3
   PC = rightmost_exact_match(pc_pos, PE, TE)
   R1 = rightmost_exact_match(r1_pos, PE, TE)
   R2 = rightmost_exact_match(r2_pos, PE, TE)
   R3 = rightmost_exact_match(r3_pos, PE, TE)

   # processor already HALTed, do not update (R=0 for this slot)
   if PC == HALT_CODE: return (0, 0)

   # fetch and decode instruction
   I_type, op_r, op_s, op_c, label_addr = get_instruction(PC)
   Rs = (R1 if op_s == 1 else R2 if op_s == 2 else R3 if op_s == 3 else
       0)
   Rr = (R1 if op_r == 1 else R2 if op_r == 2 else R3 if op_r == 3 else
       0)

   # default
   PCn, WR, WR_en = PC + 1, Rr, 0

   # instruction semantics
   if I_type == embed([LOAD]):
```

```python
        # mem_get: address=PE, value=TE (only match in memory region)
        hitp = rightmost_exact_match(Rs, (PE if in_mem else 0), (PE if
            in_mem else 0))
        WR = rightmost_exact_match(hitp, PE, (TE if in_mem else 0))
        WR_en = 1

    elif I_type == embed([STORE]): WR_en = 0
    elif I_type == embed([LOADI]): WR, WR_en = op_c, 1
    elif I_type == embed([ADD]): WR, WR_en = (Rr + Rs), 1
    elif I_type == embed([SUB]): WR, WR_en = (Rr - Rs), 1
    elif I_type == embed([AND]): WR, WR_en = (Rr & Rs), 1
    elif I_type == embed([XOR]): WR, WR_en = (Rr ^ Rs), 1
    elif I_type == embed([SHL]): WR, WR_en = (Rr << Rs)), 1
    elif I_type == embed([SHR]): WR, WR_en = (Rr >> Rs)), 1
    elif I_type == embed([BRZ]) and (Rr == 0): PCn = label_addr
    elif I_type == embed([JMP]): PCn = label_addr
    elif I_type == embed([HALT]): PCn = HALT_CODE

    # unified writeback
    R1n = (WR if (WR_en and op_r == 1) else R1)
    R2n = (WR if (WR_en and op_r == 2) else R2)
    R3n = (WR if (WR_en and op_r == 3) else R3)

    # return next state for this slot and require rewrite
    if slot == 0: return (PCn, 1)
    elif slot == 1: return (R1n, 1)
    elif slot == 2: return (R2n, 1)
    else: return (R3n, 1)

# ========= Memory zone update (also for non MASK) =========
if in_mem:
    # for all PC slots, construct STORE stream (address, value) for this
        step
    is_pc_glob = (((PE - proc_b)[:2]) == 0) and (PE >= proc_b) and (PE <
        proc_b + (pn << 2))
    PCi = (TE if is_pc_glob else 0)
    It, rd, rs, cimm, L = get_instruction(PCi)

    pid_i = ((PE - proc_b) >> 2) # only meaningful for PC slot
    r1_i = rightmost_exact_match(proc_b + (pid_i << 2) + 1, PE, TE)
    r2_i = rightmost_exact_match(proc_b + (pid_i << 2) + 2, PE, TE)
    r3_i = rightmost_exact_match(proc_b + (pid_i << 2) + 3, PE, TE)
    Rs_i = (r1_i if rs == 1 else r2_i if rs == 2 else r3_i if rs == 3 else
        0)
    Rr_i = (r1_i if rd == 1 else r2_i if rd == 2 else r3_i if rd == 3 else
        0)

    STORE_KEYS = (Rs_i if (is_pc_glob and (It == embed([STORE]))) else 0)
        # store address
    STORE_VALS = (Rr_i if (is_pc_glob and (It == embed([STORE]))) else 0)
        # store value

    hit = rightmost_exact_match(PE, STORE_KEYS, 1, 0)
    val = rightmost_exact_match(PE, STORE_KEYS, STORE_VALS, TE)

    # all halt: do not rewrite; otherwise, if hit, rewrite this address (
        even if not MASK originally)
    if hit == 1: return (val, 1)
    else: return (TE, 0)

return (TE, 0)
```

## L  PROOF OF THEOREM 5

**Definition 11** (Two-Sided Dyck-$k$). *Let $\Sigma_k = \{a_1^{\pm 1}, \ldots, a_k^{\pm 1}\}$. Define $u \Rightarrow v$ if $v$ is obtained from $u$ by deleting a factor $a_i a_i^{-1}$ or $a_i^{-1} a_i$ for some $i \in \{1, \ldots, k\}$. Write $u \Rightarrow^* v$ iff there exist $m \geq 0$ and words $u = w_0, \ldots, w_m = v$ with $w_j \Rightarrow w_{j+1}$ for all $j$. Then*

$$\mathrm{TDyck}_k := \{ w \in \Sigma_k^* : w \Rightarrow^* \varepsilon \}. \tag{87}$$

*where $\varepsilon$ is the empty word.*

For the Two-Sided Dyck-$k$ language, we define the vocabulary as:

$$\Sigma = \{a_1^{+1}, a_1^{-1}, a_2^{+1}, a_2^{-1}, \ldots, a_k^{+1}, a_k^{-1}\} \cup \{[\texttt{BOS}], [\texttt{EOS}]\} \tag{88}$$

and the extended vocabulary $\bar{\Sigma} = \Sigma \cup \{\mathsf{M}_1, \mathsf{M}_2\}$, where $\{a_i^{\pm 1}\}_{i=1}^k$ are the $2k$ bracket tokens, $[\texttt{BOS}]$ and $[\texttt{EOS}]$ are boundary tokens, and $\mathsf{M}_1, \mathsf{M}_2$ are two types of mask tokens used to handle an inherent limitation of vanilla masked diffusion when extended to the non-deterministic case (i.e. given two mask tokens, the model can not randomly generate $AA$ and $BB$ without also having probability to generate $AB$ and $BA$). Thus $|\Sigma| = 2k + 2$ and $|\bar{\Sigma}| = 2k + 4$.

Following (Merrill et al., 2022):

**Definition 12** (Hardmax). *For any $x \in \mathbb{R}^n$, define the zero-temperature softmax (Hardmax) as*

$$\mathrm{softmax}_0(x) \triangleq \lim_{\tau \to 0^+} \mathrm{softmax}_\tau(x), \quad where \quad [\mathrm{softmax}_0(x)]_i = \begin{cases} \frac{1}{|\arg\max_j x_j|}, & i \in \arg\max_j x_j, \\ 0, & otherwise. \end{cases}$$

**Definition 13** (Stochastic AP-MDM). *A stochastic AP-MDM is defined as an AP-MDM with encoder-only Transformer backbone as in § F, where instead of greedy decoding, we use sampling from Hardmax distributions. Formally, let $\mathrm{Enc}_\theta : \Sigma^* \to (\mathbb{R}^d)^*$ be the encoder Transformer (before the final projection layer) as defined in Definition 7. For each position $i$ in the input sequence, let $\mathbf{h}_i = [\mathrm{Enc}_\theta(\mathbf{x})]_i \in \mathbb{R}^d$ be the hidden state. Define logits $\ell(v \mid \mathbf{x}, i) = \langle \mathbf{h}_i, \mathrm{TE}(v) \rangle$ and the probability distribution over vocabulary $\Sigma$ as:*

$$p_\theta(v \mid \mathbf{x}, i) = [\mathrm{softmax}_0(\ell(\cdot \mid \mathbf{x}, i))]_v \tag{89}$$

*where $\mathrm{softmax}_0$ is the Hardmax function from Definition 12. The stochastic AP-MDM samples tokens according to $v_i \sim Categorical(p_\theta(\cdot \mid \mathbf{x}, i))$. For the insert operation to support two types of mask tokens ($\mathsf{M}_1$ and $\mathsf{M}_2$), we use two separate classification heads $\mathrm{PROJ}_{I_1}$ and $\mathrm{PROJ}_{I_2}$ whose outputs are thresholded for inserting $\mathsf{M}_1$ or $\mathsf{M}_2$, with priority: $\mathsf{M}_2$ takes precedence over $\mathsf{M}_1$. We disable remask and delete operations for this construction.*

**Theorem 12** (Generating Two-Sided Dyck-$k$, Formal). *For any $k \geq 2$, there exists a stochastic AP-MDM as in Definition 13 with constant-depth Transformer backbone such that the support of the induced distribution of AP-MDM over strings $w$ is exactly equal to $\mathrm{TDyck}_k$, that is,*

1. *(Coverage) For every $w \in \mathrm{TDyck}_k$, $\mathrm{Pr}_\theta[w] > 0$.*

2. *(Support exactness) For every $w \notin \mathrm{TDyck}_k$, $\mathrm{Pr}_\theta[w] = 0$.*

*Conversely, under the common hardness assumption that $\mathsf{TC}^0 \neq \mathsf{NC}^1$, for any constant-depth ARM with polylogarithmic embedding dimension, there exists $N \in \mathbb{N}$ such that the support of the distribution generated ARM cannot be exactly equal to $\mathrm{TDyck}_k$.*

*Proof.* For the claim about ARM, we will prove by contradiction. First we note that every $w \in \Sigma_k^*$, there exists a $w' \in \mathrm{TDyck}_k$ such that $w$ is a prefix of $w'$. (The existence of such $w'$ is straightforward, for example, one can construct it by taking $w' = ww^{-1}$, where the inverse is performed by viewing $w$ as an element of the corresponding free group) Now we suppose ARM can indeed generate a distribution whose support is exactly equal to $\mathrm{TDyck}_k$. This implies that the ARM must be able to determine at each generation step whether the current sequence can be terminated. Specifically, the model must assess whether the currently generated sequence satisfies the complete bracket matching condition. If the sequence is properly matched, the probability of outputting $[\texttt{EOS}]$ must be non-zero to enable termination. Therefore, the difficulty of generating matched brackets reduces to the problem

of recognizing whether a given sequence forms valid matched brackets. For the two-sided Dyck-$k$ language, this recognition problem is DLOGTIME-uniform $\mathsf{NC}^1$-hard (Robinson, 1993), which exceeds the computational capacity of constant-depth Transformers which is in $\mathsf{TC}^0$, under the hardness assumption that $\mathsf{TC}^0 \neq \mathsf{NC}^1$. Thus we conclude ARM cannot generate a distribution over $\Sigma_k^*$ whose support is exactly equal to $\mathrm{TDyck}_k$.

For the stochastic AP-MDM, we construct the following algorithm to generate all strings in the two-sided Dyck-$k$ language through the following algorithmic procedure, illustrated in Figure 2(b):

**Step 1 (Probabilistic Mask Insertion):** If the current sequence contains no mask tokens, then for any sequence position $j$ not containing an end-of-sequence token, the model inserts $\mathsf{M}_1$ with constant probability $p \in (0, 1)$ using the insert operation from § 4. The insertion probability for $\mathsf{M}_2$ is set to zero at this stage.

**Step 2 (Uniform Token Selection):** At positions containing $\mathsf{M}_1$, the model performs two operations: (i) it samples uniformly from the bracket token set $\{a_i^{\pm 1}\}_{i=1}^k$ to determine the content, and (ii) it inserts $\mathsf{M}_2$ with probability 1. Due to the priority-based insertion mechanism defined in Definition 13, $\mathsf{M}_2$ overrides $\mathsf{M}_1$, making the original insertion probability irrelevant for subsequent processing.

**Step 3 (Context-Aware Bracket Matching):** When processing $\mathsf{M}_2$ tokens, the model identifies the nearest bracket token $a_j^{\pm 1}$ to the left of the current position and generates the corresponding matching bracket according to the two-sided Dyck-$k$ reduction rules.

**Termination Condition:** The termination mechanism operates as follows: In Step 2, when the sequence contains $\mathsf{M}_1$ tokens but no $\mathsf{M}_2$ tokens, the model inserts $\mathsf{M}_2$ at the final position with a fixed probability. Subsequently, in Step 3, when processing this final $\mathsf{M}_2$ token, the model generates `[EOS]` (given the binary positional encoding we considered in § E, the model is able to identify if the token should be decoded as `[EOS]` or a matching bracket), signaling the end of the generation process.

The above algorithm admits an `E-FASP` program implementation (see § G) with the following treatment of stochastic operations:

For uniform sampling over the $2k$ bracket tokens, we exploit the fixed vocabulary size by assigning each bracket token $a_i^{\pm 1}$ to distinct dimensions in the $d$-dimensional embedding space. Specifically, when the `E-FASP` program needs to output a uniform distribution over a subset $S \subseteq \{a_i^{\pm 1}\}_{i=1}^k$, it returns a hidden state $\mathbf{h} \in \mathbb{R}^d$ where $\langle \mathbf{h}, \mathrm{TE}(v) \rangle = c$ for all $v \in S$ (for some constant $c$) and $\langle \mathbf{h}, \mathrm{TE}(v') \rangle < c$ for $v' \notin S$. The hardmax from Definition 13, ensures that the probability mass concentrates uniformly over $S$, achieving the desired uniform sampling behavior.

It is easy to see that this generation procedure can produce any string in the two-sided Dyck-$k$ language, and since any token that would violate Dyck constraints always has strictly smaller logit and hence probability 0 under Hardmax, the support of the distribution is exactly $\mathrm{TDyck}_k$. $\qquad \square$

We note the introduction of two mask tokens is for the model to distinguish between different steps, but this is not necessary if we allow some random seeds in input which mitigates the limitation of MDM when extending to the non-deterministic case.

## M    PROOF OF THEOREM 6

**Edit Triplet Encoding.**    We encode each elementary edit as a triplet of tokens $\langle \texttt{op}, \texttt{pos}, \texttt{val} \rangle \in \Sigma_{\mathrm{op}} \times \Sigma_{\mathrm{pos}} \times \Sigma$, where

$$\Sigma_{\mathrm{op}} = \{\texttt{UNMASK}, \texttt{INSERT}, \texttt{DELETE}, \texttt{REMASK}\} \tag{90}$$

The position token set $\Sigma_{\mathrm{pos}}$ reuses the earlier address/position encoding in Definition 10: a position $i \in [S(n)]$ is encoded as $\texttt{pos} = \texttt{encode}(i)$ with inverse decoding $\texttt{dec\_pos}(\texttt{pos}) = i$.

The semantics of the triplet follows the instantiation of AP-MDM considered in § 4.

**Definition 14** (Editing Sequence)**.**  *Given an input sequence* $\mathbf{x} \in \bar{\Sigma}^*$*, an editing sequence is a finite sequence of triplets*

$$T(\mathbf{x}) = \big(\langle \mathrm{op}_j, \mathrm{pos}_j, \mathrm{val}_j \rangle\big)_{j=1}^{m(\mathbf{x})}, \quad \mathrm{op}_j \in \Sigma_{\mathrm{op}}, \ \mathrm{pos}_j \in \Sigma_{\mathrm{pos}}, \ \mathrm{val}_j \in \Sigma. \tag{91}$$

*Its application to $\mathbf{x}$ is defined recursively by $\mathbf{x}^{(0)} = \mathbf{x}$ and*

$$\mathbf{x}^{(j)} = \text{Apply}\big(\langle \text{op}_j, \text{pos}_j, \text{val}_j \rangle, \mathbf{x}^{(j-1)}\big), \quad j = 1, \dots, m(\mathbf{x}). \tag{92}$$

*We write*

$$\text{Apply\_Triplets}\big(T(\mathbf{x}), \mathbf{x}\big) := \mathbf{x}^{(m(\mathbf{x}))}. \tag{93}$$

*An editing sequence is valid iff every intermediate application is well-defined under the triplet semantics (e.g.,* `UNMASK` *applies only to masks).*

**Theorem 13** (Hardness of Simulating AP-MDM, Formal)**.** *There exists an AP-MDM $F$ with a constant-depth encoder-only Transformer backbone such that no ARM or Masked-ARM $G$ (Definition 3) with a constant-depth decoder-only Transformer backbone can, on every input $\mathbf{x}$, produce an editing sequence $T_G(\mathbf{x})$ (Definition 14) that realizes $F$'s generation process; i.e., under the assumption that constant-depth Transformers do not include $\mathsf{TC}^0$,*

$$\forall G \, \exists \mathbf{x} \in \bar{\Sigma}^* : \text{Apply\_Triplets}\big(T_G(\mathbf{x}), \mathbf{x}\big) \neq \text{Apply\_Triplets}\big(T_F(\mathbf{x}), \mathbf{x}\big),$$

*or $T_G(\mathbf{x})$ is invalid.*

The ARM in the above result can be replaced by the Masked-ARM with encoder architecture used in Theorem 3 without affecting the result.

*Proof.* Fix $L \in \mathbb{N}$. Let $\mathbf{u} \in \Sigma^L$ be the base string, let $T$ be a valid editing sequence (Definition 14), and let $q \in \Sigma_{\text{pos}}$ be a query position token with index $i = \text{dec\_pos}(q)$. Encode the input as

$$\mathbf{x} = (\mathbf{u}, \text{[SEP]}, \text{flatten}(T), \text{[SEP]}, q, \text{[SEP]}) \in \Sigma^{L+3+3m(\mathbf{u})}. \tag{94}$$

Here

$$\text{flatten}(T) = (\text{op}_1, \text{pos}_1, \text{val}_1, \dots, \text{op}_m, \text{pos}_m, \text{val}_m). \tag{95}$$

The task is to output the queried symbol after applying the editing history:

$$y = \big[\text{Apply\_Triplets}\big(T(\mathbf{u}), \mathbf{u}\big)\big]_i \in \Sigma. \tag{96}$$

That is, the instance provides (i) a base string $\mathbf{u}$, (ii) an editing history $T$ as a sequence of triplets, and (iii) a query position token $q$. The model must simulate $T$ on $\mathbf{u}$ and return the symbol at the queried position $i$ in the resulting string. For AP-MDM, the simulation is intuitive and can be proven by simple `E-FASP` program which we skip in this proof.

For ARM, due to the construction of the problem, the simulation process is exactly copying the editing sequence part in the input, therefore solving the problem is equivalent to directly answer the query, which we show the equivalence to a $\mathsf{TC}^0$-hard task:

**Definition 15** (PRESERVES (Allender et al., 2006))**.** *Let $A$ be an ordered list (1-indexed). The update alphabet is*

$$\mathcal{U} = \{\texttt{insert}(i), \texttt{delete}(i) \mid i \in \mathbb{N}\}. \tag{97}$$

*For an initial list $A_0$ and an update sequence $s \in \mathcal{U}^*$, let $A_t$ be the list after applying the first $t$ updates of $s$. For indices $i, j \in \mathbb{N}$, define*

PRESERVES$(A_0, s, i, j) \iff$ *the item at position $i$ in $A_0$ still exists after $s$ and is at position $j$ in $A_{|s|}$.* (98)

*The decision problem* PRESERVES *asks, given $(A_0, s, i, j)$, whether* PRESERVES$(A_0, s, i, j)$ *holds.*

**Conjecture 14.** *The* PRESERVES *problem is $\mathsf{NC}^1$-hard under DLOGTIME-uniform reductions.*

We reduce PRESERVES to our Editing-Query task in one step. Given $(A_0, s, i, j)$ with $|A_0| = L$, let $\mathbf{u} \in \Sigma^L$ list the items of $A_0$ (unique token id per item); expand each `insert`$(p, v)$ as the triplet block $\big(\langle \text{INSERT}, \text{encode}(p), \mathsf{M} \rangle, \langle \text{UNMASK}, \text{encode}(p'), v \rangle\big)$ and each `delete`$(p)$ as $\big(\langle \text{REMASK}, \text{encode}(p), \bullet \rangle, \langle \text{DELETE}, \text{encode}(p), \bullet \rangle\big)$, where $p'$ is the position of the newly inserted mask under our convention and $\bullet$ is ignored; let $T$ be the concatenation over $s$ and set $q = \text{encode}(j)$. Then with

$$y = \big[\text{Apply\_Triplets}\big(T, \mathbf{u}\big)\big]_j, \tag{99}$$

we have

$$\text{PRESERVES}(A_0, s, i, j) \iff y = \text{id}(i). \tag{100}$$

Thus any model that solves Editing-Query on all inputs also decides PRESERVES. By Conjecture 14, PRESERVES is $\text{NC}^1$-hard, which places it beyond the computational capacity of constant-depth Transformers that are contained in $\text{TC}^0$. Under this conjecture, no constant-depth ARM or Masked-ARM can solve our Editing-Query task on all inputs. This completes the proof.

$\square$

## N   EXPERIMENT: SUDOKU PUZZLE

We provide detailed description of training data generation for the Sudoku puzzle experiment. The data generation process simulates a backtracking-based solving algorithm and records the intermediate states as supervised training trajectories for AP-MDM.

**State Representation**   The $9 \times 9$ Sudoku grid is represented as a sequence of 324 tokens, where each cell is encoded using 3 consecutive tokens: $(\text{value}, \text{color}, \text{marker})$. The vocabulary consists of 32 tokens including: EMPTY (unfilled cell), MASK (unknown position to be predicted), digits 1-9 (filled values), WHITE (default color indicating no branch), 15 branch COLOR tokens (tracking different branching paths), NORMAL (standard state), SKULL (failed branch point), BRANCH (active branch starting position), and SEPARATOR (structural delimiter between cells). The value token can be EMPTY (for unfilled cells), a digit 1-9 (for filled cells), or MASK (for positions being predicted). The color token tracks which branching decision path the cell belongs to, using WHITE for non-branched cells and one of 15 COLOR tokens for cells within branches. The marker token indicates the cell's computational status: NORMAL for standard cells, SKULL for positions that caused backtracking, and BRANCH for branch starting points.

The solving algorithm consists of several atomic operations, each translated into state-transition tuples for AP-MDM training:

**Assign and Branch:** The two most fundamental operations in Sudoku solving are deterministic assignment and branch creation, both implemented through combinations of **remask** and **unmask** operations. For Assign, when deterministically filling a cell with value $v$, we generate a 2-step transition: first, the three tokens at the target position are remasked, converting them to $(\mathsf{M}, \mathsf{M}, \mathsf{M})$; second, these masks are unmasked to the target values $(v, c, \text{NORMAL})$ where $c$ is the appropriate color token determined by the current branch context. For Branch, when creating a branch at position $(r, c)$ with candidate value $v$ and branch identifier $b$, we similarly generate a 2-step transition: first, remask the cell tokens to $(\mathsf{M}, \mathsf{M}, \mathsf{M})$; second, unmask to $(v, \text{COLOR}_b, \text{BRANCH})$ to mark this cell as a branch starting point with the corresponding branch color.

Concrete examples visualizing these two operations are shown in Figure 5.

**Contradiction Marking, Backtrack, and Recovery:** When the solver detects a contradiction (a cell with no valid candidates), before backtracking, we mark this contradicting position through a 2-step transition: first remask the cell to $(\mathsf{M}, \mathsf{M}, \mathsf{M})$, then unmask to $(\text{EMPTY}, \text{COLOR}_b, \text{SKULL})$ to visually indicate the contradiction location. When backtracking from a failed branch, multiple cells need to be cleared through a 2-step transition that handles both ordinary cells and the branch starting point differently: in the first step, ordinary cells are remasked in all three token positions while the branch starting point only has its marker token remasked; in the second step, ordinary cells are unmasked to $(\text{EMPTY}, \text{WHITE}, \text{NORMAL})$ while the branch starting point undergoes simultaneous remask and unmask operations with the first two positions remasked to $(\mathsf{M}, \mathsf{M})$ and the third position unmasked to store the failed value. After backtracking, when filling the failed position with a new value, we generate a 2-step transition to convert from the SKULL state: the first step unmasks the value and color while remasking the marker to $(\mathsf{M}, \mathsf{M}, v_{\text{old}}) \to (v_{\text{new}}, c, \mathsf{M})$, and the second step unmasks the marker to either NORMAL or BRANCH depending on whether this creates a new branch. Concrete examples visualizing these operations are shown in Figure 5.

**Training Data Statistics**   Following this generation process, each Sudoku puzzle produces 1421.3 state-transition tuples (i.e. from $\mathbf{x}_t$ to $\mathbf{x}_{t+1}$) on average depending on the puzzle difficulty and the number of backtracking steps required. For our experiments, we generated training data from 100 hard Sudoku puzzles, each yielding 25,022.6 training state-transition tuples on average.

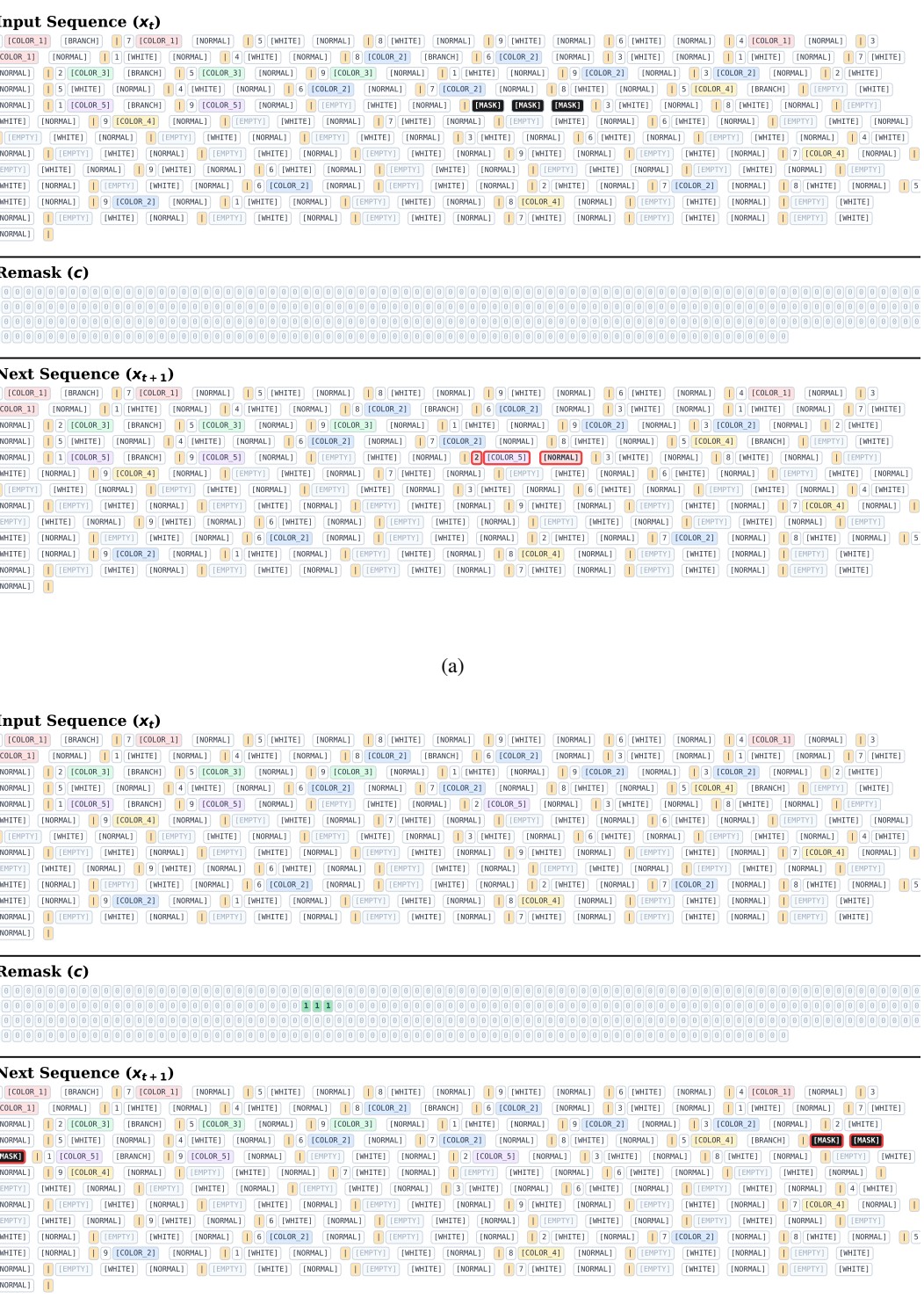

Figure 5: Demonstration of value assignment.

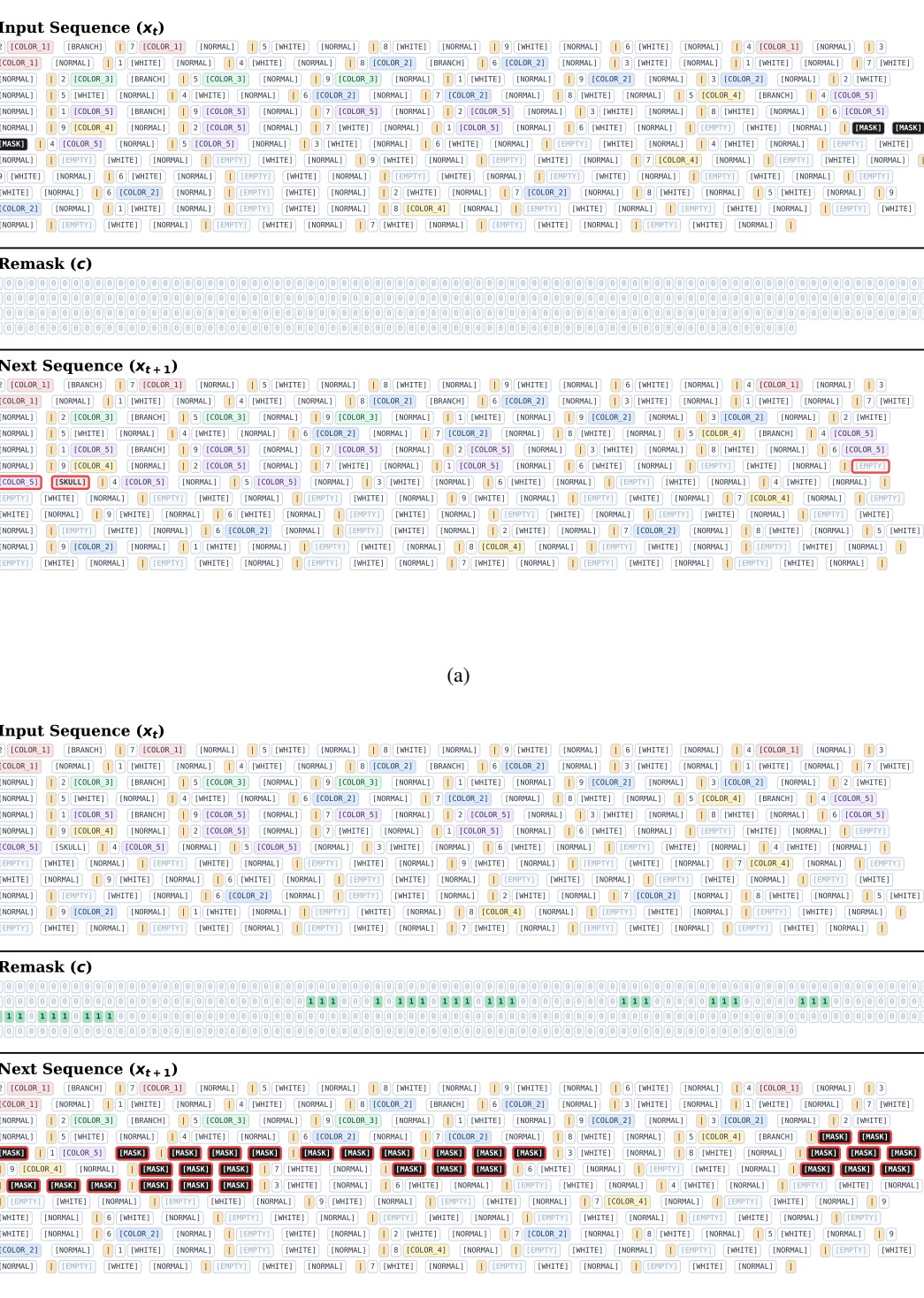

(a)

(b)

Figure 5: Demonstration of backtracking and recovery from failure operations (Part 1). (Continued on next page)

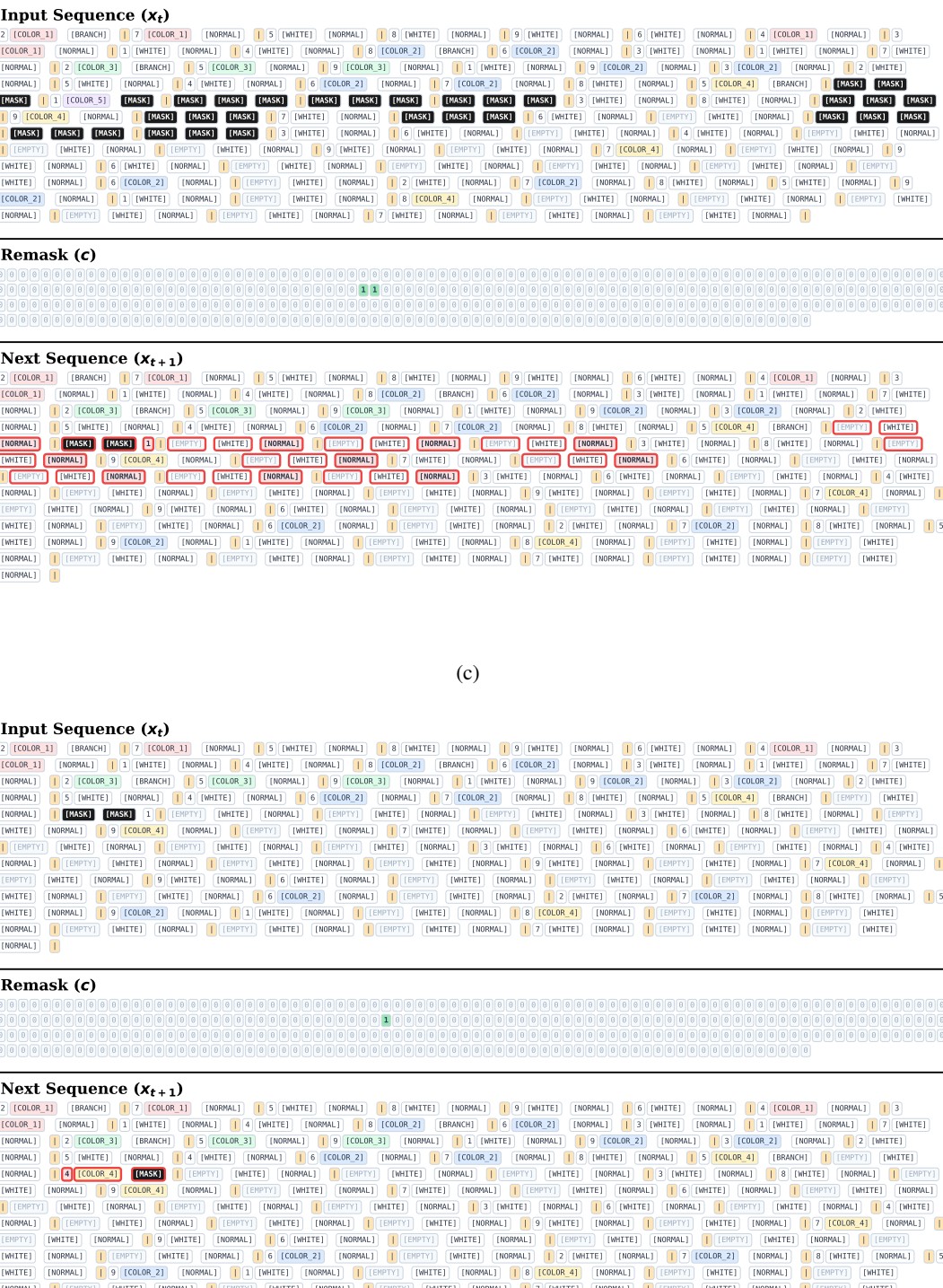

(c)

(d)

Figure 5: (Continued) Part 2.

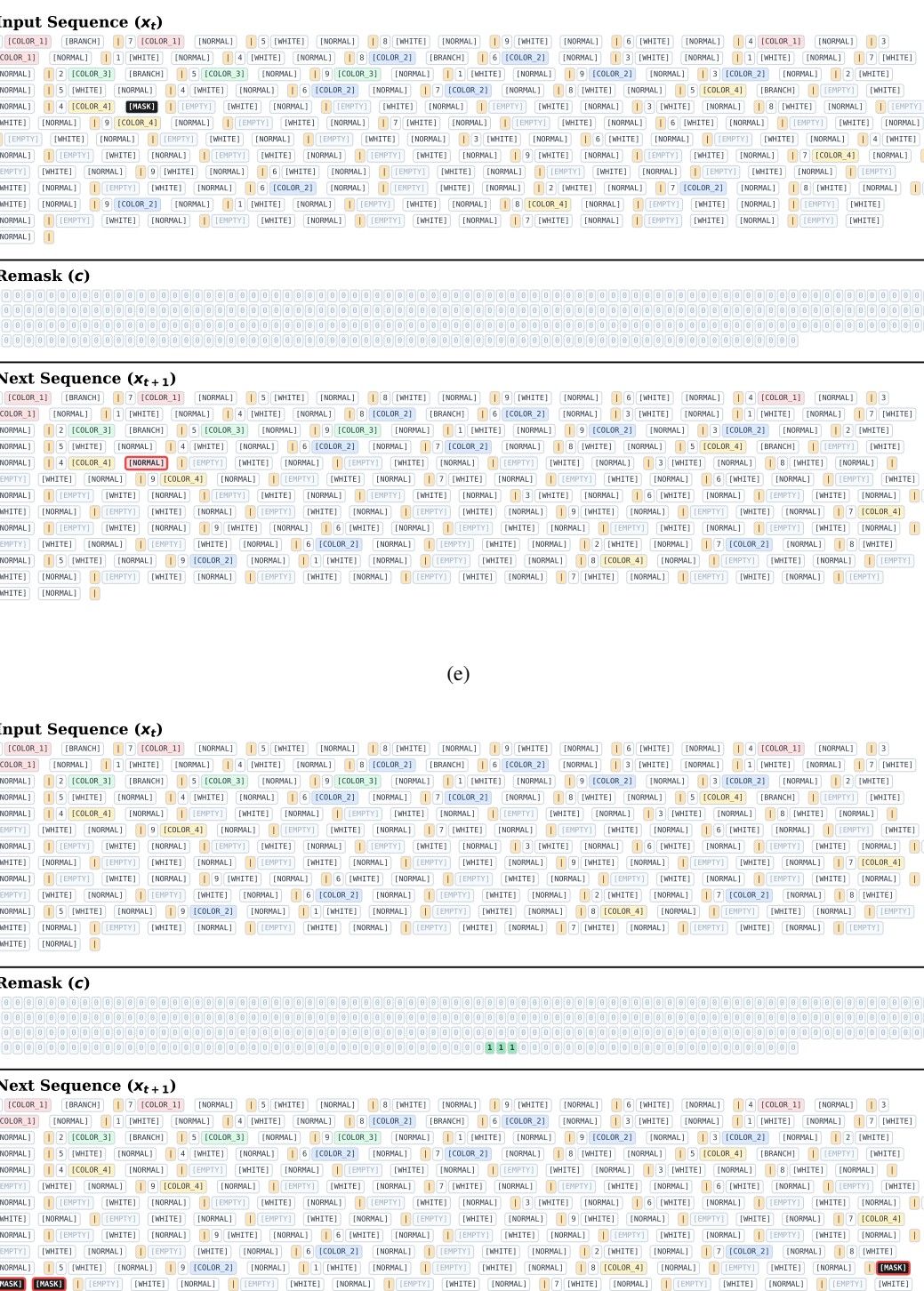

(e)

(f)

Figure 5: (Continued) Part 3.

## O   EXPERIMENT: GRAPH GENERATION

We consider a graph editing task that requires computing the minimum edge set to disconnect two specified nodes. Formally, given a directed graph $G = (V, E)$ with unit-capacity edges and two designated nodes $s, t \in V$ (source and target), the task is to generate a modified graph $G' = (V, E')$ where $E' \subseteq E$ such that there exists no path from $s$ to $t$ in $G'$, and $|E \setminus E'|$ (the number of removed edges) is minimized. This is equivalent to computing the minimum $s$-$t$ cut, which by the max-flow min-cut theorem equals the maximum flow from $s$ to $t$. The generation process involves iteratively finding augmenting paths, modifying the graph structure by reversing edge directions, and tracking intermediate states until no more augmenting paths exist. The final output is the graph with min-cut edges removed, effectively disconnecting $s$ and $t$.

We provide detailed description of training data generation for this graph editing task based on the Edmonds-Karp algorithm. The data generation process simulates the BFS-based augmenting path search and records the intermediate algorithmic states as supervised training trajectories for AP-MDM.

**State Representation**   A directed graph with $n$ nodes and $m$ edges is represented as a token sequence with three main components: prompt (source and target nodes), graph data (edge list with features), and node data (node list with features). Each edge is encoded with its endpoints $(u, v)$ and two feature slots tracking the edge's directional availability: slot1 represents forward direction availability (initially FB for forward-backward) and slot2 represents reverse direction availability (initially MASK indicating unavailable). Each node is encoded with its ID and two features: level (BFS layer, initially INF for unvisited or LVL0 for source) and parent (parent node in BFS tree, initially NIL). The vocabulary includes structural tokens (PROMPT, SRC, TGT, GRAPH, NODES, parentheses), edge feature tokens (FB, MASK), node feature tokens (LVL0-LVL9, INF, NIL, PAR), node IDs (0-299), and termination tokens (EOA, EOS).

**Atomic Operations**   The Edmonds-Karp algorithm is decomposed into atomic operations, each translated into state-transition tuples for AP-MDM training. The algorithm consists of four phases:

**Feature Expansion:** Before the algorithm begins, edge and node feature slots must be initialized through a 3-step process following a expansion-then-unmask paradigm. First, MASK tokens are inserted after each edge's second node and after each node's ID (**insert** operation). Second, another MASK is inserted while simultaneously unmasking the first MASK to FB for edges and to the appropriate level for nodes (**insert** + **unmask** operations). Third, the second MASK is unmasked to MASK for edges and to NIL for nodes (**unmask** operation). See Figure 6.

**Breadth-First Search:** The breadth-first search proceeds by discovering nodes layer by layer. When a new node is discovered through an edge, we generate a 2-step transition: first, remask the node's level and parent features to MASK (**remask** operation); second, unmask these MASKs to the new level and parent ID (**unmask** operation). This continues until either the target node is reached (proceed to augmentation) or no new nodes can be discovered (algorithm terminates). See Figure 7.

**Path Augmentation:** When an augmenting path from source to target is found, we generate a 2-step transition to flip edges along the path and reset node features. First, for each edge on the path, swap its slot1 and slot2 values (reversing the direction), and simultaneously remask all node features to MASK (**remask** operation). Second, unmask all node MASKs back to their initial values: INF for non-source nodes and LVL0 for the source node, with all parents set to NIL (**unmask** operation). After augmentation, the algorithm returns to BFS phase to search for the next augmenting path. See Figure 8.

**Termination and Structural Editing:** When no more augmenting paths exist, the final BFS identifies two disjoint sets $S$ and $T$ where $S$ contains the source and $T$ contains the target. The min-cut edges are those directed from nodes in $S$ to nodes in $T$ in the original graph, and these edges must be removed to disconnect source and target. This is accomplished through a 3-step process. First, remask all tokens representing the min-cut edges to MASK (**remask** operation). Second, delete these edges while simultaneously expanding a MASK token after EOA (**delete** + **insert** operations). Third, unmask the expanded MASK to EOS (**unmask** operation), marking algorithm completion. See Figure 9.

| # Nodes | # Edges | AP-MDM | | ARM | |
|---|---|---|---|---|---|
| | | Avg. Seq. Length | Max Seq. Length | Avg. Seq. Length | Max Seq. Length |
| 4 | 12 | 56 | 65 | 932 | 932 |
| 5 | 17 | 81 | 88 | 1,375 | 1,403 |
| 6 | 23 | 114 | 129 | 2,083 | 2,140 |
| 7 | 29 | 148 | 170 | 2,687 | 2,874 |
| 8 | 36 | 189 | 217 | 3,529 | 3,865 |
| 9 | 43 | 236 | 252 | 5,586 | 6,597 |
| 10 | 50 | 270 | 305 | 4,915 | 5,392 |

Table 1: Sequence length statistics for graphs of different sizes. AP-MDM sequences contain state-transition tuples, while ARM sequences enumerate all operations explicitly.

**ARM Baseline**   For comparison with AP-MDM, we train an autoregressive baseline that learns to generate the complete solving trajectory as a single sequence. The ARM baseline takes the initial graph state as input and must generate the full sequence of operations needed to solve the task. To enable this, we convert the AP-MDM training data into an ARM-compatible format by representing each operation as a triplet $(p, o, v)$ where $p$ is the position index (POSE0-POSE399), $o$ is the operation type (REMASK, UNMASK, INSERT, DELETE), and $v$ is the value (or NONE for operations without values). The sequence format is: [initial state $\mathbf{x}_0$] STEP [operations$_1$] STEP [operations$_2$] STEP $\cdots$ ANSWER [final state $\mathbf{x}_{\text{final}}$], where each STEP separates consecutive state transitions and ANSWER marks the beginning of the final output. For ARM training, we use the standard next-token prediction objective with teacher forcing, where the model learns to autoregressively generate the entire operation sequence given the initial state.

As shown in Table 1, the sequence length grows with graph size for both AP-MDM and ARM, but ARM requires significantly longer sequences due to the explicit operation enumeration.

## P   EXPERIMENT: PARITY

The parity task requires determining whether a binary sequence contains an even or odd number of 1s. Formally, given an input sequence $\mathbf{x} = (x_1, x_2, \ldots, x_n) \in \{0, 1\}^n$, the task is to compute $\bigoplus_{i=1}^n x_i$ (XOR of all bits), outputting 0 for even parity and 1 for odd parity.

**Algorithm and Data Generation**   For AP-MDM training, we implement an elimination algorithm that mimics how humans naturally solve parity: repeatedly remove pairs of identical elements until only the result remains. The vocabulary consists of 5 tokens: BOS (sequence start), EOS (sequence end), MASK, digit 0, and digit 1. The elimination process follows these rules: when encountering 0s in the sequence, convert them to MASK; when encountering a pair of 1s, convert both to MASK; then delete the MASK tokens. This process repeats until only BOS remains (for even parity) or BOS followed by a single 1 remains (for odd parity). Each elimination step generates a state-transition tuple: converting tokens to MASK is a **remask** operation, and removing MASKs is a **delete** operation.

**Data**   The training data consists of only 4 instances that cover all possible elimination patterns. Each sample is a single-step state transition demonstrating one atomic operation. The test set contains 1,000 randomly generated binary sequences with varying lengths. The test sequences have approximately equal distribution of even and odd parities.

**ARM Baseline**   For the ARM baseline, we train autoregressive models with chain-of-thought reasoning where the model generates intermediate cumulative XOR values at each position before outputting the final result. The sequence structure is: BOS [$x_1 \, x_2 \, \cdots \, x_n$] EOP [$s_1 \, s_2 \, \cdots \, s_n$ result] EOS, where $s_i = \bigoplus_{j=1}^i x_j$ represents the cumulative XOR up to position $i$, and result is True (for odd parity) or False (for even parity). Only the content after EOP is used for computing the training loss. We train ARM models with up to 10K training instances at various fixed lengths (e.g., length 2, 10, 50, 100) and evaluate their ability to generalize to longer unseen lengths.

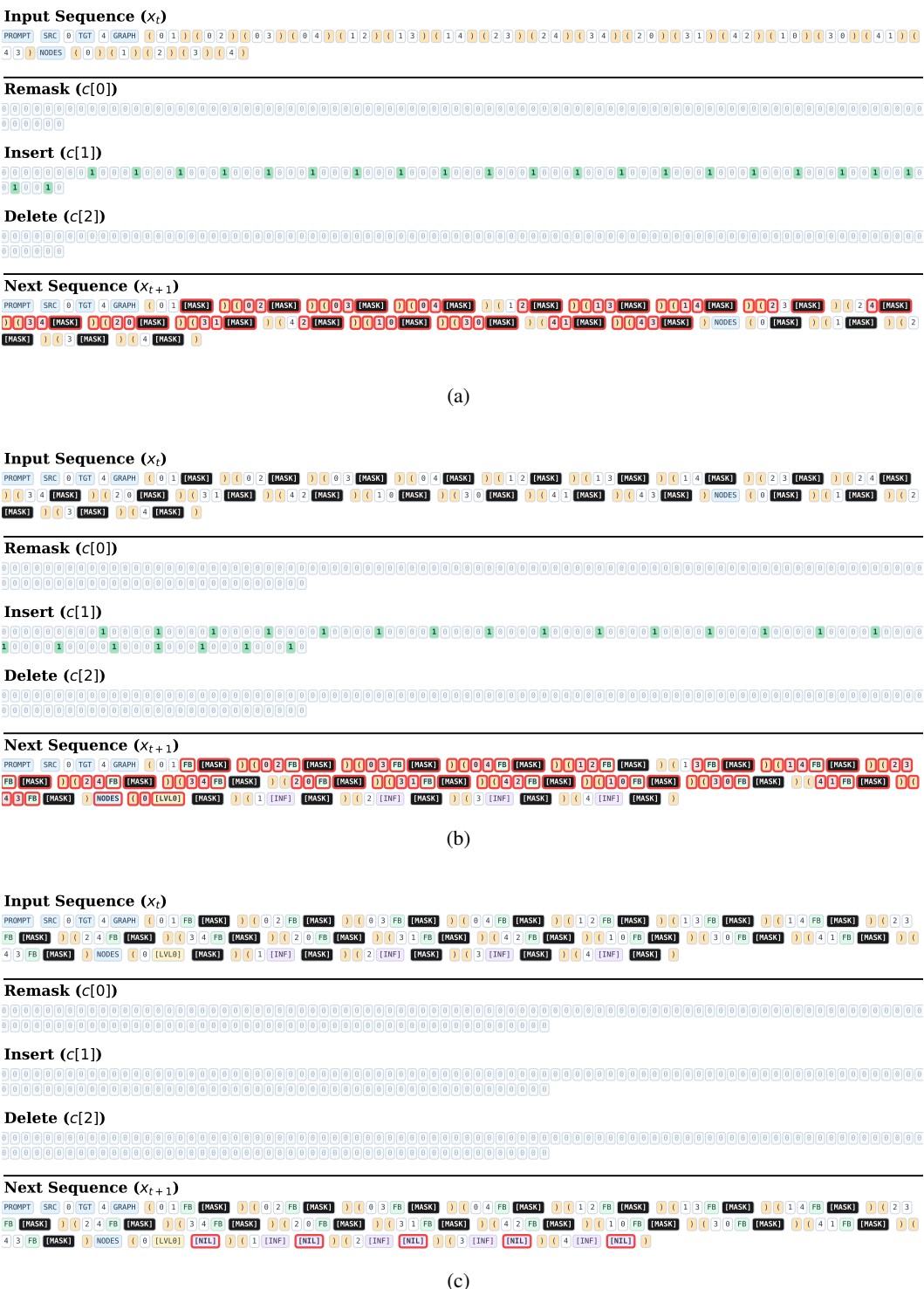

Figure 6: Feature Expansion phase in graph generation, showing the initialization process for edge and node feature slots using **insert** and **unmask** operations.

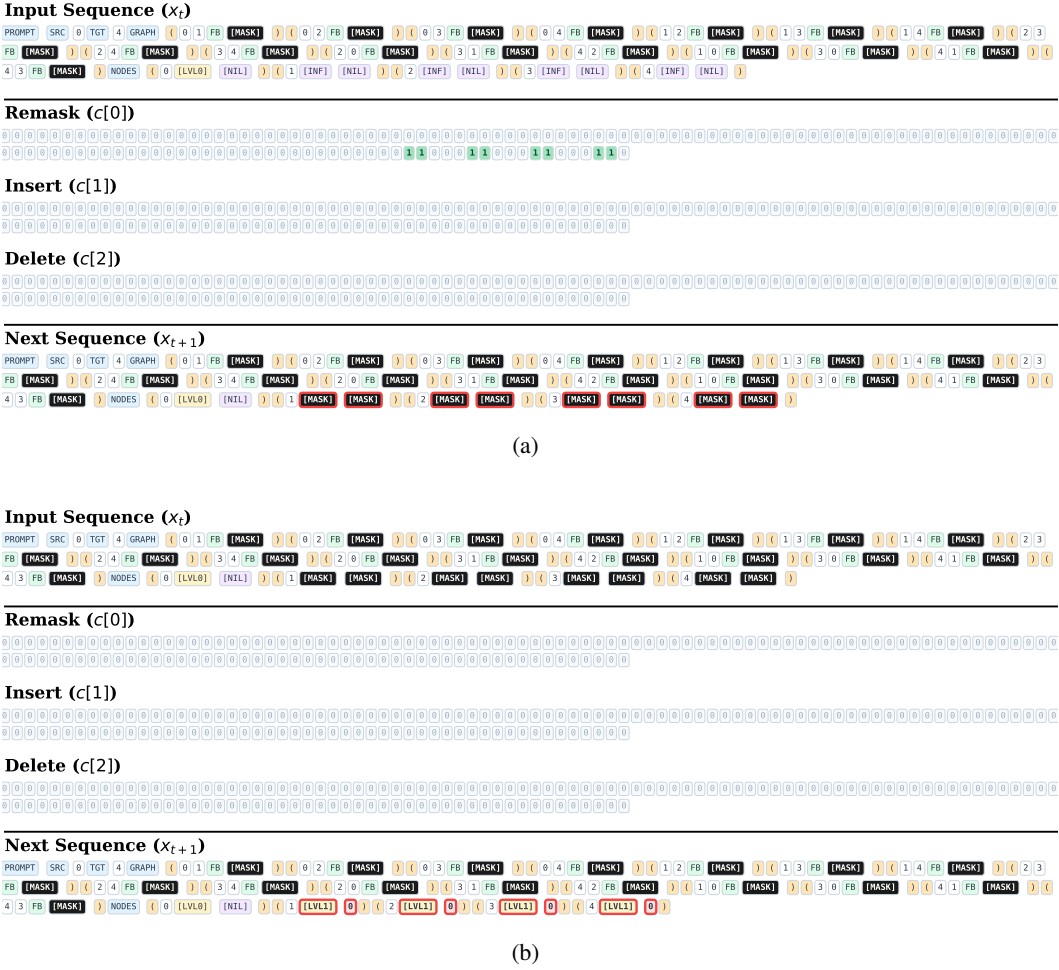

Figure 7: Parallelized BFS phase in graph generation, showing layer-by-layer node discovery with parallel processing using **remask** and **unmask** operations.

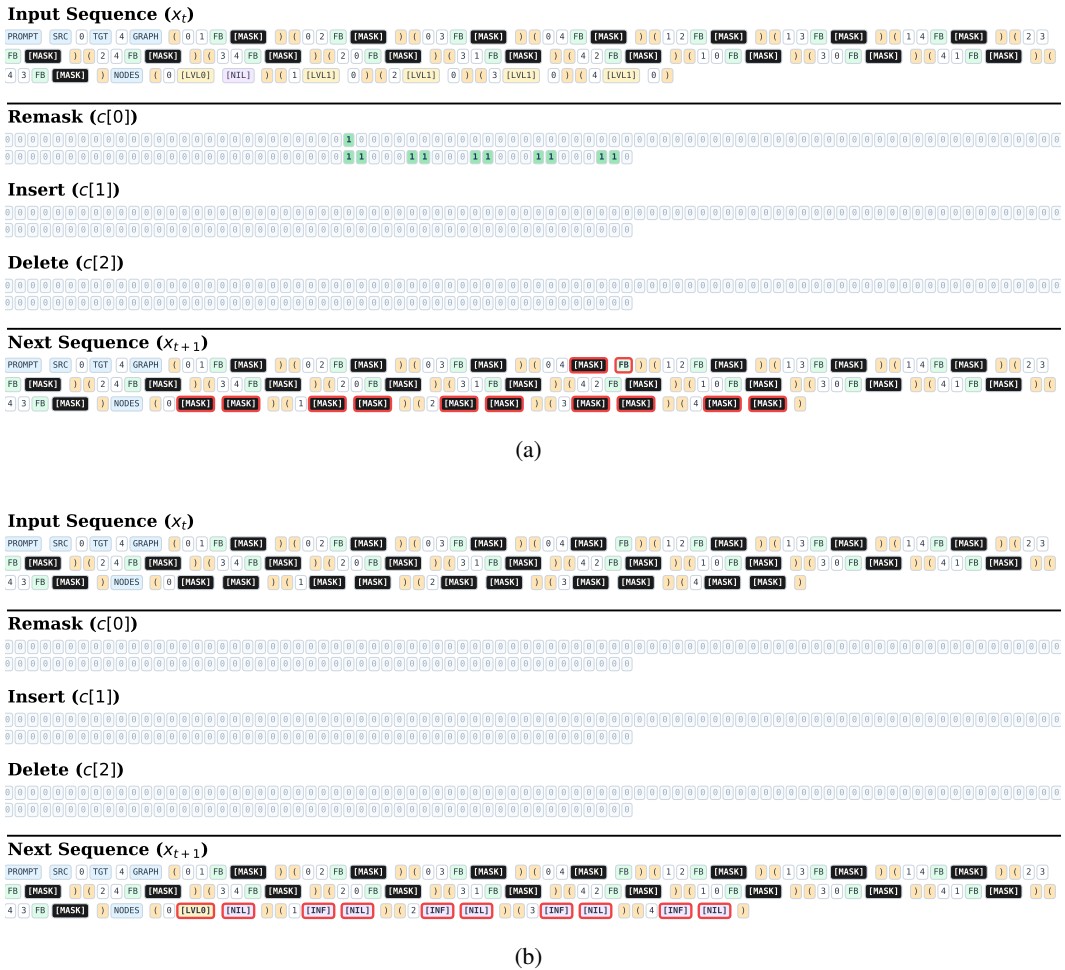

Figure 8: Path Augmentation phase in graph generation, showing edge reversal and node feature reset after finding an augmenting path using **remask** and **unmask** operations.

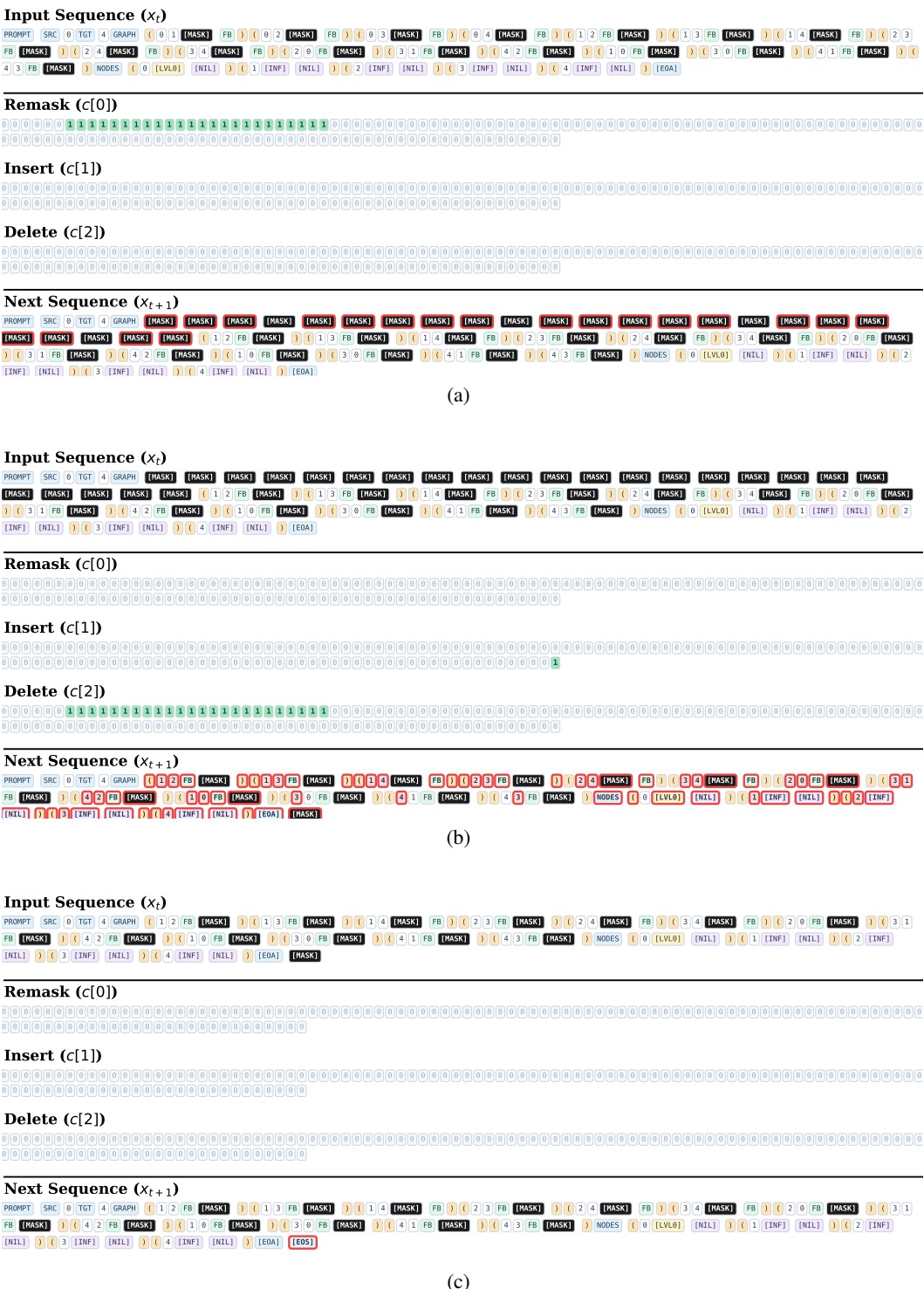

Figure 9: Termination and Structural Editing phase, showing the deletion of min-cut edges and algorithm completion using **remask** and **delete** operations.

