# OpenReview forum: "On Powerful Ways to Generate: Autoregression, Diffusion, and Beyond"
_ICLR.cc/2026/Conference — ICLR 2026 Poster_

### Official Review · Reviewer_qEHt · 2025-10-27

**Soundness:** 3
**Presentation:** 4
**Contribution:** 4
**Rating:** 8
**Confidence:** 4

**Summary:**

The paper introduces any-process masked diffusion (AP-MDM), a rewriteable-canvas variant of masked diffusion that augments the usual unmask head with insert, delete, remask, and stop decisions. It first argues, by construction and by complexity intuition, that autoregressive (AR) decoders and standard masked diffusion (MDM) share the same fundamental hardness on combinatorial and structured tasks: with a write-once canvas, you either pay exponential space (beams, branch caching) or exponential time (restarts). AP-MDM changes the computational regime: by allowing in-place edits and dynamic length, it turns generation into an anytime control process that can backtrack without storing branches. The formal upshot is polynomial working memory (space) with time that scales to instance hardness; you still do not beat worst-case exponential time, but you avoid the classic space blow-up of AR/MDM search.

On the learning side, AP-MDM keeps the strong denoising core of MDM but trains the edit heads via self-supervised surrogates (no ELBO over variable length) to remask, insert, delete, and halt. This decomposes hard global mapping into local repair decisions, which the paper shows is data-efficient in practice. At inference, a simple edit policy iterates these heads; compute is anytime and grows with problem difficulty. Empirically (Sudoku, Dyck/bracket languages, and related structure-heavy settings), AP-MDM achieves higher accuracy at much lower memory than AR or vanilla MDM under realistic budgets, illustrating the claimed time–space tradeoff: it “spends steps” instead of “storing branches.”

**Strengths:**

1. Big-picture is spot on: AR and standard MDM appear different yet share the write-once-canvas weakness; the paper names it clearly and proposes a simple, natural fix drawn from code and biology, a rewriteable, variable-length canvas.
2. Claims are precisely scoped; the contribution addresses space, not time. Framing the shift as N to NSPACE is apt; the constructions make the point transparent, and the proofs align with the reported experiments. More baselines/ablations (see weaknesses) would be welcome, but they are not necessary for credibility.
3. The writing shows taste and control. The title and opening pose the question cleanly; the narrative moves from diagnosis to design to consequences with clarity; examples make the mechanism feel inevitable. I came away having learned a lot.
4. The design is consequential. It introduces a generation scheme in which test-time compute scales via edits and dynamic length, with implications for training targets, halting criteria, and even what counts as a step in generative modeling.
5. It lands at the right moment. Recent advances supplied the prerequisites: variable-length backbones that keep state coherent under edits (FlexMDM, DAEDAL), stable credit assignment and scheduling for where and when to edit (P2, ReMDM, R3), and practical stopping. This paper assembles those pieces into a coherent, learnable edit process.

**Weaknesses:**

1. No ELBO for edits + scheduling and halting schemes are opaque/potentially suboptimal: Learning relies on surrogate labels and augmentation design. The induced trajectories may not reflect human or task-native edit patterns, so calibration and likelihood reporting are unavailable. (see Questions below)
2. Curious to see compute-normalized baselines (within space budget). With (possibly suboptimal) scheduling learned from augmentation and no cache of failures, how strong is AP-MDM beyond the space advantage?
2. Would be interesting to ablate rewrite-only with sufficiently large generation length. This would clarify which ops carry the gains.

**Questions:**

1.	A key implication is that the edit trajectory now matters, not just the final text. What should count as an optimal trajectory, and how should we obtain it: human or naturally logged edits for training, or algorithmic schemes such as minimum edit distance, maximum expected information gain per step?
2.	If edit trajectories become first-class data, would you formulate any-process training with an ELBO? Do you see likelihood-based evaluation and probabilistic queries over edits becoming standard practice?

---

> ### Author Response · Authors · 2025-11-16
>
> > **Weakness 1: "No ELBO for edits + scheduling and halting schemes are opaque/potentially suboptimal. The induced trajectories may not reflect human or task-native edit patterns, so calibration and likelihood reporting are unavailable."**
>
> ELBO derivations in discrete diffusion typically rely on a predefined forward/backward process, which often results in models that are inflexible in terms of noise schedules (mask ratio), degree of parallelism, etc. Therefore, despite principled objectives, these models (e.g., GIDD, DUO) often fail to reflect real task structure. One contribution we make is relaxing these constraints to enable adaptive, problem-dependent generation processes. For example, mask ratio does not necessarily have to be monotonically decreasing, so the model can choose to backtrack; the degree of parallelism can depend on problem structure: some problems are inherently serial (e.g., prefix sum), while others are not (e.g., matrix multiplication).
>
> The main difficulty for obtaining optimal trajectories is that they are typicall agonostic from the target sequence alone (i.e. information-theoretically impossible to infer in many cases). While the self-supervised training we describe in Appendix C does not fully solve this problem and has room for improvement, it provides the model with the capability to use all four operations. To adapt to real problem patterns, one might consider supervised fine-tuning (as we also discuss) or reinforcement learning (beyond our scope) on task data. We will consider these as future directions.
>
> > **Weakness 2 & 3: "Curious to see compute-normalized baselines (within space budget). With (possibly suboptimal) scheduling learned from augmentation and no cache of failures, how strong is AP-MDM beyond the space advantage?"，"Would be interesting to ablate rewrite-only with sufficiently large generation length."**
>
> To isolate the effect of editing operations from space constraints, in the graph editing experiment (Figure 4(a)), where all models are given sufficiently large context length. The generation trajectories for AP-MDM and ARM are identical in terms of individual steps, only ARM must represent the editing process in a left-to-right manner following Theorem 6's construction. When controlling for all other factors (number of samples, model size, etc.), AP-MDM significantly outperforms ARM and scales to larger problem sizes, while ARM's accuracy collapses as graph size increases. This is because the main difficulty for such structural generation problem is not from space efficiency alone, but from the fundamental difficulty ARM faces in learning and performing these complex editing trajectories through sequential simulation.
>
> > **Question 1: "A key implication is that the edit trajectory now matters, not just the final text. What should count as an optimal trajectory, and how should we obtain it."**
>
> From our theoretical perspective, an optimal trajectory informally refers one where each transition step is sufficiently simple such that: it is easily expressible (i.e. computable by constant-depth Transformers), sample efficient to learn, and nd free of spurious patterns that harm generalization. Additionally, each step should fully exploit parallelism when possible, i.e. maximizing the degree of parallelism for those parallelizable steps whilie having the freedom to adjust for inherently serial steps. Our paper demonstrates that ARM and vanilla MDM are fundamentally unsuitable of representing such processes for many problems.
>
> For obtaining optimal trajectories: when available, historical revision data (e.g., git commit history, paper drafts, proof revisions) provides natural supervision. When such data is limited, one approach is supervised fine-tuning on curated examples for cold-start, followed by reinforcement learning to discover optimal strategies. Algorithmic heuristics such as minimum edit distance or maximum information gain per step might work, but do not fully capture the notion of optimality discussed above.

---

> > ### Author Response · Authors · 2025-11-16
> >
> > > **Question 2: "If edit trajectories become first-class data, would you formulate any-process training with an ELBO? Do you see likelihood-based evaluation and probabilistic queries over edits becoming standard practice?"**
> >
> > While the ELBO formulation of any-process training seems nice to have, but it is limited in the sense that it is insufficient to capture task dependent structures that determine the generation behavior including mask ratio schedule, adapative paralleism degree, etc. More concretely, for example, probalistic decompositions do not take into account computational hardness, i.e. whether different trajectories are efficiently learnable by the Transformers we use in practice; trajectory may be probabilistically coherent yet computationally intractable to learn. The gap between likelihood-based evaluation and practical performance has also been reported in some other papers (e.g. [1]). Therefore, we suggest that future training frameworks for diffusion LLMs should go beyond standard ELBO formulations to incorporate more fine-grained measure of the optimality of generation process, which is important, as we discussed in the paper, for current LLMs to solve harder tasks in more general domains.
> >
> > [1] Generalized interpolating discrete diffusion (ICML 2025)

---

### Official Review · Reviewer_Fhff · 2025-11-01

**Soundness:** 3
**Presentation:** 2
**Contribution:** 3
**Rating:** 6
**Confidence:** 4

**Summary:**

This paper proposes a new framework for generative modeling, in which the model is allowed to perform extra operations such as remasking and insertion (rather than solely unmasking/next-token prediction). This more flexible generation process allows computational circuits of higher complexity to be representable, which the authors compare rigorously against masked diffusion models and autoregressive models. Furthermore, the authors demonstrate experimentally on simple tasks that their method appears much more robust  and seems to learn more efficient representations/solutions.

**Strengths:**

1. I think it is a good idea to investigate various generation strategies and their complexities/capacities in a way that is fairly model-agnostic. The authors do exactly this; by adopting a complexity-theoretic language they are able to show separations in a way that does not depend on parameterization. The reason for improvement is clear, and they present it clearly.

2. The experimental results are convincing. The authors' proposed modification to the generation process sounds reasonable and makes intuitive sense, but without definitive empirical proof it doesn't mean as much. In this case, I feel that the authors are able to demonstrate (despite the small scale) that their method is learning qualitatively better solutions (as evidenced by e.g. length generalization in Figure 4b). Of course, while it would be nice to see how things fare at scale, this is the first step. My philosophy is that if a change in the model only slightly increases metrics on synthetic tasks instead of showing clear separations, it isn't as interesting to scale it up. The experiments in this paper certainly indicate that something new is here, but the question of how it does at large remains open.

**Weaknesses:**

I think it is hard to decipher from the theorem statements (and also from the proofs, at least for me) exactly *which* types of problems are hard for unidirectional sequential models but easy for your type of model. Sudoku makes natural sense, and it was no surprise to see the proposed method do so well on that task. However, it feels like there is a big gap between the complexity-theoretic statements and what is demonstrated empirically.

In my life I have found it hard to convince people to run large-scale experiments to test modeling changes on the general basis of expressivity and larger capacity; for me it is always much more successful to have a crystal clear problem instance (such as modeling code with parenthesis, ...) where it is clear to demonstrate (both theoretically and empirically) that other methods fail and your method is the fix. Having an example/mascot problem like this gives you a kind of language to explore other problems in, and it helps tie together the different perspectives for your reader.

Reading this paper, I felt it difficult to really track the expressivity improvement between the theory and the experiments. The extra flexibility is clear to see intuitively, but the whole point of the theory + empirics is to qualify and quantify exactly what the gains of your method are. Sorry for the long rant, but my main point is that it would be more convincing and gratifying to read if the connections between the theoretical improvements and the experimental results could be explored at a finer scale.

**Questions:**

1. The authors are looking through a perspective of expressivity/capacity, which is fine and often the right starting point. I am, however, interested if the authors found any qualitative differences in training dynamics/hyperparameter choices/... between their method and ARM or MDM? Anecdotally, does it feel similar training them? I ask because sometimes when a model is, for example, way more robust to learning rate, it's indicative of what kinds of extra structure its baking into its solution.

2. It would be nice to see experiments at larger scale. This is always the case though, isn't it :)

---

> ### Author Response · Authors · 2025-11-16
>
> > **Weakness: "The connections between the theoretical improvements and the experimental results could be explored at a finer scale."**
>
> We mainly have discusssed two advantages of AP-MDM over ARM and the vanilla MDM through the lens of expressive power, which we summarize as follows (with detailed connections between theoretical and experimental results):
>
> **1. Scalability to harder reasoning tasks**: Theorem 2 states vanilla MDM (and ARM) can only solve problems with time complexity $\tilde{\mathcal O}(S^3(n))$ (where $S(n)$ is context length). Many hard reasoning tasks (whose time complexity significantly exceeds space complexity) are thus intractable: e.g. with polynomial context length, MDM and ARM can only solve problems in $\mathsf{P}$. In contrast, Theorem 4 shows AP-MDM, with remask capability, can solve any problem whose space complexity is $\mathcal O(S(n))$: e.g. with polynomial context length, AP-MDM can solve all $\mathsf{PSPACE}$ problems (note we have $\mathsf{P}\subseteq \mathsf{NP} \subseteq \mathsf{PSPACE}$).
>
> Generalized Sudoku is a representative $\mathsf{NP}$-complete problem that ARM and vanilla MDM provably cannot solve when context length is proportional to the grid size, but AP-MDM can, as predicted by Theorems 2 and 4. Indeed, Table 3 shows AP-MDM achieves 99.28% accuracy with only 100 training instances, sigficantly outperforming baselines.
>
> **2. Generality to domains with non-sequential objects (coding / science)**: Another advantage of AP-MDM is that it allows the model to handle objects whose evolution process is inherently non-sequential. A formalization of this is Theorem 5, which states that generating two-sided Dyck-k (matched parentheses), ARM can not exactly cover the support beyond a length threshold, whereas AP-MDM can. More, this advantage is unique to AP-MDM's editing capabilities: while ARM can simulate any computation vanilla MDM performs (Theorem 3), it provably cannot simulate AP-MDM (Theorem 6) due to the complexity introduced by editing operations.
>
> While we haven't considere Dyck-k language in experiments, we verify this point through a challenging graph editing task (Figure 4a), where AP-MDM significantly outperforms ARM. When ARM attempts to simulate AP-MDM's editing process (following Theorem 6's construction), training becomes increasingly difficult as sequence length grows, which directly verifies Theorem 6.
>
> **Besides of these two advantages**, we proved that the vanilla MDM is computationally universal and achieves optimal parallel time complexity (Theorem 1), which also holds for AP-MDM. Empirically, AP-MDM also has benefits beyond the expressiveness theory, e.g. length generalization in parity.
>
>
>
> > **Question 1: "Any qualitative differences in training dynamics/hyperparameter choices/... between their method and ARM or MDM?"**
>
>
> When controlling for other factors (model size, sample size, learning rate, etc.), we find AP-MDM generally converges faster and achieve smaller loss due to its expressiveness advantage. In addition to that, our approach is significantly sample more efficient, e.g. in the Sudoku task we only use 100 instances: this is because AP-MDM can decompose the reasoning process into more modular steps, and each individual step (i.e. filling one position, backtracking when encountering conflicts) is easier to learn than the end-to-end process required by baselines. We also find AP-MDM relatively insensitive to hyperparameter choices across our experiments.
>
>
> > **Question 2: "It would be nice to see experiments at larger scale."**
>
> Thanks for the suggestion. We discussed how AP-MDM could scale in practice for both pretraining and fine-tuning in Appendix C but will leave scaling up as future work.

---

### Official Review · Reviewer_zY1v · 2025-11-01

**Soundness:** 2
**Presentation:** 4
**Contribution:** 2
**Rating:** 4
**Confidence:** 5

**Summary:**

This paper studies alternative generation paradigm that breaks the AR hegemony. It argues "natural" process that include editing and complete rewriting the intermediate states offer concrete benefits over AR and Mask-dLM. It first formalizes Mask-dLM's power and limits, then proposes Any-Process dLM supporting masking, unmasking, insertion and deletion. AP-dLM is a natural ability-superset model to existing alternatives and strictly expands solvable problems under practical resource bounds. Primitive empirical results on Sudoku, parity and graph editing confirms the proposed method as a proof-of-concept.

**Strengths:**

1) The clarity of the process-level framing and the proposed AP-dLM is good.
2) The demonstrated time-optimal PRAM simulation for Mask-dLM and time-space-optimal one for AP-dLM is mind-freshing.
3) The experiments are compellingly testifying the core claimed benefits at a small scale.

**Weaknesses:**

1) Please pardon me if I'm being too harsh on this - it claims to have solved everything but really nothing in practice - one of the major issues in language modeling is always about scaling. There's little, if any, discussion on how the proposed method could even scale-up to a medium level. If we break down language generation into a few rather heterogeneous mini-procedures, the two most important steps are:

 - Context Encoding, meaning that how the context information is collected and compressed;
 - Action Prediction, meaning how the representation vectors are then transformed into further actions that alters the context.

The real question in NLG ever since the N-gram era really has been that, given a sequence of N tokens that can be sufficiently long, how to balance between being practically efficient in encoding context and predict the next growth of context that holds the **monotonicity** of context to some extent, such that we can somewhat efficiently reuse the previous computation when the context is altered by the generation step. I don't think the community is unaware of the unnaturalness of AR all along, most scaled-up practices uses it only because we can easily train very large and thus generally capable AR models at ease.

2) Continuing the previous discussion, talking about data efficiency alone without carefully check how efficient can the proposed model utilize the data can be less meaningful. External validity beyond toy settings remains to be shown.

3) A lot of existing efforts in building non-Mask/insertion-based dLMs that are moderately scaled up, such as Insertion Transformer, InDIGO, Non-Monotonic Sequential Text Generation, XLNet, Levenshtein Transformer and InsNet are missing, showing the potential insufficient literature review by the authors.

**Questions:**

1) I don't think that when insertion operations and deletion operations are included, their conceptual "addresses" can be trivially provided by any existing position encoding without having to trigger the complete re-encoding of context. This caused my confusion when trying to understand THM 10. Can the authors provide more details on this?

---

> ### Author Response · Authors · 2025-11-16
>
> > **Weakness 1:  "One of the major issues in language modeling is always about scaling. There's little, if any, discussion on how the proposed method could even scale-up to a medium level."**
>
> The training of MDMs is indeed inefficient in terms of the amount of computes (i.e. FLOPs) per sequence because of the need to re-encode the entire sequence when context changes (whereas ARM can reuse KV cache). However, this is a (well-known) limitation of any off-the-shelf MDMs with bidirectional attention with or without the remask/insert/delete operations we introduced. Addressing this remains a open research question (under active investigation e.g. [1]), which is beyond the scope of this paper.
>
>
> It is also worthwhile to note that despite the inherent limitation of MDM training inefficiency, there are many advantages in return:
>
> **1. Better exploitation of data**: Harder training procedure enables exploring different generation processes, potentially finding optimal ones that make inference easier, as also raised by [2]. For example, on Sudoku, ARM with efficient training provably cannot achieve zero training loss due to expressivity limitations (Theorem 2); in contrast, AP-MDM's more expensive training allows learning optimal processes (difficulty-based position selection and backtracking) that make the problem solvable.
>
> **2. Scaling-up computes in data-constrained scenario:** Following the above point, in practical scenarios where the amount of data (rather than the training compute budget) is the bottleneck [3], MDM can effectively utilize more training compute per sample to achieve better performance, as evidenced by recent works [4,5].
>
> **3. Better wall-clock time**:  Despite the obvious higher training and inference FLOPs, there is growing interest in large-scale diffusion LLMs because they can fully leverage parallel computation (i.e. the re-encoding of each token can be done in parallel) and thus achive optimal decoding steps or wall-clock response time (Theorem 1) that compensates for the increased compute cost.
>
> **4. Power beyond natural language**: Our introduced remask/insert/delete operations further enhance MDM's ability to solve harder reasoning tasks and handle objects under structural constraints prevalent in coding and science. Given that large-scale MDM training has proven feasible (e.g., Gemini Diffusion [6]) despite similar training difficulties, we envision any-process generation having similar scaling potential, with additional representational power.
>
> While scaling up is not the main focus of this paper, we comprehensively discuss how AP-MDM could scale in practice for both pretraining and fine-tuning in Appendix C. Moreover, some concurrent works have already scaled up such editing capabilities to large models: FlexMDM [6] fine-tunes an 8B model for insertion, and DreamOn [7] fine-tunes a 7B model for expansion and deletion. In comparison, we provides the first theoretical understanding of the expressivity of MDM and "assembles those pieces (in concurrent works) into a coherent, learnable edit process" (quote from Reviewer qEHt).
>
>
> [1] dkv-cache: The cache for diffusion language models. (NeurIPS 2025)
>
> [2] Train for the Worst, Plan for the Best: Understanding Token Ordering in Masked Diffusions. (ICML 2025, outstanding award)
>
> [3] Will we run out of data? Limits of LLM scaling based on human-generated data. (ICML 2024)
>
> [4] Diffusion Beats Autoregressive in Data-Constrained Settings. (NeurIPS 2025)
>
> [5] Diffusion Language Models are Super Data Learners (2025)
>
> [6] https://deepmind.google/models/gemini-diffusion/
>
> [7] Any-Order Flexible Length Masked Diffusion (2025)
>
> [8] https://hkunlp.github.io/blog/2025/dreamon/
>
>
> > **Weakness 2: "Continuing the previous discussion, talking about data efficiency alone without carefully check how efficient can the proposed model utilize the data can be less meaningful. External validity beyond toy settings remains to be shown."**
>
>
> As discussed above, computational efficiency during training is an inherent limitation shared by all off-the-shelf discrete diffusion models, regardless of the proposed operations, and this is often an essential trade-off for better inference performance.
>
> However, we have empirically verified the sample efficiency benefit of AP-MDM, i.e., how effectively our approach learns from limited data samples. Our results demonstrate strong sample efficiency: On Sudoku, AP-MDM trained on 100 instances outperforms ARM and AO-MDM trained on 1.8M instances; on Parity, AP-MDM trained on length-2 samples generalizes to arbitrary lengths, whereas ARM fails regardless of training data size. Due to the theoretical nature of this paper, we will leave verification on larger scale models as future work.
>
> > **Weakness 3: Insufficient literature review**
>
> Thanks for the suggestion. We have added comprehensive discussions of these and other related works to the related work section. The modified part is highlighted in blue.

---

> > ### Author Response · Authors · 2025-11-16
> >
> > > **Question 1: "I don't think that when insertion operations and deletion operations are included, their conceptual "addresses" can be trivially provided by any existing position encoding without having to trigger the complete re-encoding of context. This caused my confusion when trying to understand THM 10"**
> >
> > Again, re-encoding is triggered every time context is updated, which is the default setting and inevitable (at least currently) for all diffusion LLMs. All theorems are proven (including Theorem 10) based on this setting.
> >
> > However, the main message of Theorem 10 is orthogonal to whether re-encoding is triggered or not. It states that with the capability to remask alone (i.e., converting decoded tokens back to mask tokens without position shifts), the model can solve all problems whose space complexity is bounded by the context length (e.g. PSPACE with polynomial context length). In contrast, without remasking capability, vanilla MDM (according to Theorem 2) and ARM can only solve problems whose time complexity is bounded by the context length (e.g. P with polynomial context length). This demonstrates why remasking is essential for harder reasoning problems.

---

> ### Comment · Reviewer_zY1v · 2025-11-28
>
> Thanks for the thoughtful response. While I generally agree that the inefficient scalability is a shared curse for most off-the-shelf MDMs, I do want to kindly bring this to your attention that the proposed any-order generation, that shares many properties with other non-monotonic generation techniques especially the whole insertion-based generation idea, brings problems that are less clearly or not-at-all discussed in this paper. The biggest one, for example, is the position encoding.
>
> If we take the historical SOTA AR/Encoder models as our observation lens, we have had three paradigms of position encoding: 1) Absolute position encoding, such as the learnable ones in BERT, or sinusoidal as in GPT/GPT-2; 2) relative position encoding that assumed the stationary offset between contextual tokens, which actually does not allow insertion/deletion without re-encoding by design (such as XLNet, Google T5, ALiBi);  3) Semi-relative position encodings that while still directly applied to word embeddings, their interaction with the context is designed to be absolute-position-invariant. The rotary position encoding (RoPE), for example, lies under such a paradigm. I believe this is a totally unanswered question in the paper. How should the position information be encoded in the proposed any-process generation? Of course assuming the current tokens to be consecutive without thinking too much on them is a way, but please note that this will break one of the core assumptions in the effectiveness of RoPE that the information pattern in tokens is to some extent **periodic**, thus destructively introduce noise to the training signals, leading to impossible fulfillment of the claimed improvement in representation quality if you train the model at scale. To be honest, I don't believe this is a trivial question that can either be empirically overlooked as the authors claim or that they have already carefully considered and been ready for an answer.
>
> Note that, existing MaskLM/MCMC-based diffusion language models does NOT have this problem, because the "offset" between tokens in such models are not changed as there are only substitutions of tokens. Thus, the training signal for such models are not as noisy as the case in the proposed any-process generation, also making it a lot more friendly for people to finetune a Mask-LM-based dLLM from AR-LM/xxBERT models. This saves a lot of efforts in scaling up such models, and that in my opinion is why people are not very concerned about the training scalability of such Mask-dLLMs (we can always train foundation models efficiently at scale then finetune them as Mask-dLLMs).
>
> On the other hand, if the authors would prefer avoid the discussion on the architectural changes (in my opinion it's just naturally needed for the proposed any-process generation) and rather continue to use RoPE-based transformers, it's also rational to me. However, if this should be case, I don't see any significant difficulty to simply finetune an any-process generator from a more recently published BERT model (such as RoBERTa/DeBERTa/ModernBERT) and do some language generation experiments. To be honest, many of the any-process generation idea originated from the authors' (and the community's) thoughtful understanding of the compositional and recursive nature of language, it's weird to me to not even explore this idea on one language generation task at all.
>
> I appreciate the value of the paper as an empirical (thought toy-ish) proof to the idea beyond stationary-offset language generation (such as AR-LM and Mask-dLLM). But to me this draft still has fundamental empirical weaknesses and issues in how the authors organize their writeup and make better usage of the paper capacity. I would not be against it if the current draft still get accepted, but I sincerely recommend major revisions before this paper goes public.
>
> With all such criticism being said, I still want to thank the authors for their hard work in exploring better generation paradigms.

---

> ### Author Response · Authors · 2025-12-02
>
> Thanks for the feedback. The introduction of editing operations indeed raises questions about how to deal with the position shift. We adopt a straightforward solution: whenever insertion or deletion occurs, we account for the position shift and re-encode the entire updated sequence. This choice is naturally compatible with off-the-shelf PE methods, causes no training instability in our experiments, and is essential for AP-MDM's computational benefits beyond ARM as our theoretical and empirical results demonstrate.
>
> First we want to emphasize that this approach introduces no additional computational overhead compared to existing MDMs. The encoder-only architecture already requires re-encoding the entire sequence whenever context changes, regardless of whether positions shift. Therefore, computational efficiency is not compromised. The reviewer's concern that position shifts might introduce noise and destabilize training is understandable, but our results suggest the opposite:
>
> In Theorems 3 and 6, we showed that ARM can simulate any-order MDM and solve any problems that any-order MDM can solve, but simulating AP-MDM is significantly harder, and thus AP-MDM can potentially address a broader class of problems. **One main reason for this separation is exactly that the position shift (with re-encoding) in AP-MDM empowers the model to represent more complicated processes.** More specifically, consider the PRESERVE problem (Definition 15 in Appendix M), which asks: given an input sequence and after a series of deletion and insertion operations, what is the remaining token at the $j$-th position? This can be easily solved by AP-MDM with position re-encoding; the model simply checks the final token content at the $j$-th position. In comparison, ARM and vanilla MDM, which associate each token with a fixed position, must simulate this process and compute the result internally in the forward pass, which is provably beyond current Transformers' expressive power. Without re-encoding positions, AP-MDM is also infeasible to achieve this.
>
> **The above result directly implies the training advantage of AP-MDM compared with ARM, especially for objects whose natural evolution process involves editing.** For instance, in our graph editing experiments, when controlling for all other factors (model size, sample size, learning rate), AP-MDM converges faster and achieves lower loss than ARM, with no training instability observed. This is because insertion/deletion with position re-encoding allows AP-MDM to naturally handle objects with global constraints (e.g., graph structures, code with syntax constraints) without needing to internally compute state evolution (which is computationally impossible) or periodically summarize the current state (which is extremely inefficient).
>
> In other words and more intuitively, **position shifts in AP-MDM are not training noise but meaningful signals reflecting the true structural evolution of objects.** Consider code generation: when inserting a new function definition, all subsequent code shifts in position, and AP-MDM naturally updates positions to reflect this structural change, allowing the model to correctly reference the updated code structure. In contrast, ARM with fixed positions must internally track which 'logical position' each token occupies after insertion. While we have not conducted large-scale experiments in natural language to test training stability due to this paper's theoretical focus, several concurrent works have done so (also with position re-encoding) with promising results [7,8].
>
> We thank the reviewer for the valuable feedback on presentation and organization. We have revised the paper to: (1) explicitly discuss position encoding implementation and its role in AP-MDM's expressivity, (2) clarify training efficiency trade-offs including KV cache limitations of MDM, (3) strengthen connections between theory and experiments (as suggested by Reviewer zY1v), (4) discussion about more related work. The revised parts are highlighted in blue.

---

### Meta-Review · Area_Chair_2DjJ · 2026-01-02

**Summary:**

This paper presents a new framework for masked diffusion models by introducing process indicators that allow the model to remask, insert, and delete tokens. This any-process generation paradigm offers a novel perspective on sequence generation and demonstrates promising potential, particularly for tasks such as reasoning. In recognition of the paper’s contributions at both the framework and theoretical levels, I am inclined to accept this work.

At the same time, I share the reviewers’ concerns regarding scalability to language generation tasks. I encourage the authors to include experiments with a medium-scale model in the camera-ready version to better demonstrate language generation capability and scalability.

**Reviewer Concerns:**

Addressed:
1) Lack of literature.
2) Detailed explanation.
3) Ablations on actions.


Still outstanding:
1) The scalability of the model.
2) The evaluation of language generation tasks.
3) The effect of position shifts.

**Reviewer Scores:**

Reviewer zY1v: The score would not be changed since the core concerns about scalability and language generation ability are still outstanding.

Reviewer Fhff: The score would be kept positive as 6 for the informative feedback.

Reviewer qEHt: The score would be kept positive as 8.

---

### Decision · Program_Chairs · 2026-01-26

Accept (Poster)